# SCALABLE BAYESIAN MONTE CARLO: FAST UNCERTAINTY ESTIMATION BEYOND DEEP ENSEMBLES

## ABSTRACT

This work introduces a *new method* designed for Bayesian deep learning called scalable Bayesian Monte Carlo (SBMC). The method is comprised of a *model* and an *algorithm*. The model interpolates between a point estimator and the posterior. The algorithm is a *parallel* implementation of sequential Monte Carlo sampler (SMC$_\parallel$) or Markov chain Monte Carlo (MCMC$_\parallel$). We collectively refer to these *consistent* (asymptotically unbiased) algorithms as *Bayesian Monte Carlo* (BMC), and any such algorithm can be used in our SBMC method. The utility of the method is demonstrated on practical examples: MNIST, CIFAR, IMDb. A systematic numerical study reveals that *for a comparable wall-clock time to state-of-the-art (SOTA) methods like deep ensembles (DE)*, SBMC achieves comparable or better accuracy and substantially improved uncertainty quantification (UQ)–*in particular, epistemic UQ*. The benefit is demonstrated on the downstream task of estimating the confidence in predictions, which can be used for reliability assessment or abstention decisions. Code is available in the supplementary material.

## 1 INTRODUCTION

Uncertainty quantification (UQ) in deep learning is critical for safe and reliable deployment, yet remains a core challenge. The Bayesian formulation provides UQ in addition to Bayes optimal accuracy, by averaging realizations from the posterior distribution, rather than relying on a single point estimator. Fully Bayesian approaches like consistent Markov chain Monte Carlo (MCMC) and sequential Monte Carlo (SMC) offer asymptotically unbiased posterior estimates, but at the cost of prohibitive compute time compared to simple point estimators like the maximum a posteriori (MAP). Bayesian deep learning (BDL) often rely on scalable approximations such as Monte Carlo Dropout (Gal & Ghahramani, 2016), deep ensemble (DE) (Lakshminarayanan et al., 2017) [1], (KFAC-)Laplace approximation (Daxberger et al., 2021; Eschenhagen et al., 2021), Stochastic Weight Averaging (SWA) (Izmailov et al., 2018), SWA-Gaussian (SWAG) (Maddox et al., 2019; Wilson & Izmailov, 2020), which are fast and provide strong empirical performance, but lack formal consistency guarantees.

Given data $\mathcal{D}$, the Bayesian posterior distribution over $\theta \in \Theta \subseteq \mathbb{R}^d$ is given by

$$\pi(\theta) \propto \mathcal{L}(\theta)\pi_0(\theta), \tag{1}$$

where $\mathcal{L}(\theta) := \mathcal{L}(\theta; \mathcal{D})$ is the likelihood of the data $\mathcal{D}$ and $\pi_0(\theta)$ is the prior. The Bayes estimator of a quantity of interest $\varphi : \Theta \to \mathbb{R}$ is $\mathbb{E}[\varphi|\mathcal{D}] = \int_\Theta \varphi(\theta)\pi(\theta)d\theta$. It minimizes the appropriate Bayes risk at the population level and as such is Bayes optimal (MacKay, 1992; Neal, 2012; Andrieu et al., 2003; Bishop, 2006).

Here we present a new approximate inference method called Scalable Bayesian Monte Carlo (SBMC), which bridges the gap between fast but heuristic methods and principled yet expensive samplers. It is a general method comprised of an approximate *model* and an *algorithm* to simulate from the model. Our key insight is a model approximation $\overline{\pi}_s$ (defined precisely in equation 3), featuring a scalar interpolation parameter $s \in (0, 1)$ that allows tuning between the MAP estimator ($s = 0$) and the full Bayesian posterior ($s = 1$). For smaller $s$ the target is *easier to simulate from*, albeit

---

[1] It is well-known that deep ensembles do not provide a consistent approximation of the posterior (Wild et al., 2023), yet Bayes is arguably the best lens through which to view them (Wilson & Izmailov, 2021).

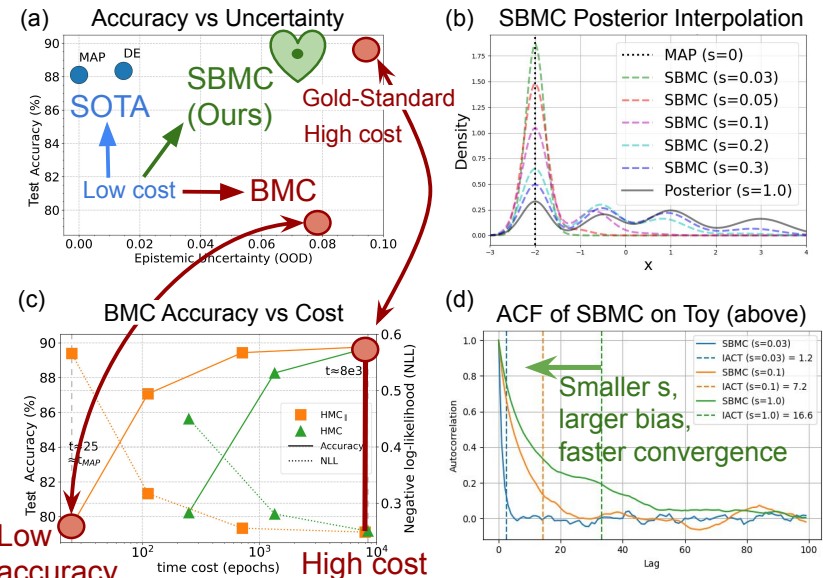

Figure 1: **Left panels**: IMDb sentiment classification. **(a)** SBMC provides a good *balance* of accuracy and UQ (quantified by epistemic entropy on OOD data), *for a comparable cost to deep ensembles* (every method runs for 25 epochs except the Gold-Standard (GS) BMC solution, which runs for 8000 epochs). **(c)** Standard implementation of HMC and HMC$_\parallel$. BMC methods typically deliver high accuracy for high cost (GS) and low accuracy for low cost. **Right panels**: SBMC *approximate models*, on a simple toy example. **(b)** The original posterior ($s = 1$) and the approximations for a range of $s$. **(d)** The autocorrelation function (ACF: correlation between samples separated by 'Lag' steps – this and integrated autocorrelation time (IACT) are defined in Appendix D.2) of SBMC for very long NUTS Hoffman et al. (2014) chains for a few choices of $s$. As $s$ decreases the target becomes simpler and hence easier to explore, but the bias (with respect to the posterior) increases.

with a larger bias with respect to the posterior. See the right panels of Figure 1. By simulating from this approximate target with *parallel* implementations of BMC algorithms, which we will denote by S-SMC$_\parallel$ and S-MCMC$_\parallel$, SBMC delivers strong performance in accuracy and UQ *at a comparable cost to SOTA methods like DE* (less than double). The prefix "S-" is for "scalable", and the scalability comes from the model approximation *in tandem* with the parallelism, denoted by the subscript $_\parallel$. Without the model approximation, the required simulation time is prohibitive.

In general the posterior (target) distribution can only be evaluated up-to a constant of proportionality, and the available consistent methods for inference (learning) are of Monte Carlo type: notably Markov chain Monte Carlo (MCMC) (Metropolis et al., 1953; Hastings, 1970; Duane et al., 1987; Gelfand & Smith, 1990; Geyer, 1992; Robert et al., 1999; Roberts & Tweedie, 1996) and sequential Monte Carlo (SMC) samplers (Jarzynski, 1997; Berzuini & Gilks, 2001; Del Moral et al., 2006; Dai et al., 2022; Chopin et al., 2020). The past several decades have seen enormous progress in methodology as well as practical applications (Galison et al., 2022; Mohan & Scaife, 2024), however standard implementations of these algorithms are still too expensive for practical BDL, and so BMC algorithms are typically used only as a benchmark for cheaper approximations (Izmailov et al., 2021). See e.g. (Angelino et al., 2016; Papamarkou et al., 2024) for recent reviews and further references. The present work aims to address this computational intractability by (i) targeting an *approximation* of equation 1, and (ii) distributing the BMC workload across many workers in parallel. We will show that these two things together provide a *practical and scalable method*. The focus of the present work is on demonstrating the value of the SBMC method itself, independently of the particular BMC algorithm used, and so we mostly focus on standard implementations of HMC and SMC. But one of the virtues of SBMC is its extensibility: stochastic gradient MCMC methods Welling & Teh (2011); Chen et al. (2014) and/or other data-parallel techniques Angelino et al. (2016); Maclaurin & Adams (2014); Rendell et al. (2020) and more sophisticated adaptive methods Hoffman et al. (2021) can be swapped in later for additional gains.

---

[1]Autocorrelation function (ACF) and integrated autocorrelation time (IACT) are defined in Appendix D.2.

The contributions of the present work are concisely summarized as follows:

- New SBMC method (e.g. S-SMC$_\parallel$ and S-MCMC$_\parallel$) targets a model which allows the practitioner to *interpolate* between the MAP (or another point) estimator for $s = 0$ (0 additional simulation time) and the full posterior for $s = 1$ (long simulation time), thus balancing their UQ demands against their budget.

- A thorough systematic empirical evaluation of SBMC on several benchmarks demonstrates that it achieves excellent performance on both accuracy and UQ *at a cost comparable to DE, where traditional BMC methods fail severely*, demonstrating its strong scalability and robustness. See the top left panel of Figure 1.

- This benefit is illustrated on the downstream task of estimating prediction confidence, which can be used to improve safety and reliability. To that end, a meta-classifier is built using seven features of the SBMC posterior which characterize the epistemic uncertainty.

The paper is organized as follows. In Section 2, we introduce the SBMC method. In Section 3 we discuss its UQ abilities and the downstream task of output confidence prediction, as motivation, and present the main results. Section 4 discusses related literature. Section 5 presents the conclusion and additional discussion.

## 2 SCALABLE BAYESIAN MONTE CARLO (SBMC) METHOD

We define *time cost* as the required *simulation time per chain/particle*, and we will measure this by *epochs*, i.e. likelihood plus gradient evaluations, as a *hardware-agnostic proxy* for wall-clock time. Parallel implementations of consistent BMC algorithms like SMC$_\parallel$ and HMC$_\parallel$ improve time cost with near linear speed-up (Liang et al., 2025), but each process still needs to run for a long time, as seen in Figure 1 (c). This section introduces the model and algorithm choices that define the SBMC method in Algorithm 1, which delivers improved performance on metrics of interest *for a comparable time cost to deep ensembles.*

---

**Algorithm 1** SBMC method

---

**Inputs**: $\mathcal{L}, \pi_0, s, N, P$.

**Compute** $\theta_{\mathsf{MAP}}$, and create $\overline{\pi}_0, \overline{\pi}$ as in equation 2, equation 3.

**for** $p = 1$ **to** $P$ (in parallel) **do**

    **Run** Algorithm 2 (S-SMC) or 3 (S-MCMC).

    **Output**: $\{\theta^{i,p}\}_{i=1}^N$ and $Z^{N,p}$.

**end for**

Build $\hat{\varphi}_{\mathsf{SBMC}} = \frac{\sum_{p=1}^P Z^{N,p} \frac{1}{N} \sum_{i=1}^N \varphi(\theta^{i,p})}{\sum_{p=1}^P Z^{N,p}}$.

---

**The model.** Assume the prior is $\pi_0 = \mathcal{N}(0, V)$ for simplicity and define the MAP estimator as

$$\theta_{\mathsf{MAP}} = \mathsf{argmax}_\theta \mathcal{L}(\theta; \mathcal{D})\pi_0(\theta),$$

where $\mathcal{L}$ is the likelihood defined in equation equation 1. For a fixed tuning parameter $s \in (0, 1)$, we define $0 \prec \Sigma(s) = \Sigma(s)^\mathsf{T} \in \mathbb{R}^{d \times d}$ and $\alpha(s) \in [0, 1]$, such that $\Sigma(0) = 0$ and $\Sigma(1) = V$, and $\alpha(0) = 1$ and $\alpha(1) = 0$. Define the new prior as

$$\overline{\pi}_0(\theta) = \mathcal{N}(\theta; \alpha(s)\theta_{\mathsf{MAP}}, \Sigma(s)). \tag{2}$$

The SBMC method then targets the following distribution

$$\overline{\pi}(\theta) \propto \mathcal{L}(\theta)\overline{\pi}_0(\theta), \tag{3}$$

which we will refer to as the *anchored* posterior. We will refer to $\theta_{\mathsf{MAP}}$ as the *anchor*.

For $s \to 0$, we recover a Dirac measure concentrated on the MAP estimator, which means no sampling is required. Conversely, as $s \to 1$, we recover the *original posterior*. Hence $s$ is a scalar interpolation parameter which allows us to tune between these limits. For simplicity we will typically consider only the standard isotropic case $V = v\mathsf{Id}$ and let $\alpha(s) = \mathbf{1}_{\{s < \frac{1}{2}\}}$ and $\Sigma = sv\mathsf{Id}$.

We show that this approximate model balances the complementary strengths of the two approaches for small $s$, and enables BMC methods to deliver *scalable gains over alternatives like deep ensembles at a comparable cost.* The method is relatively insensitive to the exact value of $s$ and we recommend a default value of $s = 0.1$. We will use the notation S-SMC$_\parallel$ and S-MCMC$_\parallel$ to distinguish the SBMC method from standard implementations of the algorithms targeting equation 1. For example, S-SMC$_\parallel$ means the SMC$_\parallel$ algorithm is used to sample from equation 3.

---

**Algorithm 2** S-SMC sampler

---

**Inputs**: $\mathcal{L}, \overline{\pi}_0, N$.

Init. $\theta_0^i \sim \overline{\pi}_0$ for $i = 1, \ldots, N$. $Z^N = 1$.
**for** $j = 1$ **to** $J$ (in serial) **do**
    (Optional) Select $\lambda_j$ s.t. ESS= $\rho N, \rho < 1$.
    **for** $i = 1$ **to** $N$ (in parallel) **do**
        Define $w_j^i \propto \tilde{w}_j^i \equiv \mathcal{L}(\theta_{j-1}^i)^{\lambda_j - \lambda_{j-1}}$.
        **Selection**: $I_j^i \sim \{w_j^1, \ldots, w_j^N\}$.
        **Mutation**: $\theta_j^i \sim \mathcal{M}_j(\theta_{j-1}^{I_j^i}, \cdot)$.
    **end for**
    Store $Z^N \leftarrow \frac{1}{N} \sum_{i=1}^{N} \tilde{w}_j^i$ .
**end for**

**Outputs**: $\{\theta^i = \theta_J^i\}_{i=1}^N$ and $Z^N$.

---

**The algorithm** can be *any BMC method*. In the present work we will focus on SMC sampler and MCMC, but any alternative is admissible. For example, SG-MCMC or other methods which allow mini-batch gradients may be quite convenient for managing the memory requirements of very large problems.

**SMC sampler.** Define a sequence of intermediate targets $\overline{\pi}_j(\theta) \propto \mathcal{L}(\theta)^{\lambda_j} \overline{\pi}_0(\theta)$, according to a tempering schedule $0 = \lambda_0, \ldots, \lambda_J = 1$, which will be chosen adaptively according to the effective sample size (ESS), as described in C.1 in the Appendix. The SMC sampler (Del Moral, 2004) alternates between *selection* by importance re-sampling, and *mutation* according to an appropriate intermediate MCMC transition kernel $\mathcal{M}_j$, such that $(\overline{\pi}_j \mathcal{M}_j)(d\theta) = \overline{\pi}_j(d\theta)$ (Geyer, 1992). This operation must sufficiently de-correlate the samples, and as such we typically define the MCMC kernels $\mathcal{M}_j$ by several steps of some basic MCMC kernel, leading to $L_j$ epochs (likelihood/gradient evaluations). We will employ two standard MCMC kernels: preconditioned Crank-Nicolson (pCN) (Bernardo et al., 1998; Cotter et al., 2013) and Hamiltonian Monte Carlo (HMC) (Duane et al., 1987; Neal et al., 2011). In the latter case, there are also several leapfrog steps for each HMC step contributing to $L_j$.

For a quantity of interest $\varphi : \Theta \to \mathbb{R}$, the S-SMC estimator from Algorithm 2 is given by

$$\overline{\pi}^N(\varphi) := \frac{1}{N} \sum_{i=1}^{N} \varphi(\theta^i) \xrightarrow{N \to \infty} \mathbb{E}_{\overline{\pi}}[\varphi] \approx \mathbb{E}_\pi[\varphi] = \mathbb{E}[\varphi \mid \mathcal{D}]. \tag{4}$$

**S-SMC$_\parallel$** refers to $P$ parallel executions of Algorithm 2, each with $N$ particles, leading to a $P$ times lower communication and memory overhead than a single S-SMC sampler with $NP$ samples. This simplification is crucial for massive problems such as BDL, which require distributed architectures. Synchronous Single Instruction, Multiple Data (SIMD) resources can be used for the $N$ communicating particles (and model- and data-parallel likelihood calculations), while all communication between the $P$ processes is eliminated. The S-SMC$_\parallel$ ratio estimator is defined for $P$ i.i.d. realizations $\overline{\pi}^{N,p}(\varphi)$ of equation 4 as

---

**Algorithm 3** S-MCMC

---

**Inputs**: $\mathcal{L}, \overline{\pi}_0, N$.

$\theta_0^i \sim \overline{\pi}_0^{(i)}$ for $i = 1, ..., N$.
**for** $i = 1$ **to** $N$ (in parallel) **do**
    **for** $j = 1$ **to** $J$ (in serial) **do**
        Draw $\theta_j^i \sim \mathcal{M}_J(\theta_{j-1}^i, \cdot)$.
    **end for**
**end for**

**Outputs**: $\{\theta_J^i\}_{i=1}^N$ and $Z^N \equiv 1$.

---

$$\hat{\varphi}_{\text{S-SMC}_\parallel} = \sum_{p=1}^{P} \omega_p \overline{\pi}^{N,p}(\varphi), \quad \omega_p = \frac{Z^{N,p}}{\sum_{p'=1}^{P} Z^{N,p'}}. \tag{5}$$

**S-MCMC$_\parallel$** refers to $P$ parallel executions of Algorithm 3, which already features $N$ parallel short chains free from any communication. The purpose of formulating MCMC in this way is to match SMC, which itself features $N$ parallel chains that need to communicate intermittently at the selection stage. The estimator is built exactly as equation 5, with $\{\theta^{i,p} = \theta_J^{i,p}\}_{i=1}^N$ in equation 4 and $Z^{N,p} = 1$.

## 3 MOTIVATION AND RESULTS

Here we introduce UQ to motivate SBMC. We experimentally validate the quality of the epistemic UQ it delivers by using very long HMC runs as a 'gold-standard' (GS), and we show that it can be used to *directly predict model confidence*. We also assess SBMC against SOTA competitor baselines using epistemic and first order metrics on several challenging benchmark datasets. In subsection 3.1 we demonstrate how features derived from the epistemic uncertainty can be used to build a meta-classifier for predicting model confidence and deciding whether to abstain or respond.

**UQ** is a crucial pain-point for neural networks, and BDL is one of the leading contenders to deliver it. Our primary UQ metric will be *epistemic entropy,* which is the difference between *total* and aleatoric entropy, defined as follows (Hüllermeier & Waegeman, 2021; Depeweg et al., 2018)

$$H_{\text{ep}}(x) = \underbrace{-\sum_{y \in \mathsf{Y}} \mathbb{E}[p(y|x,\theta)|\mathcal{D}] \log \mathbb{E}[p(y|x,\theta)|\mathcal{D}]}_{H_{\text{tot}}(x)} - \underbrace{\mathbb{E}\Big[-\sum_{y \in \mathsf{Y}} p(y|x,\theta) \log(p(y|x,\theta))|\mathcal{D}\Big]}_{H_{\text{al}}(x)} . \quad (6)$$

Aleatoric uncertainty is irreducible and can be thought of as label error (people may sometimes disagree on the label of a given hand-written digit), whereas epistemic entropy quantifies uncertainty which can be reduced with more data (Shaker & Hüllermeier, 2020; Krause & Hübotter, 2025). Our focus is the latter, as it is *only* captured by Bayesian methods. It can be viewed as the mutual information between parameter and predictive posterior random variables for input $x$, and as such is $0$ by definition for point estimators that yield deterministic predictive estimators.

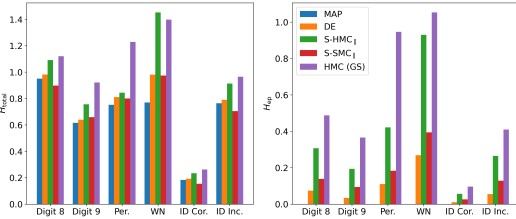

Figure 2: Average *predictive* total and epistemic entropy (having seen only ID data) over four OOD classes and correct and incorrect predictions ID for MNIST7 ($N = 10, P = 1$).

To illustrate the properties of SBMC, we conduct an experiment. The dataset is a subset of 1200 MNIST (LeCun et al., 2010) data trained on digits $0, \ldots, 7$ (MNIST7) with $8, 9$ held out as (similar but) out-of-domain (OOD). White noise inputs and randomly selected in-domain (ID) digits corrupted with white noise are considered as far OOD classes. The architecture is described in Appendix D.3.1. The prior variance is $v = 0.1$. The average predictive total and epistemic entropy for various categories of OOD test data, as well as ID test data split into correct and incorrect predictions, are presented in Figure 2 (per-digit result is given in Appendix F.1). This quantity is clearly predictive of misclassifications, and this downstream task will be revisited below. A long HMC chain is included as gold-standard (GS) for validation.

In Table 1 we compare several SOTA competitors, including the MAP (computed with SGD and early stopping on validation data), DE, MC Dropout, KFAC-Laplace approximation, SWA, SWAG, and DEI-MCMC, with (S-)HMC$_\parallel$ (S-)SMC$_\parallel$, and (S-)SGHMC$_\parallel$. All methods are run with time cost of $\approx 170$ epochs, but SBMC methods *require the MAP estimator(s)*, so their total time cost is roughly double that of the methods which do not initialize with the MAP(s). In order to also consider exactly equal cost, we ainclude runs of (S-)HMC$_\parallel$ with half as many epochs, and runs of MAP and DE with twice as many epochs. A single HMC run using $2e4 - 2e5$ epochs is also included as a GS baseline. All methods can be further parallelized with model- and data-parallel techniques, but we do not consider that here. Convergence for all methods is verified by running $5$ chains with dispersed initial conditions and measuring the standard error. Ensemble methods use $P$ independent ensembles of $N = 10$ particles, and all particles are used for estimating posterior expectations. A limited set of metrics are presented in Table 1 because of limited space. More comprehensive results, including Brier, ECE, and per-OOD-category $H_{\text{tot}}$ and $H_{\text{ep}}$, are presented in the Appendix Table 17.

The results show that when directly targeting equation 1, SMC$_\parallel$, HMC$_\parallel$, and SGHMC$_\parallel$, degrade rapidly away from convergence, to the point of catastrophic failure in first order metrics at this cost level. However, their UQ performance is still adequate–for example, some of the HMC$_\parallel$ and SGHMC$_\parallel$ $H_{\text{ep}}$ estimators are within our tolerance of $50\%$ of the GS solution (in bold). The MAP and DE quickly deliver good accuracy, but do not accurately estimate $H_{\text{ep}}$. These failure modes are perfectly *complementary*. To achieve the "best of both worlds", SBMC anchors to the MAP estimator(s) to preserve accuracy, and then uses an ensemble of short parallel runs of BMC to augment that with uncertainty. MC Dropout performs particularly well and is notably the only non-BMC method which achieves an $H_{\text{ep}}$ estimator within our tolerance of $50\%$ of the GS solution (in bold), and for a low total cost.

To account for possible beneficial impact of initialization, we consider also short DE-initialized chains (DEI-HMC) with the original target Sommer et al. (2024). Performance *significantly improves* in comparison to the prior-initialized chains, however it is still not competitive with SBMC. All HMC chains are tuned the same for a fair comparison. The details are given in Appendix E.

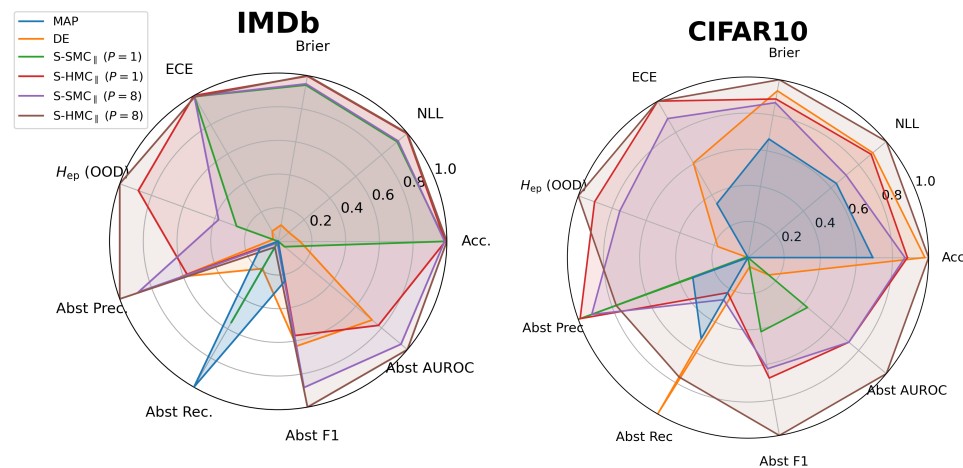

Figure 3: Test Accuracy, NLL, Brier, and ECE, $H_{\text{ep}}$ on OOD, and confidence meta-classifier abstention (Abst) metrics (all min-max scaled so 1 is best) for IMDb (left) and CIFAR10 (right).

Table 1: Comparison of methods on MNIST7 test data. SBMC methods are bold ($N = 10$). For metrics, the best gold-standard (GS) value is bold, along with others within 1% for accuracy and NLL, and 50% for entropies (entropy is harder to estimate, and also high precision is less critical). SMC$_{\parallel}$, HMC$_{\parallel}$, and SGHMC$_{\parallel}$ are highlighted in red as they are particularly bad for these very short chains, and this is precisely the problem the approximate methods like SBMC address.

| Method | P | Time Cost (epochs) ↓ | Total Cost (epochs) ↓ | Accuracy ↑ | NLL ↓ | $H_{\text{ep}}$ correct | $H_{\text{ep}}$ incorrect | $H_{\text{ep}}$ OOD |
|---|---|---|---|---|---|---|---|---|
| MAP | 1 | 160 | 160 | 92.3±0.366 | 0.253±0.012 | 0 | 0 | 0 |
| MAP | 1 | 320 | 320 | 92.1±0.264 | 0.260±0.010 | 0 | 0 | 0 |
| SWA | 1 | 160 | 160 | 92.3±0.387 | 0.27±0.017 | 0 | 0 | 0 |
| SWAG | 1 | 160 | 160 | 92.3±0.365 | 0.267±0.017 | 0.001±0.000 | 0.008±0.001 | 0.009±0.001 |
| MC Drop | 1 | 160 | 160 | **93.9±0.626** | **0.214±0.021** | 0.049±0.007 | **0.269±0.008** | 0.267±0.01 |
| Laplace | 1 | 160 | 160 | 88.2±0.235 | 0.539±0.022 | 0.504±0.036 | 0.901±0.038 | 1.22±0.033 |
| Deep Ens | 1 | 176 | 1760 | 92.4±0.150 | 0.245±0.004 | 0.011±0.000 | 0.057±0.001 | 0.123±0.011 |
| Deep Ens | 1 | 320 | 3200 | 92.2±0.157 | 0.252±0.005 | 0.007±0.000 | 0.041±0.003 | 0.104±0.013 |
| Deep Ens | 8 | 178 | 14,240 | 92.5±0.059 | 0.239±0.001 | 0.011±0.000 | 0.059±0.001 | 0.134±0.004 |
| SMS-UBU$_{\parallel}$ | 8 | 160+160 | 25600 | 92.6±0.107 | 0.247±0.002 | 0.055±0.001 | 0.201±0.001 | 0.316±0.003 |
| DEI-HMC | 8 | 176+160 | 14560 | 91.6±0.000 | 0.308±0.001 | 0.038±0.000 | 0.114±0.001 | 0.230±0.006 |
| SGHMC | 1 | 160 | 1600 | 87.7±0.742 | 0.974±0.05 | 0.652±0.031 | 0.725±0.031 | **0.883±0.036** |
| **S-SGHMC** | 1 | 160 + 160 | 1760 | 90.3±0.758 | 0.409±0.014 | 0.342±0.015 | 0.687±0.027 | **0.836±0.039** |
| **S-SGHMC$_{\parallel}$** | 8 | 160 + 160 | 12,960 | 92.3±0.160 | 0.388±0.001 | 0.434±0.005 | 0.782±0.007 | **0.965±0.003** |
| SMC | 1 | 173 | 1730 | 79.7±2.71 | 0.623±0.091 | 0.013±0.002 | 0.033±0.009 | 0.045±0.011 |
| **S-SMC** | 1 | 170 + 160 | 1860 | 92.2±0.371 | 0.267±0.014 | 0.026±0.003 | 0.129±0.014 | 0.202±0.028 |
| **S-SMC$_{\parallel}$** | 8 | 178 + 160 | 14,400 | **93.3±0.160** | **0.226±0.004** | **0.059±0.001** | **0.272±0.003** | **0.378±0.03** |
| HMC | 1 | 160 | 1600 | 78.4±2.38 | 1.27±0.085 | 0.303±0.025 | **0.325±0.026** | 0.594±0.021 |
| **S-HMC** | 1 | 160 + 160 | 1760 | **93.0±0.166** | 0.232±0.002 | **0.056±0.001** | **0.264±0.002** | 0.463±0.009 |
| **S-HMC** | 1 | 80 + 80 | 1600 | **93.0±0.057** | 0.242±0.002 | **0.059±0.001** | 0.243±0.004 | 0.417±0.013 |
| **S-HMC**(lin) | 1 | 160 + 160 | 3200 | **92.9±0.140** | 0.239±0.003 | **0.059±0.001** | 0.249±0.002 | 0.450±0.011 |
| **S-HMC$_{\parallel}$** | 8 | 160 + 160 | 12,960 | **93.1±0.085** | 0.231±0.002 | 0.070±0.000 | 0.299±0.002 | 0.531±0.011 |
| **S-HMC$_{\parallel}$** | 8 | 80 + 80 | 12800 | **93.1±0.071** | 0.237±0.001 | 0.069±0.000 | 0.272±0.002 | 0.484±0.005 |
| **S-HMC$_{\parallel}$**(lin) | 8 | 160 + 160 | 25600 | **93.1±0.069** | 0.240±0.002 | **0.073±0.000** | 0.284±0.002 | 0.518±0.011 |
| HMC (GS) | 1 | 20,000 | 20,000 | **93.6±0.415** | 0.222±0.009 | **0.096±0.004** | 0.410±0.013 | 0.768±0.084 |
| HMC (GS) | 1 | 200,000 | 200,000 | **94.8±0.211** | **0.194±0.004** | **0.120±0.004** | 0.493±0.008 | 1.04±0.122 |

For the next experiments, we look at the IMDb sentiment classification dataset (Maas et al., 2011) and the CIFAR dataset (Krizhevsky et al., 2009). Results for these cases are comparable and summarized in Figure 3 and Appendix F, along with further figures and tables. Further details on all model architectures are given in Appendix D.3.

**Small Language Model example proof of concept.** The natural next step is to apply this methodology to LLMs. Here we present some preliminary results on GPT-2, on consumer hardware (an old MacBook Pro with an M2 processor and 16GB RAM). These are early results, just to further emphasize scalability and potential utility in hallucination detection.

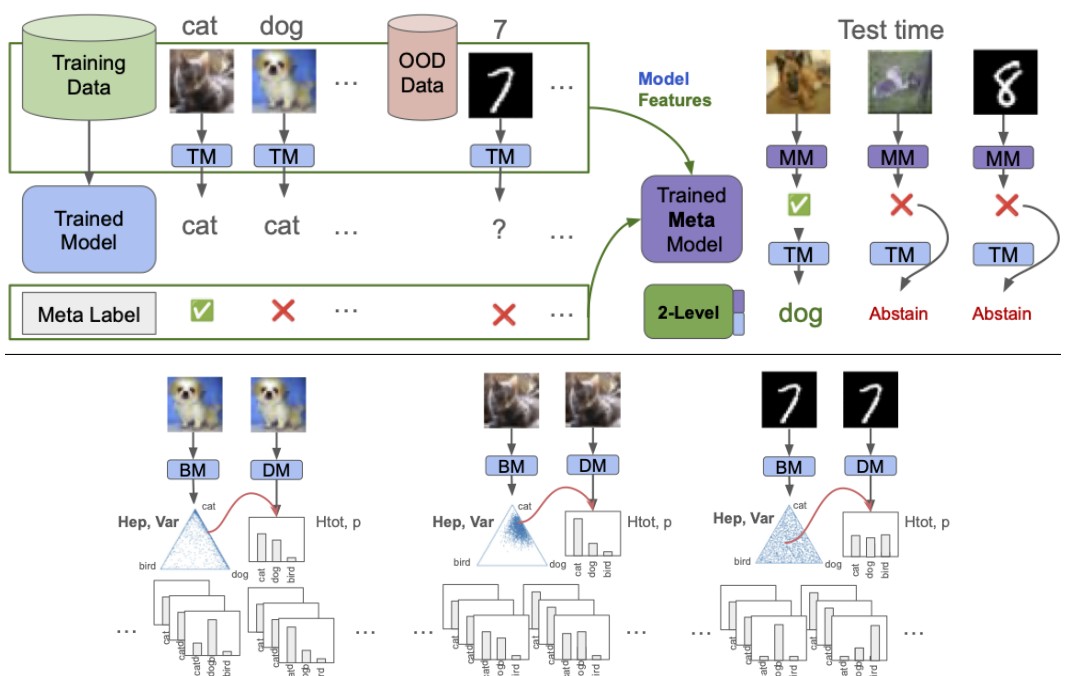

Figure 4: **Top**: graphical schematic for meta-classifier pipeline. Prediction accuracy of the trained model (TM, blue) is used to label a separate calibration dataset of in-domain and out-of-domain (OOD) data. Predictive features are built from the trained model and used to train a binary meta-classifier (MM, purple), which can then be used to measure model confidence and abstain from responding to inputs with low model confidence, for example those which are noisy, ambiguous, or OOD. **Bottom**: the richer Bayesian models (BM) estimate *a distribution over probabilities* on the simplex, whereas deterministic models (DM) *only deliver a point estimator* $\hat{p}(x)$. The model-dependent input features for the meta-classifier are built from both first order frequentist metrics that depend only on $\hat{p}(x)$, such as $H_{\text{tot}}(x)$, and also ensemble-based metrics that depend on the epistemic uncertainty delivered by the Bayesian solution, such as variances of frequentist metrics over the simplex and $H_{\text{ep}}(x)$. (3-class marginals of the 10-simplex are presented for clarity).

The starting point is the pre-trained GPT-2 model fine-tuned on Shakespeare data [2]. We then adopt a LoRA approach (Hu et al., 2022) to fine-tune an additive rank 50 adjustment with $\approx 2e5$ parameters at the last layer on tiny Shakespeare data [3][4]. Top-1 token-level predictions for an ensemble of 10 HMC runs are presented in Table 2.

Table 2: Test accuracy, NLL, and various entropy metrics for next-token prediction with GPT2 on tiny Shakespeare.

| Methods | Accuracy (%) | NLL | $H_{\text{tot}}$ correct | $H_{\text{tot}}$ incorrect | $H_{\text{ep}}$ correct | $H_{\text{ep}}$ incorrect |
|---|---|---|---|---|---|---|
| MAP | 38.66 | 3.166 | 1.554 | 3.605 | 0 | 0 |
| S-HMC | 39.36 | 3.083 | 1.571 | 3.612 | 0.047 | 0.077 |

### 3.1 CONFIDENCE META-CLASSIFIER

One clear application of UQ is inferring confidence in model predictions, i.e. whether the output is reliable or "hallucinated", to borrow the vernacular from modern LLMs (Ji et al., 2023; Guo et al., 2025). This information can be used to decide whether the model should abstain from responding or provided to the user so they can make their own decision about whether to trust the response. To that end, we propose to build a *confidence meta-classifier* of incorrect/OOD data, as follows.

---

[2] https://huggingface.co/sadia72/gpt2-shakespeare
[3] https://raw.githubusercontent.com/karpathy/char-rnn/master/data/tinyshakespeare/input.txt
[4] We truncated to the 2500 most frequent tokens, which includes tokens that appeared 11 or more times.

First, we fit our model to 1000 training data and 200 validation data for early stop procedure in MAP and DE methods/prior from MNIST7, and label incorrect predictions as $z = 1$ and correct predictions as $z = 0$. Then, we generate 2000 additional OOD meta-training data as described above, all of which get label $z = 1$. Let $p_{\max}(x, \theta) := \max_y p(y|x, \theta) = p(y^*|x, \theta)$ denote the maximum probability, and denote the difference between the top two as $\Delta_{\max}(x, \theta) := p_{\max}(x, \theta) - \max_{y' \in \mathsf{Y} \setminus y^*} p(y|x, \theta)$. Let

Table 3: Confidence meta-classifier results for MNIST7 on 2500 ID and 2500 OOD test data, including 500 *unseen far-OOD CIFAR examples*.

| $P$ | Method | Precision | Recall | F1 | AUC-ROC |
|---|---|---|---|---|---|
| – | MAP | 0.771 | 0.891 | 0.826 | 0.864 |
| – | DE | 0.823 | **0.904** | 0.861 | 0.926 |
| 1 | S-SMC$_\parallel$ | 0.794 | 0.886 | 0.837 | 0.895 |
| 8 | S-SMC$_\parallel$ | **0.848** | 0.903 | **0.875** | **0.940** |
| 1 | S-HMC$_\parallel$ | **0.837** | 0.898 | 0.867 | 0.932 |
| 8 | S-HMC$_\parallel$ | 0.844 | 0.909 | 0.875 | 0.942 |

$p_{\max}(x \mid \mathcal{D}) = \max_y \mathbb{E}[p(y|x, \theta) \mid \mathcal{D}]$. Consider as features: $p_{\max}(x \mid \mathcal{D})$, $H_{\text{total}}(x)$, $\mathbb{E}[p_{\max}(x, \theta)|\mathcal{D}]$, $\mathbb{E}[\Delta_{\max}(x, \theta)|\mathcal{D}]$, $H_{\text{ep}}(x)$, $\mathsf{Var}[p_{\max}(x, \theta)|\mathcal{D}]$, $\mathsf{Var}[\Delta_{\max}(x, \theta)|\mathcal{D}]$. Note that the last 3 are identically 0 for the MAP estimator, as they capture the epistemic uncertainty in the data. We build and standardize these features for each of our 4 models–MAP, DE, S-SMC$_\parallel$, S-HMC$_\parallel$– and train the binary meta-classifier $x \mapsto z$, using a single hidden layer MLP with 50 neurons.

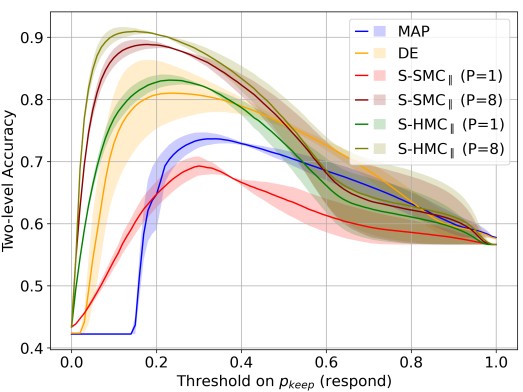

Figure 5: 2-level estimator (using confidence meta-classifier for abstention) accuracy on IMDb.

The results on 2500 ID unseen test data plus 2500 newly-generated OOD test data are presented in Table 3. The OOD data is comprised of 500 examples from each of the four classes used for meta-training, as well as 500 unseen CIFAR images, *for a test of robustness* (see Appendix G.1.1 for category-specific results, and Table 13 for CIFAR specifically). Accuracy is the ratio of true positives (not correct) and true negatives (correct) to the total testing dataset size. All estimators do surprisingly well, and our SBMC methods are the best. AUC-ROC is perhaps the most useful metric, as it measures the ability of the score $p_{\text{incorrect}}(x)$ to rank in-correctness of the subsequent inference: the probability that a randomly selected incorrect example from the test set, $x_{\text{incorrect}} \in \mathcal{D}_{\text{incorrect}}$ will have a higher score than a randomly selected correct example, $x_{\text{correct}} \in \mathcal{D}_{\text{correct}}$: $\mathbb{P}[p_{\text{incorrect}}(x_{\text{incorrect}}) > p_{\text{incorrect}}(x_{\text{correct}})]$.

Figure 5 shows the accuracy of a 2-level estimator over thresholds for IMDb. See Appendix G for further details. First we infer with the meta-classifier whether the model inference will be correct, under the assumption that OOD inputs implies incorrect inference. If yes, we infer with the original model. If no, we *abstain* from inference (abstentions are counted as correct decisions when the original model would be incorrect). Figure 3 presents summary metrics for IMDb and CIFAR10.

Table 4: Meta-classifier ablations. Area under the 2-level estimator curve (as in Figure 5) for MNIST7.

| $P$ | Method | All features | Epistemic features | $p_{\max}$ | $H_{\text{ep}}$ |
|---|---|---|---|---|---|
| - | MAP | 0.717 | 0.501 | 0.701 | 0.501 |
| - | DE | 0.795 | 0.790 | 0.734 | 0.789 |
| 1 | S-SMC$_\parallel$ | 0.747 | 0.738 | 0.690 | 0.738 |
| 8 | S-SMC$_\parallel$ | **0.820** | **0.820** | 0.718 | **0.818** |
| 1 | S-HMC$_\parallel$ | 0.804 | 0.801 | 0.712 | 0.798 |
| 8 | S-HMC$_\parallel$ | **0.822** | **0.822** | 0.716 | **0.819** |

### 3.2 Ablations and Theory

**Ablations** for SBMC are considered by varying $s$ and $P$. Small $s$ improves mixing, as shown in Figure 1 (d), but also introduces bias because $\bar{\pi} \neq \pi$ (see equation 3, equation 1). In this short-chain setting, smaller $s$ typically increases accuracy and decreases $H_{\text{ep}}(\text{OOD})$. Both algorithms improve with $P$, but SMC$_\parallel$ more so. Comprehensive results are given in Appendix H. We also

consider ablations for the meta-classifier, in order to elucidate the role of the epistemic features on the performance. Let us summarize the results of Figures like 5 as the area(s) under the 2-level estimator curve (AU2LC). A score of roughly $1/2$ corresponds to a $0-$skill estimator and a score of 1 would be perfect. Table 4 shows the AU2LC for the case of all features, only the epistemic features, the classical $p_{\mathsf{max}}$ confidence score, and $H_{\mathsf{ep}}$. Observe that we perform much better than $p_{\mathsf{max}}$, $H_{\mathsf{ep}}$ almost sufficiently encapsulates the epistemic features, and the latter are almost suitable in comparison to the estimator built from all features.

**Tuning.** Firstly, we would like to empha-size that the results are fairly insensitive to $s \in [0.05, 0.3]$, and we recommend se-lecting $s = 0.1$ as a good default choice. It is worth noting that $v$ is an important hyper-parameter *a priori*, at the level of the original Bayesian model. [5] If desired, one should first select $v$ optimally for the MAP/DE, and then select $s$. Both can be done with CV. See Section 4 for further discussion.

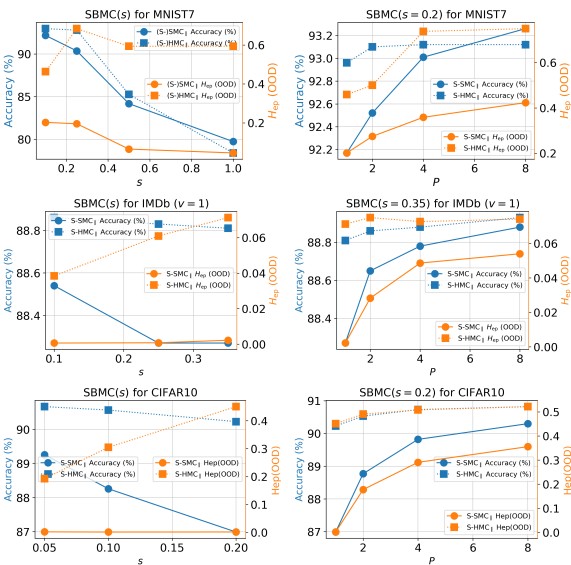

**Theory sketch.** We can understand the SBMC model as an incremental incorpo-ration of the data. Let $\sigma^2 = v/a$ and consider the original problem with prior $\mathcal{N}(0, \sigma^2)$. Now *split* the log-likelihood into $(1-a)\ell + a\ell$, and consider incorporating *only the $a\ell$ part*. It is easy to see that the MAP estimator $\hat{\theta}_{\mathsf{MAP}}$ for this problem is equivalent to the MAP estimator associated with the prior $\mathcal{N}(0, v)$. The *Laplace ap-proximation* however, will differ, depend-ing on which one we consider. Let us con-

Figure 6: Dual axis Accuracy and $H_{\mathsf{ep}}$(OOD) ablations over $s$ in column 1 and $P$ in column 2 for MNIST7 (row 1), IMDb (row 2), and CIFAR10 (row 3).

sider the $\mathcal{N}(0, \sigma^2)$ prior, and now it is time to incorporate the rest of the data $(1-a)\ell$. The Hessian of our Laplace approximation is

$$a\nabla^2\ell(\hat{\theta}_{\mathsf{MAP}}) + \frac{1}{2(v/a)}\mathsf{Id}.$$

This could be carried through rigorously, but let's swap out $N_{\mathrm{train}}\mathsf{Id}$ for $\nabla^2\ell(\hat{\theta}_{\mathsf{MAP}}) = \sum_{i=1}^{N_{\mathrm{train}}} \nabla^2\ell_i(\hat{\theta}_{\mathsf{MAP}})$. To arrive at the SBMC prior we must equate the following

$$(2vN_{\mathrm{train}} + 1)/(2v/a) = 1/(2sv) \quad \Rightarrow \quad a = s^{-1}/(2vN_{\mathrm{train}} + 1).$$

Suppose $v = 0.1$ (common) and $s = 0.1$ (our recommendation). Then $a = 10/(N_{\mathrm{train}}/5 + 1) \ll 1$, for large datasets. Therefore, $\ell \approx (1-a)\ell$, and we arrive at a reasonable approximation. Note this is also typically a positive thing for the prior, which is effectively $\mathcal{N}(0, \sigma^2 = v/a)$, since broader priors and less inductive bias typically deliver better performance, and small variance priors are often chosen more as a matter of convenience.

**Hessian.** Note the Hessian of the posterior $\pi$ in equation 1 with $\mathcal{N}(0, v)$ prior is $\nabla^2\ell(\theta) + \frac{1}{v}\mathsf{Id}$. If we assume that the minimum eigenvalue of $\nabla^2\ell(\theta)$ is 0, and the maximum is $\lambda_{\mathsf{max}}$, then the *condition number* of the Hessian of the posterior is $\lambda_{\mathsf{max}}v + 1$. Meanwhile, the Hessian of the SBMC target $\bar{\pi}$ in equation 3 with $\mathcal{N}(\theta_{\mathsf{MAP}}, vs)$ prior will have condition number $\lambda_{\mathsf{max}}vs + 1$. Therefore, the parameter $s < 1$ allows us to "tune away" the ill-conditioning of the posterior.

## 4 DISCUSSION OF RELATED WORK

There has been a growing amount of work recently in many-short-chain MCMC, e.g. (Vehtari et al., 2000; Wilkinson, 2006; Chen et al., 2016; Sommer et al., 2024; Margossian et al., 2024; Nguyen

---

[5]Prior tuning is relevant for *all methods*, Bayesian and Frequentist (weight decay), and is not particular to SBMC. A good rule of thumb is *enough but not too much, i.e. $v$ as large as possible* Izmailov et al. (2021).

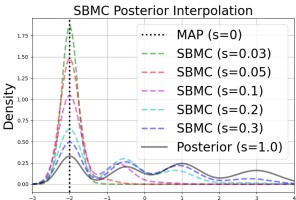 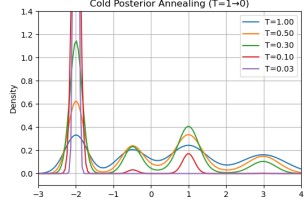 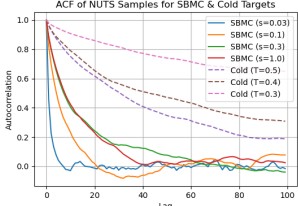

Figure 7: Left: SBMC for various $s$. Middle: Cold posterior for various $T$. Right: Autocorrelation functions using NUTS sampler, showing that SBMC *improves* mixing, while CP *hinders* mixing.

et al., 2025; Sommer et al., 2025; Duffield et al., 2024). The island-SMC method Vergé et al. (2015) considers interacting SMCs, which is necessary for consistency *unless the estimator is carefully constructed with appropriate weights* Whiteley et al. (2016); Dai et al. (2022). See e.g. Liang et al. (2025) for further discussion.

The idea of *MAP-anchored priors* is intuitive, and closely related to a number of successful methods. In addition to augmenting data with adversarial perturbations of the inputs, as proposed in the original DE paper, another intuitive idea to promote spread and generalization is to randomize the model itself. Randomized maximum likelihood (RML) approaches do this by anchoring each ensemble member to random draws from the prior and/or data (Gu & Oliver, 2007; Bardsley et al., 2014; Pearce et al., 2020). SBMC can easily bootstrap DE or RML ideas by initializing each process from a different MAP estimator. It is worth noting that MC Dropout Gal & Ghahramani (2016) could also do this.

The method most closely related to our work is Paulin et al. (2025), who anchor to the SWA estimator by adding a Gaussian factor, and simulate an ensemble of ULAs. They also observed an extreme speedup in mixing time. We experimented with a similar formulation with a factor of $\mathcal{N}(\theta_{\mathsf{MAP}}, s\mathsf{Id})$, $s \in (0, \infty)$, which also interpolates between the posterior and the MAP estimator and is arguably more elegant and theoretically appealing. But, the effective prior centers on $\frac{v}{s+v}\theta_{\mathsf{MAP}}$ (or SWA), and in practice we found that this version did not perform as well as centering the prior on $\theta_{\mathsf{MAP}}$ itself. The former is shown in Table 1 as S-HMC$_{\parallel}$(lin).

**Cold posteriors** (Wenzel et al., 2020) also interpolate between $\delta_{\theta_{\mathsf{MAP}}}$ and the posterior via *annealing* (or 'tempering') the posterior equation 1 with an inverse temperature $T < 1$, as $\tilde{\pi}_T(\theta) \propto \mathcal{L}(\theta; \mathcal{D})^{1/T} \pi_0^{1/T}$. This *sharpens* the posterior, as shown in Figure 7 (middle) which makes the target distribution *more difficult to simulate* and *slows down* MCMC mixing, as shown in Figure 7 (right).

The SBMC likelihood is *effectively flattened relative to the Gaussian prior* by the factor $s$, while the missing information is represented in the sharper prior, as shown in Figure 7 (left). In practice, this means that the nonlinear and irregular component of the gradients has a smaller relative magnitude, the total Hessian of the posterior is better conditioned, and the chains mix faster.

## 5 CONCLUSION

The SBMC method has been introduced and shown to be within reach of modern practical applications. It comprises a judicious **model** which uses a scalar parameter $s$ to interpolate between $\delta_{\theta_{\mathsf{MAP}}}$ ($s = 0$) and the posterior ($s = 1$), and is hence able to balance the benefits of each and achieve strong performance in accuracy and UQ metrics *at a cost comparable to SOTA approaches like DE*. Both MCMC$_{\parallel}$ and SMC$_{\parallel}$ are attractive **algorithm** options, which are consistent for the given target model[6] Therefore we have a mechanism for controlling the approximation between two reasonable choices if convergence is ensured. However, since the method no longer targets the posterior for any $s < 1$, we would recommend adopting a heuristic approach to convergence as with other SOTA methods, rather than chasing more rigorous convergence guarantees. Any BMC algorithm can be used, and SG-MCMC methods are particularly attractive since they are amenable to mini-batching and close to SGD, for which ample deep learning tooling is readily available.

---

[6]SMC$_{\parallel}$ has stronger theoretical guarantees but MCMC$_{\parallel}$ is easier to implement and is amenable to SG-MCMC approaches.

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

## A  FUTURE DIRECTIONS

The most obvious next step is UQ for modern large language models (LLMs) (Guo et al., 2025), where robustness and hallucination detection are crucial pain points (Vashurin et al., 2025). It has been recently shown that high-quality entropy metrics are valuable for identifying untrustworthy outputs there (Gustafsson et al., 2020; Arteaga et al., 2024; Farquhar et al., 2024). In the context of LLMs, where the training itself is extremely computationally expensive, it becomes particularly important to have add-on plug-in type methods that can be applied post-training, such as (Qiu et al., 2019; Farquhar et al., 2024; Qiu & Miikkulainen, 2024). But those methods are constrained to the uncertainty already encoded in the point estimator of model weights, which may already be under-estimated. The work (Arteaga et al., 2024) has shown that batch ensembles of fine-tuned LLMs can also work well for UQ hallucination detection, and also that epistemic uncertainty provides valuable information for that task. Based on existing benchmarks against deep ensembles, we believe our SBMC($s$) approach will perform even better. Furthermore, an even simpler and cheaper version is to *learn the last layer* only, so all the data can be pre-processed once and for all by the frozen pre-trained LLM parameters, and then we simply run the last layer through SBMC. This can be applied at the pre-training or post-training stage, although the value of the method on downstream tasks will be most clear at post-training, while there would be some necessary design choices for how to leverage a pre-trained ensemble, instead of a point estimator, during post-training.

### A.1  OVERCOMING OTHER COMPUTATIONAL BOTTLENECKS

Our sampler relies only on forward/back-prop evaluations, so every mainstream hardware scheme can be stacked on top of it: data-parallel all-reduce for moderate models(Goyal et al., 2017); optimizer-state sharding (ZeRO/FSDP) when parameters no longer fit(Rajbhandari et al., 2020); tensor model-parallelism for in-layer splits (Shoeybi et al., 2019) and pipeline model-parallelism for depthwise splits (Huang et al., 2019); and, finally, the full hybrid of DP/sharding/tensor/pipeline that is now routine in trillion-parameter language models (Chowdhery et al., 2023; Black et al., 2022).

### A.2  FURTHER DIRECTIONS

Further directions at the methodological level include

- DE-SBMC: an obvious extension, would be to condition the HMC ensemble, or SMC ensemble ($P > 1$) with the DE, in case there may be any gain to be had.
- One could condition SGLD/SGHMC with the MAP(s) from SGD (or DE). In this way, there is an initial phase which aims to recover a good point estimator, and then a second phase of essentially the same method, which aims to quantify the spread.
- $P-$parallelizing $N-$ensemble MCMC methods such as Gilks et al. (1994); Goodman & Weare (2010); Vrugt et al. (2009); Hoffman & Sountsov (2022).
- Leveraging $N-$ensemble MCMC methods within SMC for better mutations (with the cost of more communication).
- Parallel stochastic-gradient-MCMC methods like SGLD (Welling & Teh, 2011) and SG-HMC (Chen et al., 2014), and ensemblized versions thereof.

- Related to above, mini-batch gradients can be used in lieu of full gradients, which may have some advantages in terms of scalability and convergence. For SMC samplers, we have unbiased estimators $\widehat{\ell w}$ of log weights using mini-batches, and could use $\exp(\widehat{\ell w})$ for a non-negative and biased estimator or Bernoulli/Poisson augmentation to achieve (non-negative) unbiased weights (Gunawan et al., 2020; Bardenet et al., 2017).

- Evidential models explicitly model uncertainty parametrically with point estimators (Charpentier et al., 2020), but it has been observed that deep ensembles of such estimators perform better Charpentier et al. (2022). It would be natural to build Bayesian evidential models and simulate them with SBMC.

## B  More related work

*Empirical Bayes* methods (Efron & Morris, 1973) fit higher level parameters in hierarchical models through optimization of the marginal likelihood. An SBMC model could be built in principle with a general prior $\pi_\phi(\theta)$, for example $\mathcal{N}(\theta; \mu, \Sigma)$ for $\phi = (\mu, \Sigma)$, and solved by EB. There is a significant cost overhead for optimizing the marginal-likelihood, but that could be offset in principle with Laplace approximation Bishop (2006) or other approaches. The particular SBMC($s$) model considered here could also utilize EB for selecting $s$ and/or $\alpha$, as an alternative to cross-validation. And even before this, EB could be used to define the prior variance $v$, or a more general prior.

The work LogME You et al. (2021) use EB for fitting prior and likelihood variance in the context of transfer learning for regression, and then they extended this idea for building estimators from an ensemble of pre-trained models You et al. (2022). The latter could naturally be combined with other ensemble approaches described above, and plugged into SBMC. The *Laplace Redux* work of Daxberger et al. (2021) provides an off-the-shelf Laplace module with block diagonal Hessian approximations to plug pre-trained models into for transfer learning. This could naturally augment SBMC in a number of ways, from leveraging it in marginal likelihood calculations for EB, to using it as a drop in alternative for the prior, or leveraging their Hessian approximation in various other ways. It is worth momentarily digressing on an approach inspired by this observation. If the data is split into $N_\alpha$ pre-training and $N - N_\alpha$ fine-tuning sets, or if the likelihood is split by $\alpha$ and $1 - \alpha$ scalar fractions, then one could build a Laplace approximation (or another variational approximation) of the original $\alpha$ posterior, and use that as a prior for the remaining likelihood fraction. This posterior approximation may be closer to the original GS, and since each problem features an explicitly tempered likelihood, they should both be easier solve. This may help with potentially overfitting, although we did not observe much of an issue in that respect.

The recent work Li et al. (2025) points out that supervised OOD meta-classification involving OOD calibration data suffers from an inevitable problem of meta-OOD data, and calls for the development of fundamentally new approaches to OOD detection for this reason. This concern could be mitigated with the Bayesian approach, since Bayesian methods directly deliver a reasonable abstention score in $H_{\text{ep}}$ itself by design, as shown in Figure 2 and Table 4. In other words, we could either bypass the meta-classifier altogether, or we build a Bayesian meta-classifier and then stop there. The latter may be preferable, since the meta-classifier delivers improvement and is cheap to learn. The work Izmailov et al. (2021) shows that BDL is more effective for the harder problem of identifying near-OOD data and not as good as custom-designed OOD models for the easier problem of identifying far-OOD data, so a 2-level approach may work best where we first filter out far OOD data using an approach like Lee et al. (2018) and subsequently evaluate prediction confidence.

## C  Techniques for $\text{SMC}_{\|}$ in practice

**Adaptive tempering.**  As mentioned, adaptive tempering is used to ensure a dense tempering regime and provide stability(Syed et al., 2024).

**Example C.1** (Adaptive tempering).  In order to keep the sufficient diversity of sample population, we let the effective sample size to be at least $\text{ESS}_{\min} = N/2$ at each tempering $\lambda_{j-1}$ and use it compute the next tempering $\lambda_j$. For $j$th tempering, we have weight samples $\{w_{j-1}^k, \theta_{j-1}^k\}_{k=1}^N$, then

the ESS is computed by

$$\text{ESS} = \frac{1}{\sum_{k=1}^{N}(w_{j-1}^{k})^2},$$

where $w_{j-1}^{k} = \mathcal{L}(\theta_{j-1}^{k})^{\lambda_j - \lambda_{j-1}} / \sum_{k=1}^{N} \mathcal{L}(\theta_{j-1}^{k})^{\lambda_j - \lambda_{j-1}}$. Let $h = \lambda_j - \lambda_{j-1}$, the effective sample size can be presented as a function of $h$, $\text{ESS}(h)$. Using suitable root finding method, one can find $h^*$ such that $\text{ESS}(h^*) = \text{ESS}_{\min}$, then set the next tempering $\lambda_j = \lambda_{j-1} + h^*$.

Note that the partition function estimator $Z^N$ is no longer unbiased once we introduce adaptation, which means that in principle we should do short pilot runs and then keep everything fixed to preserve the integrity of the theory, but we have found this does not make a difference in practice.

**Adaptive number of mutation steps.** The number of mutation steps $M$ is chosen adaptively. After resampling at a given tempering step, let $\theta^{i,0}$ denote the $i$-th sample and $\theta^{i,m}$ its state after $m$ mutation steps. We monitor the mean displacement from the post-resampling state,

$$\text{dist}_m = \frac{1}{N} \sum_{i=1}^{N} \left\| \theta^{i,m} - \theta^{i,0} \right\|_2,$$

and terminate the mutation update at the smallest $M \geq 2$ for which the displacement has stabilized:

$$\frac{|\text{dist}_M - \text{dist}_{M-1}|}{\text{dist}_{M-1}} \leq \eta,$$

with tolerance $\eta > 0$. This criterion automatically increases $M$ when the tempering increment is large or the target becomes tighter (requiring more mixing to decorrelate the resampled particles), and conversely saves computation when the resampled state is already close to stationary at the new tempering level.

**Numerical stability: nested Log-sum-exp.** When computing likelihoods in Sequential Monte Carlo (SMC) algorithms, numerical underflow frequently arises because likelihood values can become extremely small, often beyond computational precision. To address this, one standard practice is to work with log-likelihoods rather than likelihoods directly. By operating in the log domain, the computer can safely store and manipulate extremely small values without loss of precision.

Specifically, the standard *log-sum-exp* trick can be applied to stabilize computations. For instance, consider a scenario with nested sums and products in parallel SMC. For each processor $p = 1, \ldots, P$, we initially have:

$$Z^{N,p} = \prod_{j=1}^{J} \sum_{i=1}^{N} \omega_j^{i,p}.$$

To avoid numerical instability, each sum within the product is computed using the log-sum-exp trick:

$$\sum_{i=1}^{N} \omega_j^{i,p} = \exp\left( \max_i \log(w_j^{i,p}) \right) \sum_{i=1}^{N} \exp\left( \log(w_j^{i,p}) - \max_i \log(w_j^{i,p}) \right).$$

This procedure yields the decomposition:

$$Z^{N,p} = K^p \hat{Z}^p,$$

where

$$K^p = \prod_{j=1}^{J} \exp\left( \max_i \log(w_j^{i,p}) \right), \quad \text{and} \quad \hat{Z}^p = \prod_{j=1}^{J} \sum_{i=1}^{N} \exp\left( \log(w_j^{i,p}) - \max_i \log(w_j^{i,p}) \right).$$

In parallel SMC, an additional stabilization step is applied across processors. The global normalization constant across processors can also suffer from numerical instability. To address this, the log-sum-exp trick is applied again at the processor level:

$$Z^{N,p} = \exp\left( \log(\hat{Z}^p) + \log(K^p) - \log(K) \right) K,$$

with

$$\log(K) = \max_p \left( \log(\hat{Z}^p) + \log(K^p) \right).$$

Since the factor $K$ cancels out when calculating the parallel SMC estimator, it suffices to compute only:

$$\exp \left( \log(\hat{Z}^p) + \log(K^p) - \log(K) \right),$$

which ensures numerical stability even when $K$ itself is computationally very small.

Thus, by recursively applying the log-sum-exp trick at both the particle and processor levels, parallel SMC estimators can robustly handle computations involving extremely small numbers without numerical underflow.

## D  COMPLEMENTARY DESCRIPTION OF SIMULATIONS

### D.1  COMPUTATION OF ERROR BARS

Assume running $R$ times of experiments to get $R$ square errors/loss between simulated estimator $\hat{\varphi}$ and the ground truth, $\text{SE}(\hat{\varphi})^r$ for $r = 1, ..., R$. Take the MSE as an example, the MSE is the mean of $\text{SE}(\hat{\varphi})^r$ over $R$ realizations, and the standard error of MSE (s.e.) is computed by

$$\frac{\sqrt{\frac{1}{R} \sum_{r=1}^{R} (\text{SE}(\hat{\varphi})^r - \text{MSE})^2}}{\sqrt{R}}. \tag{7}$$

### D.2  INTEGRATED AUTOCORRELATION TIME

Integrated Autocorrelation Time (IACT) means the time until the chain is uncorrelated with its initial condition. The precise mathematical definition is as follows.

Let $\theta_0, \ldots, \theta_t, \ldots$ denote the Markov chain, and let $\varphi(\theta)$ be a scalar function of the state. We first define the *autocovariance function* (ACF) at lag $s$:

$$\gamma_s(\varphi) = \mathbb{E} \left[ \left( \varphi(\theta_{t+s}) - \mathbb{E}[\varphi(\theta)] \right) \left( \varphi(\theta_t) - \mathbb{E}[\varphi(\theta)] \right) \right],$$

and the ACF at lag $s$ as the normalized quantity

$$\rho_s(\varphi) = \frac{\gamma_s(\varphi)}{\gamma_0(\varphi)},$$

where $\gamma_0(\varphi)$ is the variance of $\varphi(\theta)$.

Then the *integrated autocorrelation time* (IACT) of $\varphi$ is then defined in terms of the ACF by

$$\text{IACT}(\varphi) = 1 + 2 \sum_{s=1}^{\infty} \rho_s(\varphi).$$

### D.3  DETAILS OF THE BAYESIAN NEURAL NETWORKS

Let weights be $A_i \in \mathbb{R}^{n_i \times n_{i-1}}$ and biases be $b_i \in \mathbb{R}^{n_i}$ for $i \in \{1, ..., D\}$, we denote $\theta := ((A_1, b_1), ..., (A_D, b_D))$. The layer is defined by

$$g_1(x, \theta) := A_1 x + b_1,$$
$$g_d(x, \theta) := A_i \sigma_{n_{i-1}}(g_{i-1}(x)) + b_i, \quad i \in \{2, ..., D-1\},$$
$$g(x, \theta) := A_D \sigma_{n_{D-1}}(g_{D-1}(x)) + b_D,$$

where $\sigma_i(u) := (\nu(u_1), ..., \nu(u_i))^T$ with ReLU activation $\nu(u) = \max\{0, u\}$.

Consider the discrete data set in a classification problem, we have $\mathbb{Y} = \{1, ..., K\}$ and $n_D = K$, then we instead define the so-called *softmax* function as

$$h_k(x, \theta) = \frac{\exp(g_k(x, \theta))}{\sum_{j=1}^{K} \exp(g_j(x, \theta))}, \quad k \in \mathbb{Y}, \tag{8}$$

and define $h(x, \theta) = (h_1(x, \theta), ..., h_K(x, \theta))$ as a categorical distribution on $K$ outcomes based on data $x$. Then we assume that $y_i \sim h(x_i)$ for $i = \{1, ..., m\}$.

Now we describe the various neural network architectures we use for the various datasets.

### D.3.1 MNIST7 CLASSIFICATION EXAMPLE

The architecture is a simple CNN with (i) one hidden layer with $4$ channels of $3 \times 3$ kernels with unit stride and padding, followed by (ii) ReLU activation and (iii) $2 \times 2$ max pooling, (iv) a linear layer, and (v) a softmax. The parameter prior and dataset is built as follows

- Training is conducted on a sub-dataset consisting of the first 1200 training samples with labels 0 through 7. Evaluation is performed on first $N_{\text{id}}$ in-domain test images with labels 0 through 7 and the on the four generated out-of-domain dataset ($N_{\text{ood}}$ total number of data).

- The OOD dataset is generated as follows: two of the datasets are the first $N_{\text{ood}}/4$ out-of-domain test images with labels 8 and 9, respectively. The third dataset, the white noise image (wn), is a set of $N_{\text{ood}}/4$ synthetic $28 \times 28$ "images" with pixels drawn uniformly at random from $[0, 1]$. The fourth dataset, the perturbed image (per.), is a set of the first $N_{\text{ood}}/4$ MNIST test images of digits0–7, each pixel perturbed by Gaussian noise (standard error as $0.5$) while retaining its original label.

- MAP and DE are estimated using an initialization and regularization based on the prior $N(0, v\mathsf{Id})$, where $d = 6320$ and $v = 0.1$. The tuning parameter in SBMC methods is $s$. The batchsize is $64$.

- The gold-standard is computed by the single HMC over 5 realizations, called HMC (GS), with $N = B, T = 1$ and $L = 1$.

- SWA. Starting from the estimated MAP weights, we train with SGD (momentum $0.9$, lr $10^{-3}$). After a 25-epoch warm-up, we update an `AveragedModel` each epoch (1 weight sample per epoch) and use `SWALR` with `swa_lr` $= 5 \times 10^{-4}$. After training we use the SWA weights for prediction. We run $R{=}5$ independent replicates.

- MC Dropout. We enable a 30% dropout in the (only) dropout layer after flattening (before the FC). Starting from the MAP estimation, we train with Adam (lr $10^{-3}$). At test time, we keep dropout on and average $T{=}10$ stochastic forward passes for predictive probabilities. We run $R{=}5$ independent replicates.

- Laplace. We fit a Laplace approximation with a Kronecker-factored approximation of the Hessian[7]Daxberger et al. (2021). The starting model is the estimated MAP model; we estimate the posterior over the last layer and draw $T{=}10$ predictive weight samples with `pred_type='nn'` (re-linearization) for MC predictive inference. We run $R{=}5$ independent replicates.

### D.3.2 IMDb CLASSIFICATION EXAMPLE

Here we use SBERT embeddings Reimers & Gurevych (2019) based on the model all-mpnet-base-v2 Song et al. (2020) [8]. In other words, frozen weights from all-mpnet-base-v2 until the 768 dimensional [CLS] output. The NN model and parameter prior for IMDb[9] are built as follows

- NN is followed by (i) no hidden layer, (ii) ReLU activation, (iii) a final linear layer, and (iv) softmax output.

- Training is conducted on the whole train set (25000 data). Evaluation is performed on the whole test images as the in-domain dataset (25000 data) and on the four generated out-of-domain datasets ($N_{\text{ood}}$ total number of data).

- The OOD dataset is generated as follows: four of these datasets (each dataset has $N_{\text{ood}}/5$ data) use textual data from the Appliances domain, which is distinct from the in-domain IMDb movie review data. Specifically, four OOD datasets were constructed from Amazon

---

[7]https://github.com/aleximmer/Laplace

[8]https://huggingface.co/sentence-transformers/all-mpnet-base-v2

[9]https://huggingface.co/datasets/stanfordnlp/imdb

Reviews 2023 Appliances data Hou et al. (2024) [10], containing customer reviews and product metadata. Two datasets directly used the two JSON files, and two text-based OOD datasets were generated as follows. From `Appliances.jsonl`, we extracted the review text, representing natural language expressions of user opinions but unrelated to movies; from `meta_Appliances.jsonl`, we constructed meta descriptions by concatenating each product's title and listed features. The last dataset, Lipsum, is a collection of 100 very short, meaningless text strings, each consisting of between one and ten randomly selected words drawn from the classic "Lorem ipsum" filler vocabulary.

- MAP and DE are estimated using an initialization and regularization based on the prior $N(0, v\mathsf{I}d)$, where $d = 1538$. The tuning parameter in SBMC methods is $s$. The batchsize is 64.

- The gold-standard is computed by the single HMC over 5 realizations, called HMC (GS), with $N = B, T = 1$ and $L = 1$.

### D.3.3 CIFAR-10 CLASSIFICATION EXAMPLE

Here, the architecture is ResNet-50 pre-trained from ImageNet with all parameters frozen until the final pooled 2048 dimensional features. The NN model and parameter prior for CIFAR10 are as follows.

- NN is followed by (i) no hidden layer, (ii) ReLU activations, (iii) a final linear layer, and (iv) softmax output.

- Training is conducted on the whole train set (50000 data). Evaluation is performed on the whole test images as the in-domain dataset (10000 data) and on the three generated out-of-domain datasets ($N_{\mathsf{ood}}$ total number of data).

- The OOD dataset is generated as follows:

  - Close OOD (CIFAR-100 "not in CIFAR-10"). Drawn $N_{\mathsf{ood}}/3$ data from the 90 fine-grained CIFAR-100 classes that don't overlap with the 10 classes inCIFAR-10. All images are $32 \times 32$ RGB natural photographs with nearly identical color distribution and textures to CIFAR-10.
  - Corrupt OOD (CIFAR-10-C). Select $N_{\mathsf{ood}}/3$ CIFAR-10 test images and subject them to 15 types of realistic distortions—Gaussian/impulse noise (motion/defocus blur, frost, fog, brightness/contrast shifts, JPEG compression, pixelation, etc.) at five different severity levels. The pixel-level statistics are methodically disturbed, yet the original labels stay the same.
  - Far OOD (SVHN). Select $N_{\mathsf{ood}}/3$ data from 26032 32x32 RGB test photos of house-number digits (0–9) that have been cut from Google Street View. The SVHN displays centred white numbers on colourful, frequently cluttered urban backgrounds, in contrast to CIFAR's multi-object array of natural-scene photos.

- MAP and DE are estimated using an initialization and regularization based on the prior $N(0, v\mathsf{I}d)$, where $d = 20490$ and $v = 0.2$. The tuning parameter in SBMC methods is $s$. The batchsize is 128.

- The gold-standard is computed by the single HMC over 5 realizations, called HMC (GS), with $N = B, T = 1$ and $L = 1$.

### D.4 HARDWARE DESCRIPTION

The main CPU cluster we access has nodes with $2 \times 16$-core Intel Skylake Gold 6130 CPU @ 2.10GHz, 192GB RAM *without communication* in between, so it can only run $N/P = 32$ particles in parallel with one particle per core. There are also unconnected AMD "Genoa" compute nodes, with $2 \times 84$-core AMD EPYC 9634 CPUs and 1.5TB RAM.

---

[10]https://amazon-reviews-2023.github.io/

# E HYPER-PARAMETER TABLES

Table 5: Hyper-parameters for core methods (MAP, DE, S-HMC$_\parallel$ and S-SMC$_\parallel$) on MNIST7, IMDb and CIFAR10. HMC, SMC, and their variants adopt the same hyper-parameters but run without a pre-trained model and directly target the full posterior distribution. Importantly, DEI-HMC is initialized using the Deep Ens rather than a random sample from the prior.

| | | | Experiments | | |
|---|---|---|---|---|---|
| Method | Hyper-parameter | Was tuned | MNIST7 | IMDb | CIFAR-10 |
| MAP | Prior variance $v$ | ✓ | 1e-1 | 2.5e-2 | 2e-1 |
| | Optimizer (Learning rate) | × | Adam (1e-3) | Adam (1e-3) | Adam (1e-3) |
| | Batch size | ✓ | 64 | 64 | 128 |
| | Early stopping (mov. avg., patience)$^\star$ | × | 10, 5 | 10, 5 | - |
| Deep Ens ($P$) | Pre-trained model | ✓ | MAP | MAP | MAP |
| | Prior variance $v$ | ✓ | 1e-1 | 2.5e-2 | 2e-1 |
| | Optimizer (Learning rate) | × | Adam (1e-3) | Adam (1e-3) | Adam (1e-3) |
| | Batch size | ✓ | 64 | 64 | 128 |
| | Early stopping (mov. avg., patience) | × | 10, 5 | 10, 5 | - |
| | Ensemble size | × | $10P$ | $10P$ | $10P$ |
| S-HMC$_\parallel$ ($P$) | Pre-trained model | ✓ | MAP | MAP | MAP |
| | MAP prior variance $v$ | ✓ | 1e-1 | 2.5e-2 | 2e-1 |
| | Sampling prior scale $s$ | ✓ | 1e-1 | 1e-1 | 1e-1 |
| | Sampling prior variance $sv$ | ✓ | 1e-2 | 2.5e-3 | 2e-2 |
| | Step size $\epsilon$ (initial)$^\dagger$ | ✓ | 1e-2 | 1.8e-2 | 2e-3 |
| | Leapfrog steps $L$ | × | 1 | 1 | 1 |
| | Burn-in particles | ✓ | 160 | 25 | 200 |
| | Num. particles per chain | × | 1 | 1 | 1 |
| | Num. of chains | × | $10P$ | $10P$ | $10P$ |
| | Total posterior particles | × | $10P$ | $10P$ | $10P$ |
| S-SMC$_\parallel$ ($P$) | Pre-trained model | ✓ | MAP | MAP | MAP |
| | MAP prior variance $v$ | ✓ | 1e-1 | 2.5e-2 | 2e-1 |
| | Sampling prior scale $s$ | ✓ | 1e-1 | 1e-1 | 1e-1 |
| | Sampling prior variance $sv$ | ✓ | 1e-2 | 2.5e-3 | 2e-2 |
| | Step size $\epsilon$ (initial)$^\dagger$ | ✓ | 3.5e-2 | 2.2e-2 | 7.0e-2 |
| | ESS tempering threshold | ✓ | $N/2$ | $N/2$ | $N/2$ |
| | Leapfrog steps $L$ | × | 1 | 1 | 1 |
| | HMC transitions per tempering step $M$ | ✓ | 10 | 1 | 4 |
| | Num. tempering steps | ✓ | Adaptive | Adaptive | Adaptive |
| | Num. particles $N$ | × | 10 | 10 | 10 |
| | Num. of chains | × | $P$ | $P$ | $P$ |
| | Total posterior particles | × | $10P$ | $10P$ | $10P$ |

$^\star$Mov. avg. is a smoothed validation metric that reduces noise; patience is the number of steps allowed without improvement before early stopping.

$^\dagger$Step size is adapted during sampling based on the acceptance rate.

Table 6: Hyper-parameters for other baseline models on MNIST7. SGHMC adopts the same hyper-parameters but run without a pre-trained model and directly target the full posterior distribution.

| Method | Hyper-parameter | Was tuned | MNIST7 |
|---|---|---|---|
| MC Dropout | Prior variance $v$ | ✓ | 1e-1 |
| | Optimizer (learning rate) | ✗ | Adam (1e-3) |
| | Dropout rate | ✓ | 3e-1 |
| | Training iterations | ✓ | 160 |
| | Num. particles | ✗ | 10 |
| SWA | Prior variance $v$ | ✓ | 1e-1 |
| | Optimizer (learning rate) | ✗ | SGD (1e-3) |
| | SWA start iteration | ✓ | 25 |
| | Total iterations | ✓ | 160 |
| | SWA learning rate | ✗ | 5e-4 |
| | Momentum | ✗ | 9e-1 |
| SWAG | Prior variance $v$ | ✓ | 1e-1 |
| | Optimizer (learning rate) | ✗ | SGD (1e-3) |
| | SWAG burn-in iterations | ✓ | 25 |
| | Snapshot frequency | ✓ | 2 |
| | Max snapshots | ✓ | 20 |
| | Sampling scale | ✗ | 1.0 |
| | Use low-rank covariance | ✗ | True |
| | Num. particles | ✗ | 10 |
| Laplace | Prior variance $v$ | ✓ | 1e-1 |
| | Subset of weights | ✗ | Last layer |
| | Hessian structure | ✓ | Kronecker |
| | Num. particles | ✗ | 10 |
| S-SGHMC$_\parallel$ $(P)$ | Pre-trained model | ✓ | MAP |
| | MAP prior variance $v$ | ✓ | 1e-1 |
| | Sampling prior scale $s$ | ✓ | 1e-1 |
| | Sampling prior variance $sv$ | ✓ | 1e-2 |
| | Optimizer (learning rate) | ✗ | Adam (1e-3) |
| | Step size $\epsilon$ | ✓ | 6e-3 |
| | SGHMC friction | ✓ | 2e-1 |
| | Burn-in particles | ✓ | 160 |
| | Num. particles per chain | ✗ | 1 |
| | Num. of chains | ✗ | $10P$ |
| | Total posterior particles | ✗ | $10P$ |
| SMS-UBU$_\parallel$ $(P)$ | Pre-trained model | ✓ | MAP |
| | prior variance $v$ | ✓ | 1e-1 |
| | Localization strength | ✓ | 1e-2 |
| | Step size | ✓ | 2.5e-5 |
| | Momentum friction | ✓ | $\sqrt{50}$ |
| | Number of forward sweeps | ✓ | 80 |
| | Number of backward sweeps | ✓ | 80 |
| | Burn-in sweeps | ✓ | 160 |
| | Num. particles per chain | ✗ | 1 |
| | Num. of chains | ✗ | $10P$ |
| | Total posterior particles | ✗ | $10P$ |

# F   FURTHER RESULTS FOR UQ

Results in this section further support the statement mentioned in the main text, that is, (i) SBMC significantly outperforms the MAP estimator, as well as a DE of MAP estimators, (ii) DE systematically underestimates $H_{\text{ep}}$ for the same ensemble size as SBMC.

### F.1 MNIST7

In the MNIST7 case, the full setting is described in Appendix D.3.1, where we let $N_{id} = 7000$ and $N_{ood} = 2000$, where each dataset has 500 data. Selected results appear in the main text in Figure 2, where the full data table is given in Table 25. Additional detailed results of the per-digit analysis are provided below, see Figure 8, and the full data table in Table 21.

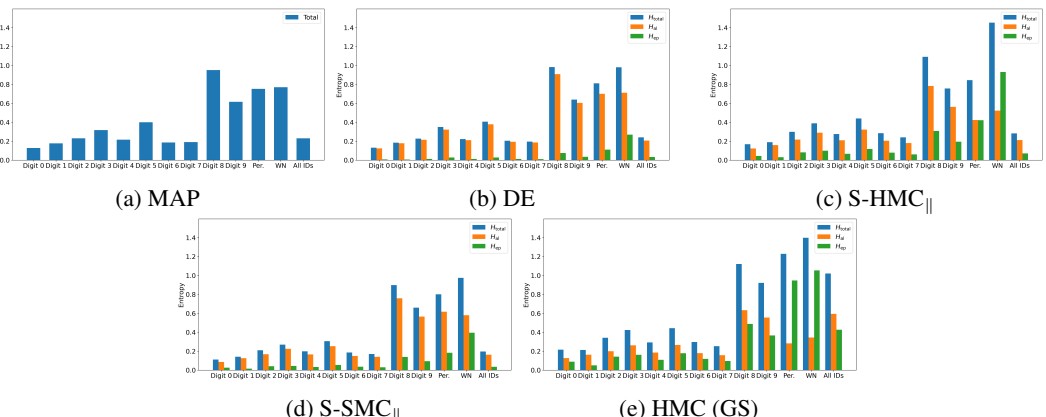

(a) MAP     (b) DE     (c) S-HMC$_\parallel$

(d) S-SMC$_\parallel$     (e) HMC (GS)

Figure 8: Comparison of entropy across groups for MNIST7. S-SMC$_\parallel$ ($P = 1$ chain with $N = 10$), S-HMC$_\parallel$ ($NP$ chains), HMC (GS) ($2e4$ samples), DE ($N$ models) and MAP, with fixed number of leapfrog $L = 1$, $v = 0.1$ and $s = 0.1$ (5 realizations).

### F.2 IMDB

In the IMDb case, the full setting is described in Appendix D.3.2, where we let $N_{ood} = 500$, and each dataset has 100 data.

**Experiments with $v = \frac{1}{40}$.** Results of entropy comparison among MAP, DE and SBMCs are given in Figure 9, showing comparison in the OOD datasets and the correct/incorrect predictions in the ID domain. Additional detailed results of the per-digit analysis are provided below, see Figure 10, and the full data table in Table 22.

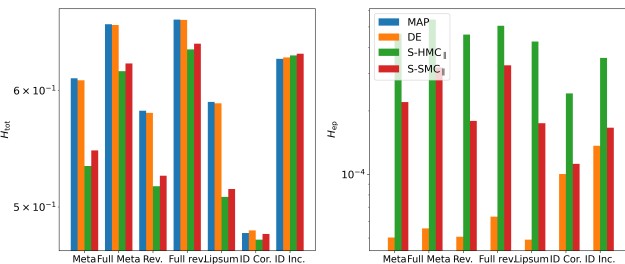

Figure 9: Comparison of average total and epistemic entropy over four out-of-domain classes and correct/incorrect predictions in-domain for IMDb. S-SMC$_\parallel$ ($P = 1$ chain with $N = 10$), S-HMC$_\parallel$ ($NP$ chains), DE ($N$ models) and MAP, with fixed number of leapfrog $L = 1$, $B = 25$, $M = 1$, $v = 0.025$ and $s = 0.1$ (5 realizations).

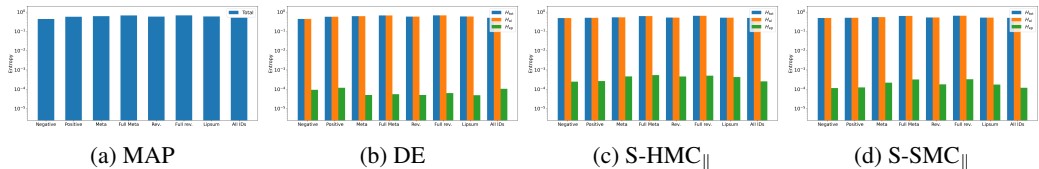

(a) MAP          (b) DE          (c) S-HMC$_\parallel$          (d) S-SMC$_\parallel$

Figure 10: Comparison of entropy across groups for IMDb. S-SMC$_\parallel$ ($P = 1$ chain with $N = 10$), S-HMC$_\parallel$ ($NP$ chains), DE ($N$ models) and MAP, with fixed number of leapfrog $L = 1$, $B = 25$, $M = 1$, $v = 0.025$ and $s = 0.1$ (5 realizations).

**Experiments with $v = 1$.** Results of entropy comparison among MAP, DE and SBMCs are given in Figure 11, showing comparison in the OOD datasets and the correct/incorrect predictions in the ID domain. Additional detailed results of the per-digit analysis are provided below, see Figure 12, and the full data table in Table 23.

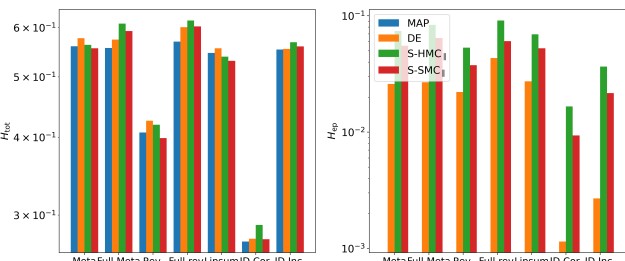

Figure 11: Comparison of average total and epistemic entropy over four out-of-domain classes and correct/incorrect predictions in-domain for IMDb. S-SMC$_\parallel$ ($P = 8$ chain with $N = 10$), S-HMC$_\parallel$ ($NP$ chains), DE ($N$ models) and MAP, with fixed number of leapfrog $L = 1$, $B = 26$, $M = 2$, $v = 1$ and $s = 0.35$ (5 realizations).

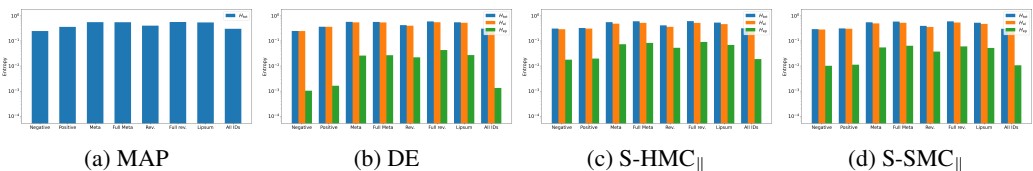

(a) MAP          (b) DE          (c) S-HMC$_\parallel$          (d) S-SMC$_\parallel$

Figure 12: Comparison of entropy across groups for IMDb. S-SMC$_\parallel$ ($P = 8$ chain with $N = 10$), S-HMC$_\parallel$ ($NP$ chains), DE ($N$ models) and MAP, with fixed number of leapfrog $L = 1$, $B = 26$, $M = 2$, $v = 1$ and $s = 0.35$ (5 realizations).

## F.3 CIFAR10

In the CIFAR10 case, the full setting is described in Appendix D.3.3, where we let $N_{\text{id}} = 10000$ and $N_{\text{ood}} = 300$, and each dataset has 100 data points. Results of entropy comparison among MAP, DE and SBMCs are given in Figure 13, showing comparison in the OOD datasets and the correct/incorrect prediction in the ID domain.

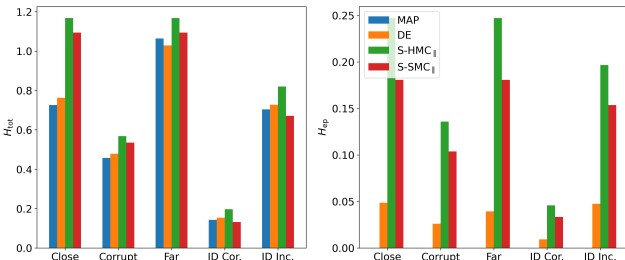

Figure 13: Comparison of average total and epistemic entropy over three out-of-domain classes and correctly/incorrectly predicted ID for CIFAR10. S-SMC$_\parallel$ ($P = 8$ chains with $N = 10$), S-HMC$_\parallel$ ($NP$ chains), DE ($N$) and MAP, with fixed number of leapfrog $L = 1$, $B = 200$, $M = 4$, $v = 0.2$ and $s = 0.05$ (5 realizations).

## G  FURTHER RESULTS OF OOD INFERENCE

Establishment of the meta-classifier of incorrect/OOD data is given in the main text under Out-of-domain inference. Here, the OOD detection is performed in the default and optimal $F_1$ decision rule, respectively.

**The default decision rule**  treats the output probability of "abstain" (out-of-domain or likely misclassified) in the meta-classifier as a binary decision with a fixed cut-off at $0.5$. That is, if the model predicts that there is at least a $50\%$ probability of the data being OOD or incorrectly predicted, it abstains; otherwise, it classifies the data as correctly predicted ID. This rule requires no adjustment beyond the choice of $0.5$. Its behaviour is totally dependent on whether the model's confidence in abstention exceeds the halfway level.

**The optimal $F_1$ decision rule**  adapts the abstention threshold to maximize the $F_1$ score on a held-out set. In practices, the meta-classifier's probabilities are assessed over a grid of potential thresholds ranging from 0 to 1, the F1 score are calculated for each threshold, and the threshold with the highest $F_1$ score is chosen as the optimal $F_1$ threshold. This customised threshold balances false positives and false negatives in the most effective way for the given data distribution, at the cost of requiring a representative validation set. It often outperforms the default decision rule when class proportions or costs of errors are skewed.

### G.1  MNIST7

In the MNIST7 case, the full setting is described in Appendix D.3.1, where we let $N_{\text{id}} = 2000$ and $N_{\text{ood}} = 2000$, where each dataset has $500$ data. Metrics of Precision, Recall, F1 and AUC-ROC metrics are given in Table 7, the normalized confusion rate matrices to show how the OOD domain has been detected from the ID domain are given in Figure 14. Plots for ROC curve and 2-level estimator accuracy are given in Figure 15.

Table 7: Evaluation Metrics using thresholds. S-SMC$_\parallel$ ($P = 1, 8$ chains with $N = 10$) and S-HMC$_\parallel$ ($NP$ chains), with fixed number of leapfrog $L = 1$, $B = 160$, $M = 10$, $v = 0.1$ and $s = 0.1$, on MNIST (5 realizations, $\pm$ s.e. in metrics and bold the first $30\%$ data in mean).

(a) Default decision threshold (0.5).

| $P$ | Method | Precision | Recall | F1 | AUC-ROC |
|---|---|---|---|---|---|
| – | MAP | 0.846±0.014 | 0.162±0.016 | 0.271±0.024 | 0.828±0.013 |
| – | DE | 0.876±0.007 | 0.213±0.011 | 0.342±0.015 | 0.855±0.003 |
| 1 | S-SMC$_\parallel$ | 0.845±0.020 | 0.216±0.037 | 0.338±0.049 | 0.824±0.017 |
| 8 | S-SMC$_\parallel$ | 0.894±0.015 | 0.389±0.046 | 0.537±0.052 | 0.884±0.006 |
| 1 | S-HMC$_\parallel$ | **0.906±0.004** | **0.432±0.020** | **0.584±0.020** | **0.885±0.002** |
| 8 | S-HMC$_\parallel$ | **0.907±0.001** | **0.470±0.007** | **0.619±0.006** | **0.892±0.001** |

(b) Optimal $F_1$ decision threshold.

| $P$ | Method | Precision | Recall | F1 | AUC-ROC |
|---|---|---|---|---|---|
| – | MAP | 0.707±0.013 | 0.898±0.006 | 0.791±0.009 | 0.828±0.013 |
| – | DE | 0.734±0.004 | 0.897±0.003 | 0.807±0.002 | 0.855±0.003 |
| 1 | S-SMC$_\parallel$ | 0.701±0.021 | 0.890±0.015 | 0.783±0.010 | 0.824±0.017 |
| 8 | S-SMC$_\parallel$ | **0.753±0.004** | **0.915±0.004** | **0.826±0.001** | 0.884±0.006 |
| 1 | S-HMC$_\parallel$ | 0.750±0.003 | 0.906±0.004 | 0.820±0.002 | **0.885±0.002** |
| 8 | S-HMC$_\parallel$ | **0.752±0.003** | **0.913±0.004** | **0.825±0.001** | **0.892±0.001** |

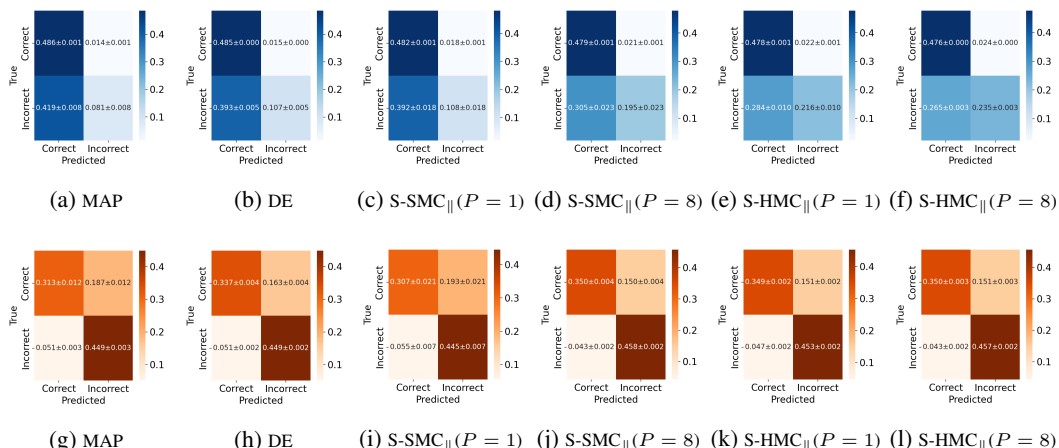

(a) MAP    (b) DE    (c) S-SMC$_\|$ ($P=1$)    (d) S-SMC$_\|$ ($P=8$)    (e) S-HMC$_\|$ ($P=1$)    (f) S-HMC$_\|$ ($P=8$)

(g) MAP    (h) DE    (i) S-SMC$_\|$ ($P=1$)    (j) S-SMC$_\|$ ($P=8$)    (k) S-HMC$_\|$ ($P=1$)    (l) S-HMC$_\|$ ($P=8$)

Figure 14: Averaged confusion rate matrices for OOD prediction on MNIST7, with default decision threshold (top) and optimal $F_1$ decision threshold (bottom). S-SMC$_\|$ ($P = 1, 8$ chains with $N = 10$) and S-HMC$_\|$ ($NP$ chains), with fixed number of leapfrog $L = 1$, $B = 160$, $M = 10$, $v = 0.1$ and $s = 0.1$, on MNIST (5 realizations and $\pm$ s.e. in metrics).

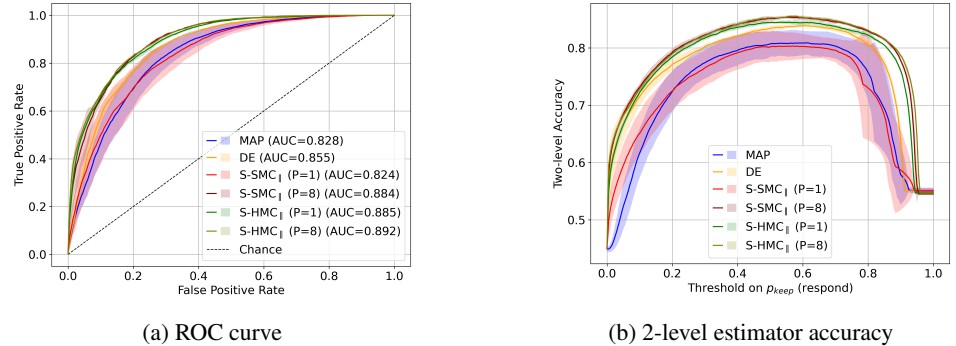

(a) ROC curve        (b) 2-level estimator accuracy

Figure 15: Averaged curve plots for OOD detection on MNIST7. S-SMC$_\|$ ($P = 1, 8$ chains with $N = 10$) and S-HMC$_\|$ ($NP$ chains), with fixed number of leapfrog $L = 1$, $B = 160$, $M = 10$, $v = 0.1$ and $s = 0.1$, on MNIST (5 realizations and $\pm$ s.e. in metrics).

### G.1.1 UNSEEN FAR OOD DETECTION

The meta-classifier is trained following the same procedure used for the MNIST7 before. Evaluation is applied on disjoint test sets constructed for each OOD category. We consider five test configurations, each comprising 1,000 images obtained by pairing the same 500 ID test images with 500 OOD examples of a specific type (digit 8, digit 9, perturbed ID, white noise, or CIFAR-10). For each configuration, we report the area under the corresponding two-level estimator curve (AU2LC) in Table 8 and additionally present the confidence (the optimal $F_1$ decision rule) meta-classifier results in Tables 9,10, 11,12,13.

Table 8: Meta-classifier different OOD sets. Area under the 2-level estimator curve. S-SMC$_\parallel$ ($P = 1, 8$ chains with $N = 10$) and S-HMC$_\parallel$ ($NP$ chains), with fixed number of leapfrog $L = 1$, $B = 160$, $M = 10$, $v = 0.1$ and $s = 0.1$, on MNIST (5 realizations, $\pm$ s.e. in metrics).

| | | Near OOD | | | Far OOD | |
|---|---|---|---|---|---|---|
| $P$ | Method | Digit 8 | Digit 9 | Pert. | White noise | CIFAR10 |
| - | MAP | 0.754 | 0.660 | 0.701 | 0.746 | 0.705 |
| - | DE | 0.792 | 0.646 | 0.785 | 0.867 | 0.847 |
| 1 | S-SMC$_\parallel$ | 0.723 | 0.654 | 0.730 | 0.809 | 0.801 |
| 8 | S-SMC$_\parallel$ | 0.781 | 0.685 | 0.845 | 0.882 | 0.881 |
| 1 | S-HMC$_\parallel$ | 0.774 | 0.660 | 0.812 | 0.878 | 0.873 |
| 8 | S-HMC$_\parallel$ | 0.790 | 0.672 | 0.852 | 0.880 | 0.879 |

Table 9: **Digit 8 OOD set**. Confidence meta-classifier results (optimal $F_1$ decision threshold, MNIST7) for Digit 8 OOD set. S-SMC$_\parallel$ ($P = 1, 8$ chains with $N = 10$) and S-HMC$_\parallel$ ($NP$ chains), with fixed number of leapfrog $L = 1$, $B = 160$, $M = 10$, $v = 0.1$ and $s = 0.1$ (5 realizations, $\pm$ s.e. in metrics).

| $P$ | Method | Precision | Recall | F1 | AUC-ROC |
|---|---|---|---|---|---|
| − | MAP | 0.815±0.008 | 0.896±0.009 | 0.854±0.005 | 0.899±0.007 |
| − | DE | 0.822±0.010 | 0.925±0.011 | 0.870±0.003 | 0.921±0.006 |
| 1 | S-SMC$_\parallel$ | 0.787±0.022 | 0.899±0.016 | 0.838±0.010 | 0.878±0.015 |
| 8 | S-SMC$_\parallel$ | 0.827±0.008 | 0.903±0.006 | 0.863±0.002 | 0.912±0.004 |
| 1 | S-HMC$_\parallel$ | 0.823±0.012 | 0.902±0.016 | 0.860±0.005 | 0.914±0.006 |
| 8 | S-HMC$_\parallel$ | 0.829±0.007 | 0.916±0.009 | 0.870±0.002 | 0.916±0.001 |

Table 10: **Digit 9 OOD set**. Confidence meta-classifier results (optimal $F_1$ decision threshold, MNIST7) for Digit 9 OOD set. S-SMC$_\parallel$ ($P = 1, 8$ chains with $N = 10$) and S-HMC$_\parallel$ ($NP$ chains), with fixed number of leapfrog $L = 1$, $B = 160$, $M = 10$, $v = 0.1$ and $s = 0.1$ (5 realizations, $\pm$ s.e. in metrics).

| $P$ | Method | Precision | Recall | F1 | AUC-ROC |
|---|---|---|---|---|---|
| − | MAP | 0.716±0.010 | 0.917±0.008 | 0.804±0.006 | 0.809±0.008 |
| − | DE | 0.719±0.006 | 0.937±0.010 | 0.814±0.001 | 0.807±0.004 |
| 1 | S-SMC$_\parallel$ | 0.728±0.008 | 0.908±0.011 | 0.808±0.004 | 0.816±0.007 |
| 8 | S-SMC$_\parallel$ | 0.726±0.008 | 0.930±0.008 | 0.815±0.004 | 0.831±0.003 |
| 1 | S-HMC$_\parallel$ | 0.714±0.005 | 0.938±0.003 | 0.811±0.003 | 0.814±0.003 |
| 8 | S-HMC$_\parallel$ | 0.720±0.004 | 0.934±0.008 | 0.813±0.001 | 0.816±0.003 |

Table 11: **Perturbed OOD set**. Confidence meta-classifier results (optimal $F_1$ decision threshold, MNIST7) for Perturbed OOD set. S-SMC$_\parallel$ ($P = 1, 8$ chains with $N = 10$) and S-HMC$_\parallel$ ($NP$ chains), with fixed number of leapfrog $L = 1$, $B = 160$, $M = 10$, $v = 0.1$ and $s = 0.1$ (5 realizations, $\pm$ s.e. in metrics).

| $P$ | Method | Precision | Recall | F1 | AUC-ROC |
|---|---|---|---|---|---|
| − | MAP | 0.759±0.015 | 0.866±0.004 | 0.808±0.008 | 0.840±0.011 |
| − | DE | 0.805±0.007 | 0.907±0.008 | 0.853±0.004 | 0.914±0.008 |
| 1 | S-SMC$_\parallel$ | 0.782±0.023 | 0.882±0.013 | 0.827±0.008 | 0.875±0.012 |
| 8 | S-SMC$_\parallel$ | 0.879±0.010 | 0.920±0.005 | 0.899±0.006 | 0.963±0.004 |
| 1 | S-HMC$_\parallel$ | 0.839±0.011 | 0.911±0.007 | 0.873±0.005 | 0.941±0.005 |
| 8 | S-HMC$_\parallel$ | 0.913±0.015 | 0.901±0.012 | 0.906±0.004 | 0.968±0.002 |

Table 12: **White noise OOD set**. Confidence meta-classifier results (optimal $F_1$ decision threshold, MNIST7) for white noise OOD set. S-SMC$_\parallel$ ($P = 1, 8$ chains with $N = 10$) and S-HMC$_\parallel$ ($NP$ chains), with fixed number of leapfrog $L = 1$, $B = 160$, $M = 10$, $v = 0.1$ and $s = 0.1$ (5 realizations, $\pm$ s.e. in metrics).

| $P$ | Method | Precision | Recall | F1 | AUC-ROC |
|---|---|---|---|---|---|
| – | MAP | 0.809±0.029 | 0.957±0.009 | 0.876±0.019 | 0.889±0.029 |
| – | DE | 0.960±0.022 | 0.940±0.010 | 0.949±0.007 | 0.987±0.004 |
| 1 | S-SMC$_\parallel$ | 0.880±0.047 | 0.932±0.005 | 0.902±0.025 | 0.950±0.024 |
| 8 | S-SMC$_\parallel$ | 0.987±0.006 | 0.941±0.004 | 0.963±0.001 | 0.992±0.000 |
| 1 | S-HMC$_\parallel$ | 0.988±0.006 | 0.932±0.006 | 0.959±0.001 | 0.991±0.001 |
| 8 | S-HMC$_\parallel$ | 0.997±0.001 | 0.927±0.003 | 0.961±0.001 | 0.991±0.000 |

Table 13: **CIFAR OOD set: meta-OOD wrt meta-train**. Confidence meta-classifier results (optimal $F_1$ decision threshold, MNIST7) for CIFAR10 OOD set. S-SMC$_\parallel$ ($P = 1, 8$ chains with $N = 10$) and S-HMC$_\parallel$ ($NP$ chains), with fixed number of leapfrog $L = 1$, $B = 160$, $M = 10$, $v = 0.1$ and $s = 0.1$ (5 realizations, $\pm$ s.e. in metrics).

| $P$ | Method | Precision | Recall | F1 | AUC-ROC |
|---|---|---|---|---|---|
| – | MAP | 0.743±0.044 | 0.906±0.020 | 0.812±0.026 | 0.835±0.042 |
| – | DE | 0.922±0.032 | 0.918±0.010 | 0.919±0.016 | 0.970±0.012 |
| 1 | S-SMC$_\parallel$ | 0.882±0.033 | 0.885±0.033 | 0.883±0.032 | 0.934±0.035 |
| 8 | S-SMC$_\parallel$ | 0.979±0.007 | 0.938±0.005 | 0.958±0.001 | 0.991±0.000 |
| 1 | S-HMC$_\parallel$ | 0.970±0.007 | 0.924±0.006 | 0.946±0.003 | 0.988±0.001 |
| 8 | S-HMC$_\parallel$ | 0.990±0.004 | 0.923±0.003 | 0.955±0.001 | 0.990±0.000 |

## G.2 IMDB

In the IMDb case, the full setting is described in Appendix D.3.2, where we let $N_\text{ood} = 25000$, and each dataset has 5000 data points.

**Experiment with $v = \frac{1}{40}$** Metrics of Precision, Recall, F1 and AUC-ROC metrics are given in Table 14, the normalized confusion rate matrices to show how the OOD domain has been detected from ID domain are given in Figure 16. Plots for ROC curve and 2-level estimator accuracy are given in Figure 17.

Table 14: Performance at the optimal $F_1$ decision threshold. S-SMC$_\parallel$ ($P = 1$ chain with $N = 10$), S-HMC$_\parallel$ ($NP$ chains), DE ($N$ models) and MAP, with fixed number of leapfrog $L = 1$, $B$, $M$, $v = 0.025$ and $s = 0.1$ (5 realizations, $\pm$ s.e. in metrics and bold the first 30% data in mean).

| $P$ | Method | Precision | Recall | F1 | AUC-ROC |
|---|---|---|---|---|---|
| – | MAP | 0.707 ± 0.003 | **0.953 ± 0.001** | 0.811 ± 0.001 | 0.768 ± 0.005 |
| – | DE | 0.856 ± 0.041 | 0.890 ± 0.017 | 0.869 ± 0.016 | 0.896 ± 0.025 |
| 1 | S-SMC$_\parallel$ | 0.673 ± 0.005 | **0.919 ± 0.002** | 0.777 ± 0.004 | 0.777 ± 0.002 |
| 8 | S-SMC$_\parallel$ | **0.935 ± 0.003** | 0.876 ± 0.002 | **0.905 ± 0.000** | **0.935 ± 0.002** |
| 1 | S-HMC$_\parallel$ | 0.844 ± 0.002 | 0.876 ± 0.004 | 0.859 ± 0.001 | 0.905 ± 0.001 |
| 8 | S-HMC$_\parallel$ | **0.970 ± 0.003** | 0.879 ± 0.003 | **0.922 ± 0.000** | **0.944 ± 0.002** |

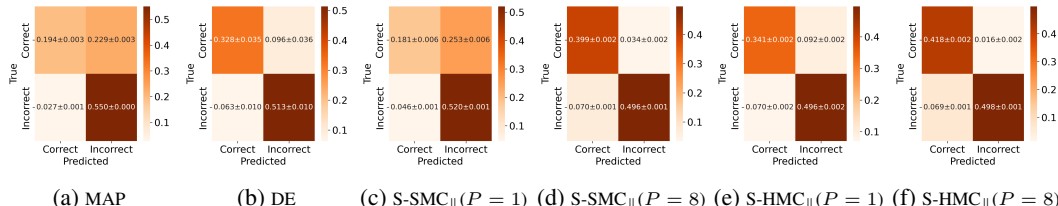

(a) MAP  (b) DE  (c) S-SMC$_{\parallel}$ ($P=1$) (d) S-SMC$_{\parallel}$ ($P=8$) (e) S-HMC$_{\parallel}$ ($P=1$) (f) S-HMC$_{\parallel}$ ($P=8$)

Figure 16: Averaged confusion rate matrices for OOD prediction on IMDb, with optimal $F_1$ decision threshold. S-SMC$_{\parallel}$ ($P=1, 8$ chain with $N=10$), S-HMC$_{\parallel}$ ($NP$ chains), DE ($N$ models) and MAP, with fixed number of leapfrog $L=1$, $B=25$, $M=1$, $v=0.025$ and $s=0.1$ (5 realizations and $\pm$ s.e. in metrics).

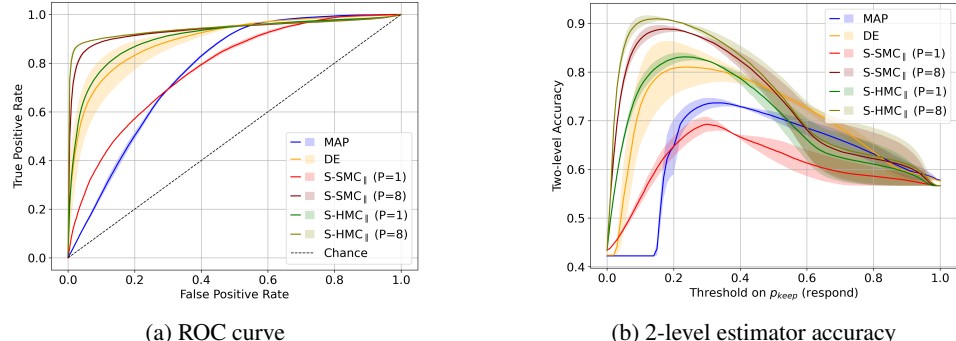

(a) ROC curve         (b) 2-level estimator accuracy

Figure 17: Averaged curve plots for OOD detection in IMDb. S-SMC$_{\parallel}$ ($P=1, 8$ chain with $N=10$), S-HMC$_{\parallel}$ ($NP$ chains), DE ($N$ models) and MAP, with fixed number of leapfrog $L=1$, $B=25$, $M=1$, $v=0.025$ and $s=0.1$ (5 realizations and $\pm$ s.e. in metrics).

**Experiment with $v=1$.** Metrics of Precision, Recall, F1 and AUC-ROC metrics are given in Table 15, the normalized confusion rate matrices to show how the OOD domain has been detected from the ID domain are given in Figure 18. The plots for the ROC curve and 2-level estimator accuracy are given in Figure 19.

Table 15: Performance at the optimal $F_1$ decision threshold. S-SMC$_{\parallel}$ ($P=1$ chain with $N=10$), S-HMC$_{\parallel}$ ($NP$ chains), DE ($N$ models) and MAP, with fixed number of leapfrog $L=1$, $B$, $M$, $v=1$ and $s=0.35$ (5 realizations, $\pm$ s.e. in metrics and bold the first 30% data in mean).

| $P$ | Method | Precision | Recall | F1 | AUC-ROC |
|---|---|---|---|---|---|
| – | MAP | 0.733±0.004 | **0.897±0.003** | 0.807±0.002 | 0.809±0.006 |
| – | DE | **0.968±0.004** | 0.880±0.004 | **0.922±0.001** | **0.959±0.003** |
| 1 | S-SMC$_{\parallel}$ | 0.791±0.000 | 0.871±0.003 | 0.829±0.001 | 0.889±0.001 |
| 8 | S-SMC$_{\parallel}$ | 0.947±0.003 | 0.878±0.003 | 0.911±0.000 | 0.944±0.002 |
| 1 | S-HMC$_{\parallel}$ | 0.923±0.002 | **0.885±0.001** | 0.904±0.001 | 0.943±0.002 |
| 8 | S-HMC$_{\parallel}$ | **0.965±0.004** | 0.880±0.003 | **0.920±0.000** | **0.948±0.002** |

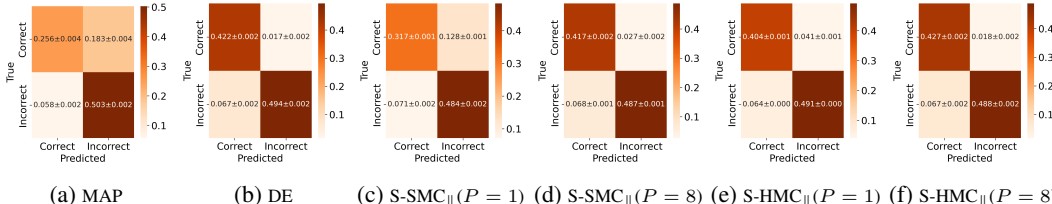

(a) MAP   (b) DE   (c) S-SMC$_\parallel$($P=1$)   (d) S-SMC$_\parallel$($P=8$)   (e) S-HMC$_\parallel$($P=1$)   (f) S-HMC$_\parallel$($P=8$)

Figure 18: Averaged confusion rate matrices for OOD prediction on IMDb, with optimal $F_1$ decision threshold. S-SMC$_\parallel$ ($P = 1, 8$ chain with $N = 10$), S-HMC$_\parallel$ ($NP$ chains), DE ($N$ models) and MAP, with fixed number of leapfrog $L = 1$, $B = 26$, $M = 2$, $v = 1$ and $s = 0.35$ (5 realizations and $\pm$ s.e. in metrics).

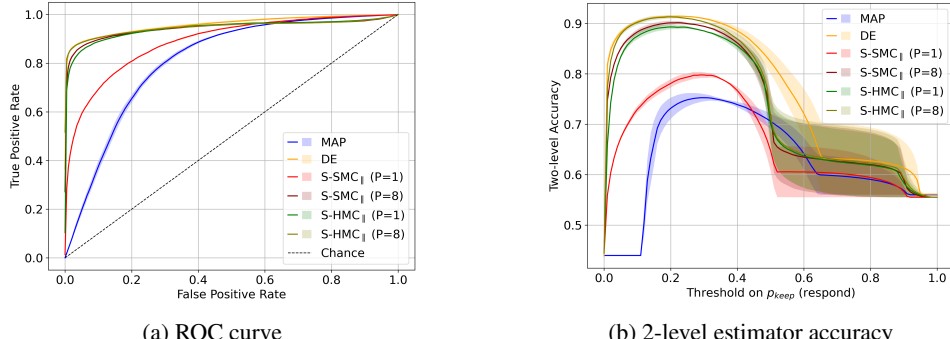

(a) ROC curve   (b) 2-level estimator accuracy

Figure 19: Averaged curve plots for OOD detection in IMDb. S-SMC$_\parallel$ ($P = 1, 8$ chain with $N = 10$), S-HMC$_\parallel$ ($NP$ chains), DE ($N$ models) and MAP, with fixed number of leapfrog $L = 1$, $B = 26$, $M = 2$, $v = 1$ and $s = 0.35$ (5 realizations and $\pm$ s.e. in metrics).

### G.3   CIFAR10

In the CIFAR10 case, the full setting is described in Appendix D.3.3, where we let $N_{\text{id}} = 9000$ and $N_{\text{ood}} = 9000$, and each dataset has 3000 data points. Metrics of Precision, Recall, F1 and AUC-ROC metrics are given in Table 16, the normalized confusion rate matrices to show how the OOD domain has been detected from the ID domain are given in Figure 20. Plots for ROC curve and 2-level estimator accuracy are given in Figure 21.

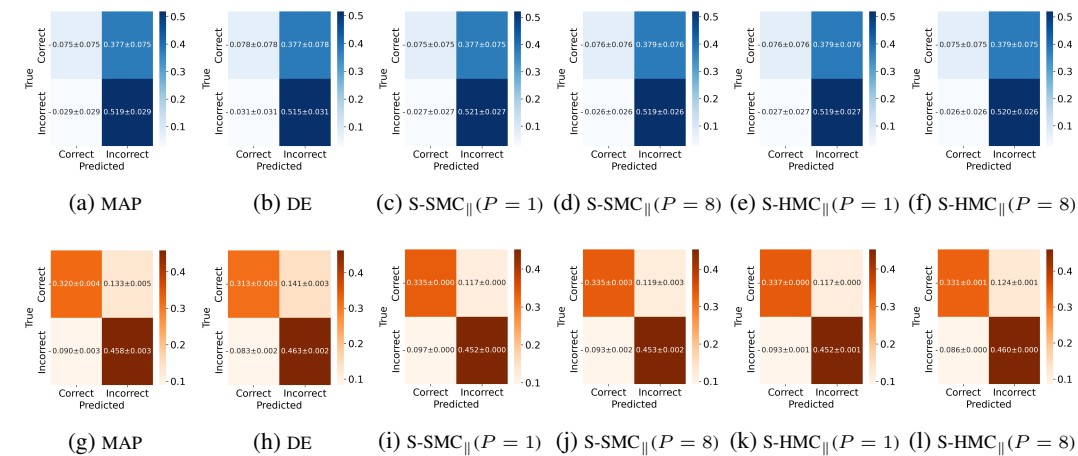

Figure 20: Averaged confusion rate matrices for OOD prediction on CIFAR10, with default decision threshold (top) and optimal $F_1$ decision threshold (bottom). S-SMC$_\parallel$ ($P = 1, 8$ chains with $N = 10$), S-HMC$_\parallel$ ($NP$ chains), DE ($N$) and MAP, with fixed number of leapfrog $L = 1$, $B = 200$, $M = 4$, $v = 0.2$ and $s = 0.05$ (5 realizations and $\pm$ s.e. in metrics).

Table 16: Evaluation Metrics using thresholds. S-SMC$_\parallel$ ($P = 1, 8$ chains with $N = 10$), S-HMC$_\parallel$ ($NP$ chains), DE ($N$) and MAP, with fixed number of leapfrog $L = 1$, $B = 200$, $M = 4$, $v = 0.2$ and $s = 0.05$ (5 realizations, $\pm$ s.e. in metrics and bold the first 30% data in mean).

(a) Default decision threshold (0.5).

| $P$ | Method | Precision | Recall | F1 | AUC-ROC |
|---|---|---|---|---|---|
| – | MAP | $0.606 \pm 0.058$ | $0.947 \pm 0.053$ | $0.723 \pm 0.015$ | $0.856 \pm 0.001$ |
| – | DE | $\mathbf{0.608 \pm 0.062}$ | $0.943 \pm 0.057$ | $0.721 \pm 0.015$ | $0.858 \pm 0.002$ |
| 1 | S-SMC$_\parallel$ | $\mathbf{0.607 \pm 0.059}$ | $0.951 \pm 0.049$ | $\mathbf{0.726 \pm 0.018}$ | $0.861 \pm 0.001$ |
| 8 | S-SMC$_\parallel$ | $0.606 \pm 0.060$ | $\mathbf{0.952 \pm 0.048}$ | $0.725 \pm 0.019$ | $\mathbf{0.864 \pm 0.000}$ |
| 1 | S-HMC$_\parallel$ | $0.606 \pm 0.060$ | $0.951 \pm 0.049$ | $0.724 \pm 0.019$ | $\mathbf{0.864 \pm 0.000}$ |
| 8 | S-HMC$_\parallel$ | $0.605 \pm 0.060$ | $\mathbf{0.953 \pm 0.047}$ | $0.725 \pm 0.019$ | $\mathbf{0.867 \pm 0.000}$ |

(b) Optimal $F_1$ decision threshold.

| $P$ | Method | Precision | Recall | F1 | AUC-ROC |
|---|---|---|---|---|---|
| – | MAP | $0.776 \pm 0.005$ | $0.836 \pm 0.006$ | $0.805 \pm 0.001$ | $0.856 \pm 0.001$ |
| – | DE | $0.767 \pm 0.003$ | $\mathbf{0.848 \pm 0.003}$ | $0.805 \pm 0.001$ | $0.858 \pm 0.002$ |
| 1 | S-SMC$_\parallel$ | $\mathbf{0.794 \pm 0.000}$ | $0.824 \pm 0.001$ | $0.809 \pm 0.000$ | $0.861 \pm 0.001$ |
| 8 | S-SMC$_\parallel$ | $0.792 \pm 0.003$ | $0.830 \pm 0.003$ | $\mathbf{0.811 \pm 0.000}$ | $\mathbf{0.864 \pm 0.000}$ |
| 1 | S-HMC$_\parallel$ | $\mathbf{0.794 \pm 0.000}$ | $0.829 \pm 0.001$ | $\mathbf{0.811 \pm 0.000}$ | $\mathbf{0.864 \pm 0.000}$ |
| 8 | S-HMC$_\parallel$ | $0.788 \pm 0.001$ | $\mathbf{0.842 \pm 0.001}$ | $\mathbf{0.814 \pm 0.000}$ | $\mathbf{0.867 \pm 0.000}$ |

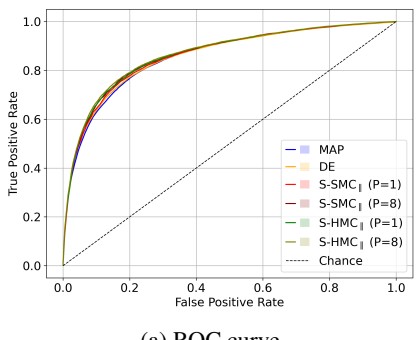

(a) ROC curve

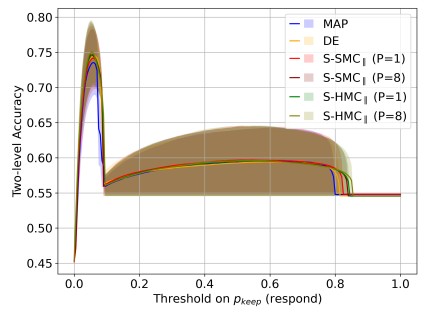

(b) Total accuracy over thresholds

Figure 21: Averaged curve plots for OOD detection in CIFAR10. S-SMC$_\parallel$ ($P = 1, 8$ chains with $N = 10$), S-HMC$_\parallel$ ($NP$ chains), DE ($N$) and MAP, with fixed number of leapfrog $L = 1$, $B = 200$, $M = 4$, $v = 0.2$ and $s = 0.05$ (5 realizations and $\pm$ s.e. in metrics).

# H  FURTHER RESULTS OF ABLATIONS IN PRACTICAL SBMC($s < \frac{1}{2}$)

## H.1  MNIST7

Experiments in this section are tested on the (filtered) MNIST7 dataset with the model setting stated in Appendix D.3.1.

Table 17 shows the performance as the tuning parameter $s$ varies. Figure 22 and 23 show the trend of the SBMC$_\parallel$ in different values of the tuning parameter $s$ as $P$ increases. Table 24 and 25 give the corresponding full data results of the below figures.

Table 17: Comparison of different $s$ of (S-)SMC$_\parallel$ ($P = 1, 8$ chain with $N = 10$), (S-)HMC$_\parallel$ ($NP$ chains), MAP and DE ($NP$ models), with fixed number of leapfrog $L = 1$ and $v = 0.1$, on MNIST7 (5 realizations and $\pm$ s.e. in accuracy).

| $s$ | Method | Epochs | Accuracy | NLL | Brier | $H_{\mathrm{ep}}$ | | | | | |
| --- | --- | --- | --- | --- | --- | --- | --- | --- | --- | --- | --- |
| | | | | | | ID | | OD | | | |
| | | | | | | cor. | inc. | 8 | 9 | wn | per. |
| 1 | HMC (GS) | 2e4 | 93.61±0.41 | 2.224e-1 | 1.015e-1 | 9.621e-2 | 4.097e-1 | 4.614e-1 | 3.119e-1 | 1.126e+0 | 7.919e-1 |
| | HMC (GS) | 2e5 | 94.77±0.21 | 1.942e-1 | 8.700e-2 | 1.204e-1 | 4.928e-1 | 5.635e-1 | 4.067e-1 | 1.602e+0 | 1.031e+0 |
| | HMC (GS) | 1.8e6 | 95.13±0.02 | 1.882e-1 | 8.345e-2 | 1.281e-1 | 5.185e-1 | 5.856e-1 | 4.244e-1 | 1.682e+0 | 1.112e+0 |
| 1 | SMC$_\parallel$ | 173.0 | 79.74±2.71 | 6.230e-1 | 2.920e-1 | 1.337e-2 | 3.339e-2 | 3.321e-2 | 2.775e-2 | 6.482e-2 | 5.512e-2 |
| | HMC$_\parallel$ | 160 | 78.41±2.39 | 1.273e+0 | 5.799e-1 | 3.026e-1 | 3.247e-1 | 3.173e-1 | 2.988e-1 | 6.626e-1 | 1.099e+0 |
| 0.5 | S-SMC$_\parallel$ | 161.0 | 84.18±0.64 | 4.827e-1 | 2.304e-1 | 1.556e-2 | 4.082e-2 | 4.000e-2 | 3.234e-2 | 1.238e-1 | 6.418e-2 |
| | S-HMC$_\parallel$ | 160 | 85.26±1.06 | 8.234e-1 | 3.672e-1 | 2.993e-1 | 3.704e-1 | 3.627e-1 | 3.271e-1 | 8.832e-1 | 8.030e-1 |
| 0.25 | S-SMC$_\parallel$ | 166.6 | 90.35±0.26 | 3.300e-1 | 1.441e-1 | 2.257e-2 | 1.094e-1 | 1.146e-1 | 7.791e-2 | 3.888e-1 | 1.996e-1 |
| | $P = 8$ | 161.5 | 93.00±0.11 | 2.366e-1 | 1.096e-1 | 8.828e-2 | 3.717e-1 | 2.984e-1 | 2.089e-1 | 7.488e-1 | 4.585e-1 |
| | S-HMC$_\parallel$ | 160 | 92.79±0.19 | 2.571e-1 | 1.156e-1 | 1.133e-1 | 4.232e-1 | 4.985e-1 | 3.225e-1 | 1.289e+0 | 6.280e-1 |
| | $P = 8$ | 160 | 93.15±0.05 | 2.490e-1 | 1.127e-1 | 1.384e-1 | 4.788e-1 | 5.572e-1 | 3.678e-1 | 1.349e+0 | 7.311e-1 |
| 0.1 | S-SMC$_\parallel$ | 170.0 | 92.17±0.37 | 2.671e-1 | 1.186e-1 | 2.642e-2 | 1.288e-1 | 1.384e-1 | 9.406e-2 | 3.943e-1 | 1.832e-1 |
| | $P = 8$ | 178.0 | 93.26±0.16 | 2.259e-1 | 1.025e-1 | 5.871e-2 | 2.725e-1 | 2.440e-1 | 1.637e-1 | 7.238e-1 | 3.823e-1 |
| | S-HMC$_\parallel$ | 160 | 92.96±0.17 | 2.326e-1 | 1.071e-1 | 5.624e-2 | 2.645e-1 | 3.072e-1 | 1.941e-1 | 9.304e-1 | 4.216e-1 |
| | $P = 8$ | 160 | 93.12±0.08 | 2.310e-1 | 1.072e-1 | 6.982e-2 | 2.993e-1 | 3.524e-1 | 2.258e-1 | 1.067e+0 | 4.780e-1 |
| 0.01 | S-SMC$_\parallel$ | 183.6 | 92.57±0.37 | 2.439e-1 | 1.121e-1 | 1.149e-2 | 5.904e-2 | 6.445e-2 | 4.602e-2 | 2.187e-1 | 1.008e-1 |
| | S-HMC$_\parallel$ | 162 | 92.95±0.10 | 2.289e-1 | 1.069e-1 | 1.912e-2 | 1.015e-1 | 1.238e-1 | 7.814e-2 | 4.678e-1 | 1.945e-1 |
| 0 | MAP | 160.2 | 92.32±0.37 | 2.527e-1 | 1.163e-1 | 0 | 0 | 0 | 0 | 0 | 0 |
| | DE ($N$) | 176.5 | 92.40±0.15 | 2.455e-1 | 1.148e-1 | 1.059e-2 | 5.646e-2 | 7.433e-2 | 3.468e-2 | 2.690e-1 | 1.1056e-1 |
| | DE ($8N$) | 178.38 | 92.54±0.06 | 2.393e-1 | 1.124e-1 | 1.111e-2 | 5.980e-2 | 7.846e-2 | 4.016e-2 | 2.935e-1 | 1.188e-1 |

| $s$ | Method | $H_{\mathrm{tot}}$ | | | | | |
| --- | --- | --- | --- | --- | --- | --- | --- |
| | | ID | | OD | | | |
| | | cor. | inc. | 8 | 9 | wn | per. |
| 1 | HMC (GS) | 2.621e-1 | 9.652e-1 | 1.110e+0 | 8.198e-1 | 1.492e+0 | 1.081e+0 |
| | HMC (GS) | 2.852e-1 | 1.033e+0 | 1.204e+0 | 9.322e-1 | 1.915e+0 | 1.296e+0 |
| | HMC (GS) | 2.948e-1 | 1.057e+0 | 1.223e+0 | 9.532e-1 | 2.012e+0 | 1.384e+0 |
| 1 | SMC$_\parallel$ | 5.506e-1 | 1.078e+0 | 1.138e+0 | 9.851e-1 | 6.426e-1 | 9.171e-1 |
| | HMC$_\parallel$ | 1.854e+0 | 1.962e+0 | 1.988e+0 | 1.927e+0 | 1.965e+0 | 1.844e+0 |
| 0.5 | S-SMC$_\parallel$ | 4.363e-1 | 1.019e+0 | 1.127e+0 | 9.294e-1 | 8.128e-1 | 8.712e-1 |
| | S-HMC$_\parallel$ | 1.427e+0 | 1.752e+0 | 1.837e+0 | 1.667e+0 | 1.857e+0 | 1.694e+0 |
| 0.25 | S-SMC$_\parallel$ | 1.508e-1 | 6.679e-1 | 8.384e-1 | 5.945e-1 | 8.495e-1 | 7.445e-1 |
| | $P = 8$ | 1.247e-1 | 6.641e-1 | 1.000e+0 | 7.354e-1 | 1.177e+0 | 9.931e-1 |
| | S-HMC$_\parallel$ | 3.149e-1 | 1.026e+0 | 1.220e+0 | 8.606e-1 | 1.624e+0 | 1.019e+0 |
| | $P = 8$ | 3.456e-1 | 1.070e+0 | 1.267e+0 | 9.025e-1 | 1.714e+0 | 1.111e+0 |
| 0.1 | S-SMC$_\parallel$ | 1.536e-1 | 7.042e-1 | 8.975e-1 | 6.591e-1 | 9.743e-1 | 8.001e-1 |
| | $P = 8$ | 1.374e-1 | 7.075e-1 | 1.001e+0 | 7.307e-1 | 1.216e+0 | 9.805e-1 |
| | S-HMC$_\parallel$ | 2.343e-1 | 9.132e-1 | 1.091e+0 | 7.567e-1 | 1.452e+0 | 8.443e-1 |
| | $P = 8$ | 2.553e-1 | 9.380e-1 | 1.127e+0 | 7.893e-1 | 1.543e+0 | 8.937e-1 |
| 0.01 | S-SMC$_\parallel$ | 1.737e-1 | 7.571e-1 | 9.632e-1 | 6.607e-1 | 9.254e-1 | 8.511e-1 |
| | S-HMC$_\parallel$ | 1.995e-1 | 8.232e-1 | 1.003e+0 | 6.991e-1 | 1.234e+0 | 6.726e-1 |
| 0 | MAP | 1.833e-1 | 7.645e-1 | 9.507e-1 | 6.157e-1 | 7.682e-1 | 7.839e-1 |
| | DE ($N$) | 1.919e-1 | 7.899e-1 | 9.821e-1 | 6.393e-1 | 9.806e-1 | 8.112e-1 |
| | DE ($8N$) | 1.938e-1 | 7.988e-1 | 1.001e+0 | 6.532e-1 | 1.067e+0 | 8.133e-1 |

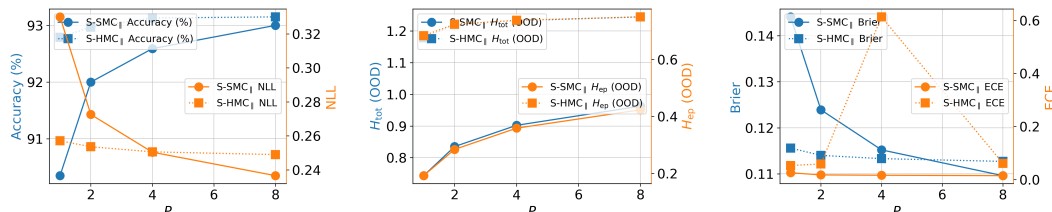

Figure 22: Comparison of S-SMC$_\parallel$ ($P$ chains with $N = 10$) and S-HMC$_\parallel$ ($NP$ chains), with fixed number of leapfrog $L = 1$, $B = 160$, $M = 7$, $v = 0.1$ and $s = 0.25$, on MNIST7 (5 realizations).

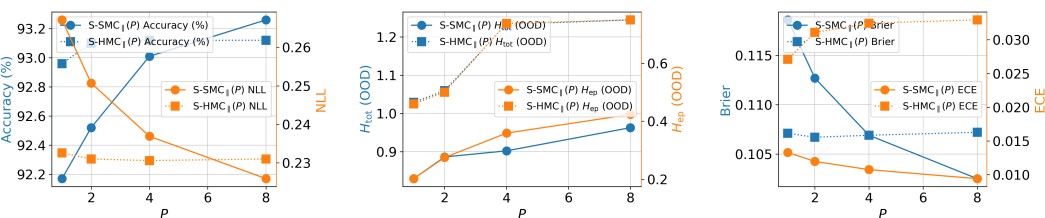

Figure 23: Comparison of S-SMC$_\parallel$ ($P$ chains with $N = 10$) and S-HMC$_\parallel$ ($NP$ chains), with fixed number of leapfrog $L = 1$, $B = 160$, $M = 10$, $v = 0.1$ and $s = 0.1$, on MNIST7 (5 realizations).

### H.2 IMDB

The experiments in this section are tested on the IMDb dataset with the model setting stated in Appendix D.3.2.

**Experiment with ($v = \frac{1}{40}$).** Summary metrics of IMDb dataset with $v = 0.025$ and $s = 0.1$ is shown in the left spider-plot in Figure 3. Table 18 shows the performance as the tuning parameter $s$ varies. Figure 24 shows the trend for the SBMC methods as $P$ increases, where the full data can be found in Table 26.

Table 18: Comparison of S-SMC$_\parallel$ ($N = 10$), S-HMC$_\parallel$ ($N$ chains), DE ($N$ models) and MAP, with fixed number of leapfrog $L = 1$, $B = 25$, $M = 1$ and $v = 0.025$, on IMDb (5 realizations and $\pm$ s.e. in accuracy).

| $s$ | Method | Ep. | Acc. | NLL | | $H_{ep}$ | | | | | |
|---|---|---|---|---|---|---|---|---|---|---|---|
| | | | | | | ID | | OD | | | | |
| | | | | | | cor. | inc. | reviews | meta | lipsum | full reviews | full meta |
| 0.1 | S-SMC$_\parallel$ | 18.60 | 86.70±0.03 | 3.655e-1 | | 1.122e-4 | 1.664e-4 | 1.792e-4 | 2.200e-4 | 1.749e-4 | 3.285e-4 | 3.212e-4 |
| | $P = 8$ | 19.15 | 86.69±0.01 | 3.653e-1 | | 2.531e-4 | 3.697e-4 | 4.744e-4 | 4.971e-4 | 4.862e-4 | 5.876e-4 | 6.105e-4 |
| | S-HMC$_\parallel$ | 25 | 86.70±0.01 | 3.634e-1 | | 2.418e-4 | 3.565e-4 | 4.598e-4 | 4.633e-4 | 4.260e-4 | 5.057e-4 | 5.410e-4 |
| | $P = 8$ | 25 | 86.72±0.00 | 3.633e-1 | | 2.766e-4 | 4.022e-4 | 5.694e-4 | 6.062e-4 | 5.637e-4 | 7.438e-4 | 7.051e-4 |
| 0 | MAP | 25.00 | 84.47±0.09 | 3.911e-1 | | 0 | 0 | 0 | 0 | 0 | 0 | 0 |
| | DE ($N$) | 25.86 | 84.76±0.08 | 3.888e-1 | | 1.005e-04 | 1.366e-04 | 5.064e-5 | 5.026e-5 | 4.909e-5 | 6.302e-5 | 5.548e-5 |

| $s$ | Method | Brier | ECE | | $H_{tot}$ | | | | | |
|---|---|---|---|---|---|---|---|---|---|---|
| | | | | | | ID | | OD | | | |
| | | | | | | cor. | inc. | reviews | meta | lipsum | full reviews | full meta |
| 0.1 | S-SMC$_\parallel$ | 1.093e-1 | 3.699e-1 | | 4.792e-1 | 6.357e-1 | 5.251e-1 | 5.463e-1 | 5.142e-1 | 6.457e-1 | 6.261e-1 |
| | $P = 8$ | 1.092e-1 | 3.698e-1 | | 4.788e-1 | 6.355e-1 | 5.213e-1 | 5.406e-1 | 5.115e-1 | 6.430e-1 | 6.236e-1 |
| | S-HMC$_\parallel$ | 1.086e-1 | 3.694e-1 | | 4.750e-1 | 6.340e-1 | 5.165e-1 | 5.331e-1 | 5.079e-1 | 6.400e-1 | 6.186e-1 |
| | $P = 8$ | 1.086e-1 | 3.701e-1 | | 4.752e-1 | 6.341e-1 | 5.184e-1 | 5.359e-1 | 5.085e-1 | 6.426e-1 | 6.209e-1 |
| 0 | MAP | 1.204e-1 | 4.389e-1 | | 4.800e-1 | 6.306e-1 | 5.814e-1 | 6.117e-1 | 5.894e-1 | 6.705e-1 | 6.658e-1 |
| | DE ($N$) | 1.193e-1 | 4.340e-1 | | 4.819e-1 | 6.319e-1 | 5.793e-1 | 6.098e-1 | 5.882e-1 | 6.702e-1 | 6.649e-1 |

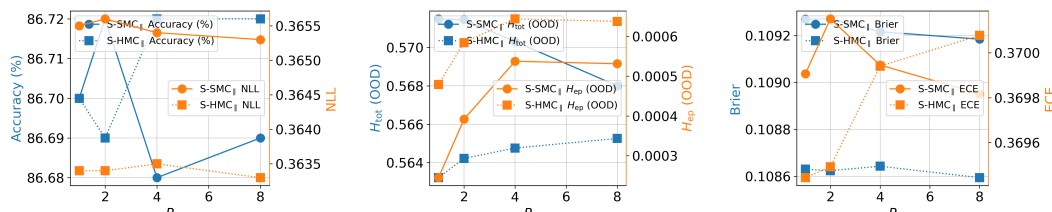

Figure 24: Comparison of S-SMC$_\parallel$ ($P$ chains with $N = 10$) and S-HMC$_\parallel$ ($NP$ chains), with fixed $L = 1$, $B = 25$, $M = 1$, $v = 0.025$, $s = 0.1$, on IMDb (5 realizations)

**Experiments with $v = 1$.** Summary metrics of IMDb dataset with $v = 15$ and $s = 0.35$ are shown in the spider-plot in Figure 25. Table 19 shows the performance as the tuning parameter $s$ vary. Figure 26, 27 and 28 give the full convergence of SBMC$_\parallel$ with increasing $P$. The corresponding full data results are given in the Table 27 , 28 and 29.

Table 19: Comparison of different $s$ of S-SMC$_\parallel$ ($N = 10$) and S-HMC$_\parallel$ ($N$ chains), with fixed number of leapfrog $L = 1$ and $v = 1$, on IMDb (5 realizations and $\pm$ s.e. in accuracy).

| $s$ | Method | Ep. | Acc. | NLL | $H_{ep}$ | | | | | | |
|---|---|---|---|---|---|---|---|---|---|---|---|
| | | | | | ID | | OD | | | | |
| | | | | | cor. | inc. | reviews | meta | lipsum | full reviews | full meta |
| 0.35 | S-SMC$_\parallel$ | 27.40 | 88.27±0.07 | 2.803e-1 | 6.177e-4 | 1.460e-3 | 1.581e-3 | 2.495e-3 | 2.187e-3 | 2.279e-3 | 2.504e-3 |
| | $P = 8$ | 27.63 | 88.88±0.03 | 2.714e-1 | 9.342e-3 | 2.164e-2 | 3.756e-2 | 5.515e-2 | 5.260e-2 | 6.049e-2 | 6.435e-2 |
| | S-HMC$_\parallel$ | 26 | 88.81±0.01 | 2.750e-1 | 1.565e-2 | 3.463e-2 | 5.414e-2 | 7.021e-2 | 6.872e-2 | 8.407e-2 | 7.930e-2 |
| | $P = 8$ | 26 | 88.93±0.02 | 2.737e-1 | 1.662e-2 | 3.651e-2 | 5.315e-2 | 7.391e-2 | 6.917e-2 | 9.098e-2 | 8.360e-2 |
| 0.25 | S-SMC$_\parallel$ | 29.6 | 88.27±0.10 | 2.807e-1 | 2.512e-4 | 6.069e-4 | 4.733e-4 | 7.166e-4 | 8.590e-4 | 9.313e-4 | 9.520e-4 |
| | $P = 8$ | 28.5 | 88.87±0.03 | 2.720e-1 | 8.124e-3 | 1.872e-2 | 3.066e-2 | 5.057e-2 | 4.691e-2 | 6.403e-2 | 5.777e-2 |
| | S-HMC$_\parallel$ | 26 | 88.83±0.02 | 2.745e-1 | 1.269e-2 | 2.830e-2 | 4.522e-2 | 5.927e-2 | 5.886e-2 | 7.342e-2 | 6.803e-2 |
| | $P = 8$ | 26 | 88.92±0.02 | 2.734e-1 | 1.337e-2 | 2.964e-2 | 4.442e-2 | 6.215e-2 | 5.863e-2 | 7.869e-2 | 7.108e-2 |
| 0.1 | S-SMC$_\parallel$ | 24 | 88.54±0.11 | 2.762e-1 | 1.820e-4 | 4.315e-4 | 4.819e-4 | 5.915e-4 | 6.766e-4 | 7.711e-4 | 6.500e-4 |
| | $P = 8$ | 23.7 | 88.92±0.01 | 2.711e-1 | 4.207e-3 | 9.768e-3 | 2.182e-2 | 3.140e-2 | 2.700e-2 | 3.352e-2 | 3.540e-2 |
| | S-HMC$_\parallel$ | 26 | 88.86±0.02 | 2.726e-1 | 5.753e-3 | 1.319e-2 | 2.712e-2 | 3.551e-2 | 3.561e-2 | 5.110e-2 | 4.289e-2 |
| | $P = 8$ | 26 | 88.93±0.01 | 2.721e-1 | 6.065e-3 | 1.386e-2 | 2.792e-2 | 3.888e-2 | 3.653e-2 | 5.408e-2 | 4.638e-2 |
| 0 | MAP | 52 | 87.97±0.04 | 2.854e-1 | – | 0 | 0 | 0 | 0 | 0 | 0 |
| | DE ($N$) | 26.52 | 87.75±0.02 | 2.921e-1 | 3.144e-3 | 7.055e-3 | 4.514e-2 | 5.054e-2 | 5.386e-2 | 7.963e-2 | 5.318e-2 |
| | DE ($8N$) | 25.86 | 87.70 ±0.01 | 2.925e-1 | 3.469e-3 | 7.608e-3 | 4.394e-2 | 5.622e-2 | 5.089e-2 | 7.435e-2 | 6.198e-2 |

| $s$ | Method | Brier | ECE | $H_{tot}$ | | | | | | |
|---|---|---|---|---|---|---|---|---|---|---|
| | | | | ID | | OD | | | | |
| | | | | cor. | inc. | reviews | meta | lipsum | full reviews | full meta |
| 0.35 | S-SMC$_\parallel$ | 8.547e-2 | 3.832e-1 | 2.643e-1 | 5.482e-1 | 3.802e-1 | 5.116e-1 | 5.172e-1 | 5.581e-1 | 5.286e-1 |
| | $P = 8$ | 8.206e-2 | 3.899e-1 | 2.744e-1 | 5.596e-1 | 3.987e-1 | 5.555e-1 | 5.304e-1 | 6.025e-1 | 5.920e-1 |
| | S-HMC$_\parallel$ | 8.298e-2 | 3.889e-1 | 2.890e-1 | 5.681e-1 | 4.289e-1 | 5.583e-1 | 5.556e-1 | 6.133e-1 | 6.120e-1 |
| | $P = 8$ | 8.254e-2 | 3.904e-1 | 2.893e-1 | 5.683e-1 | 4.188e-1 | 5.626e-1 | 5.386e-1 | 6.156e-1 | 6.088e-1 |
| 0.25 | S-SMC$_\parallel$ | 8.548e-2 | 3.825e-1 | 2.673e-1 | 5.500e-1 | 3.562e-1 | 4.872e-1 | 4.727e-1 | 5.548e-1 | 5.609e-1 |
| | $P = 8$ | 8.220e-2 | 3.901e-1 | 2.772e-1 | 5.609e-1 | 3.891e-1 | 5.447e-1 | 5.340e-1 | 6.036e-1 | 5.874e-1 |
| | S-HMC$_\parallel$ | 8.286e-2 | 3.896e-1 | 2.873e-1 | 5.667e-1 | 4.239e-1 | 5.585e-1 | 5.533e-1 | 6.117e-1 | 6.076e-1 |
| | $P = 8$ | 8.249e-2 | 3.904e-1 | 2.871e-1 | 5.665e-1 | 4.138e-1 | 5.595e-1 | 5.338e-1 | 6.135e-1 | 6.047e-1 |
| 0.1 | S-SMC$_\parallel$ | 8.387e-2 | 3.869e-1 | 2.721e-1 | 5.544e-1 | 3.970e-1 | 5.202e-1 | 5.063e-1 | 5.686e-1 | 5.684e-1 |
| | $P = 8$ | 8.190e-2 | 3.906e-1 | 2.752e-1 | 5.588e-1 | 3.920e-1 | 5.407e-1 | 5.143e-1 | 5.959e-1 | 5.865e-1 |
| | S-HMC$_\parallel$ | 8.232e-2 | 3.898e-1 | 2.802e-1 | 5.618e-1 | 4.088e-1 | 5.505e-1 | 5.384e-1 | 6.066e-1 | 5.975e-1 |
| | $P = 8$ | 8.217e-2 | 3.906e-1 | 2.801e-1 | 5.613e-1 | 4.028e-1 | 5.510e-1 | 5.231e-1 | 6.081e-1 | 5.961e-1 |
| 0 | MAP | 8.714e-2 | 4.350e-1 | 2.721e-1 | 5.531e-1 | 4.071e-1 | 5.598e-1 | 5.462e-1 | 5.695e-1 | 5.561e-1 |
| | DE ($N$) | 8.928e-2 | 4.343e-1 | 2.850e-1 | 5.580e-1 | 4.497e-1 | 5.808e-1 | 5.609e-1 | 6.185e-1 | 6.032e-1 |
| | DE ($8N$) | 8.941e-2 | 4.354e-1 | 2.859e-1 | 5.588e-1 | 4.533e-1 | 5.888e-1 | 5.708e-1 | 6.187e-1 | 6.068e-1 |

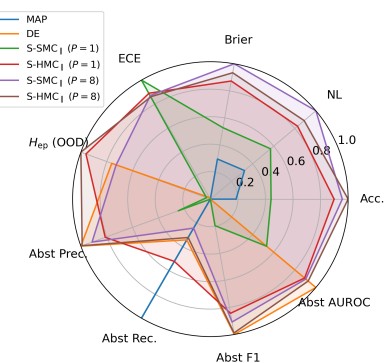

Figure 25: Summary metrics for IMDb in all methods. S-SMC$_\parallel$ ($P = 1$ chain with $N = 10$), S-HMC$_\parallel$ ($NP$ chains), DE ($N$ models) and MAP, with fixed number of leapfrog $L = 1$, $B = 26$, $M = 2$, $v = 1$ and $s = 0.35$ (5 realizations).

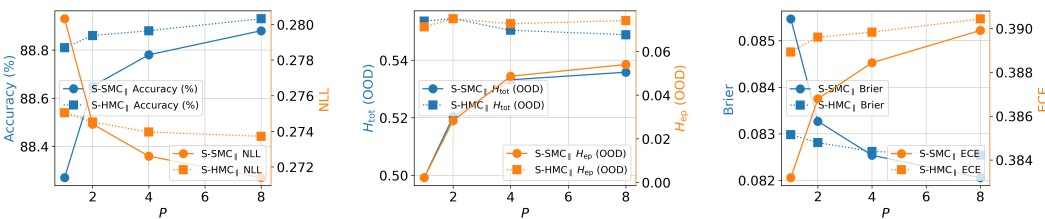

Figure 26: Comparison of S-SMC$_\parallel$ ($P$ chains with $N = 10$) and S-HMC$_\parallel$ ($NP$ chains), with fixed number of leapfrog $L = 1$, $B = 26$, $M = 1$, $v = 1$ and $s = 0.35$, on IMDb (5 realizations and $\pm$s.e. in accuracy ).

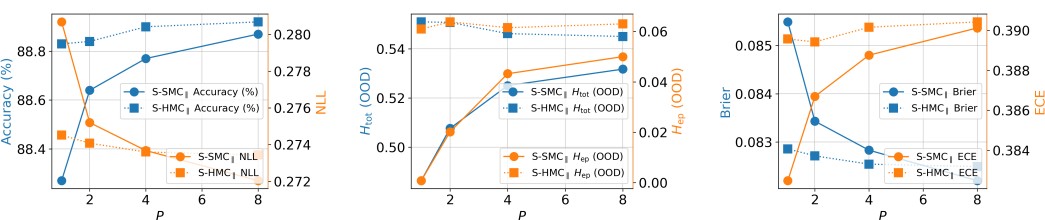

Figure 27: Comparison of S-SMC$_\parallel$ ($P$ chains with $N = 10$) and S-HMC$_\parallel$ ($NP$ chains), with fixed number of leapfrog $L = 1$, $B = 26$, $M = 2$, $v = 1$ and $s = 0.25$, on IMDb (5 realizations and $\pm$ s.e. in accuracy).

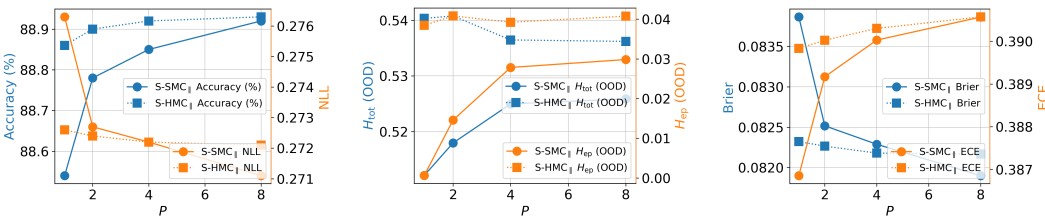

Figure 28: Comparison of S-SMC$_\parallel$ ($P$ chains with $N = 10$) and S-HMC$_\parallel$ ($NP$ chains), with fixed number of leapfrog $L = 1$, $B = 26$, $M = 2$, $v = 1$ and $s = 0.1$, on IMDb (5 realizations).

## H.3 CIFAR10

Experiments in this section are tested on the CIFAR10 dataset with the model setting stated in Appendix D.3.3.

The summary metrics on CIFAR10 are shown in a spider-plot in Figure 3. Table 20 shows the performance as the tuning parameter $s$ vary. Figure 29, 30 and 31 give the full convergence of SBMC$_\parallel$ with increasing $P$. The corresponding full data results are given in Table 30 , 31 and 32.

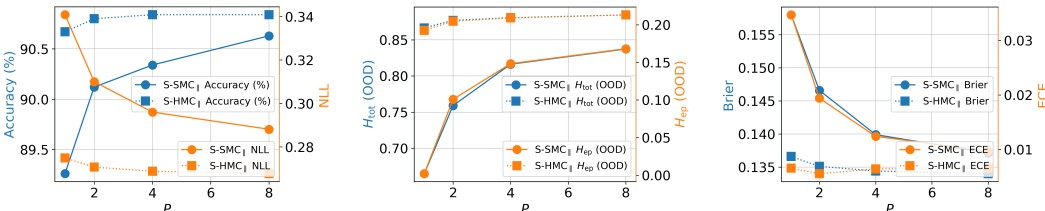

Figure 29: Comparison of S-SMC$_\parallel$ ($P$ chains with $N = 10$) and S-HMC$_\parallel$ ($NP$ chains), with fixed number of leapfrog $L = 1$, $B = 200$, $M = 4$, $v = 0.2$ and $s = 0.05$, on CIFAR10 (5 realizations).

Table 20: Comparison of S-SMC$_\parallel$ ($N = 10$), S-HMC$_\parallel$ ($N$ chains), DE ($N$) and MAP, with fixed number of leapfrog $L = 1$, $B = 200$, $M = 4$ and $v = 0.2$, on CIFAR10 (5 realizations and $\pm$ s.e. in accuracy).

| $s$ | Method | Ep. | Acc. | NLL | Brier | ECE |
|---|---|---|---|---|---|---|
| 0.2 | S-SMC$_\parallel$ | 289.6 | $86.99 \pm 0.08$ | 4.710e-1 | 2.007e-1 | 6.462e-2 |
| | $P = 8$ | 289.3 | $90.30 \pm 0.03$ | 3.217e-1 | 1.445e-1 | 1.180e-2 |
| | S-HMC$_\parallel$ | 200 | $90.23 \pm 0.08$ | 2.990e-1 | 1.466e-1 | 2.518e-2 |
| | $P = 8$ | 200 | $90.82 \pm 0.03$ | 2.810e-1 | 1.395e-1 | 3.481e-2 |
| 0.1 | S-SMC$_\parallel$ | 229.6 | $88.26 \pm 0.07$ | 3.855e-1 | 1.770e-1 | 4.593e-2 |
| | $P = 8$ | 225.3 | $90.45 \pm 0.06$ | 2.980e-1 | 1.400e-1 | 7.737e-3 |
| | S-HMC$_\parallel$ | 200 | $90.57 \pm 0.04$ | 2.823e-1 | 1.398e-1 | 1.073e-2 |
| | $P = 8$ | 200 | $90.83 \pm 0.03$ | 2.701e-1 | 1.353e-1 | 1.517e-2 |
| 0.05 | S-SMC$_\parallel$ | 168.8 | $89.26 \pm 0.07$ | 3.408e-1 | 1.580e-1 | 3.470e-2 |
| | $P = 8$ | 174.3 | $90.63 \pm 0.05$ | 2.881e-1 | 1.371e-1 | 9.720e-3 |
| | S-HMC$_\parallel$ | 200 | $90.67 \pm 0.03$ | 2.749e-1 | 1.366e-1 | 6.598e-3 |
| | $P = 8$ | 200 | $90.84 \pm 0.03$ | 2.677e-1 | 1.340e-1 | 6.601e-3 |
| 0 | MAP | 200 | $90.39\pm0.07$ | 2.913e-1 | 1.420e-1 | 2.502e-2 |
| | DE ($N$) | 200 | $90.81\pm0.03$ | 2.741e-1 | 1.355e-1 | 1.770e-2 |

| $s$ | Method | $H_{ep}$ | | | | | $H_{tot}$ | | | | |
|---|---|---|---|---|---|---|---|---|---|---|---|
| | | ID | | OOD | | | ID | | OOD | | |
| | | cor. | inc. | close | corrupt | far | cor. | inc. | close | corrupt | far |
| 0.2 | S-SMC$_\parallel$ | 3.682e-4 | 1.947e-3 | 2.063e-3 | 1.092e-3 | 1.629e-3 | 1.136e-1 | 5.613e-1 | 5.440e-1 | 3.630e-1 | 7.756e-1 |
| | $P = 8$ | 8.362e-2 | 3.326e-1 | 4.244e-1 | 2.361e-1 | 4.065e-1 | 1.071e-1 | 5.954e-1 | 9.675e-1 | 6.080e-1 | 1.160e+0 |
| | S-HMC$_\parallel$ | 1.159e-1 | 4.091e-1 | 5.195e-1 | 2.993e-1 | 5.333e-1 | 2.676e-1 | 9.231e-1 | 1.059e+0 | 6.768e-1 | 1.297e+0 |
| | $P = 8$ | 1.405e-1 | 4.604e-1 | 6.042e-1 | 3.476e-1 | 6.129e-1 | 2.945e-1 | 9.687e-1 | 1.146e+0 | 7.256e-1 | 1.364e+0 |
| 0.1 | S-SMC$_\parallel$ | 2.948e-4 | 1.507e-3 | 1.603e-3 | 8.256e-4 | 1.450e-3 | 1.309e-1 | 6.364e-1 | 6.121e-1 | 4.027e-1 | 9.110e-1 |
| | $P = 8$ | 5.539e-2 | 2.369e-1 | 3.054e-1 | 1.636e-1 | 2.937e-1 | 1.217e-1 | 6.453e-1 | 9.184e-1 | 5.690e-1 | 1.156e+0 |
| | S-HMC$_\parallel$ | 7.055e-2 | 2.795e-1 | 3.591e-1 | 1.972e-1 | 3.581e-1 | 2.241e-1 | 8.596e-1 | 9.703e-1 | 6.102e-1 | 1.219e+0 |
| | $P = 8$ | 1.035e-1 | 3.121e-1 | 4.095e-1 | 2.244e-1 | 4.031e-1 | 2.367e-1 | 8.884e-1 | 1.023e+0 | 6.371e-1 | 1.257e+0 |
| 0.05 | S-SMC$_\parallel$ | 4.258e-4 | 2.273e-3 | 2.515e-3 | 1.297e-3 | 2.185e-3 | 1.351e-1 | 6.620e-1 | 6.639e-1 | 4.300e-1 | 9.008e-1 |
| | $P = 8$ | 3.507e-2 | 1.584e-1 | 2.027e-1 | 1.058e-1 | 1.961e-1 | 1.311e-1 | 6.684e-1 | 8.643e-1 | 5.368e-1 | 1.111e+0 |
| | S-HMC$_\parallel$ | 4.060e-2 | 1.804e-1 | 2.308e-1 | 1.228e-1 | 2.243e-1 | 1.917e-1 | 8.073e-1 | 8.900e-1 | 5.558e-1 | 1.154e+0 |
| | $P = 8$ | 4.579e-2 | 1.966e-1 | 2.564e-1 | 1.358e-1 | 2.472e-1 | 1.971e-1 | 8.203e-1 | 9.173e-1 | 5.684e-1 | 1.168e+0 |
| 0 | MAP | 0 | 0 | 0 | 0 | 0 | 1.423e-1 | 7.037e-1 | 7.258e-1 | 4.577e-1 | 1.065e+0 |
| | DE ($N$) | 9.291e-3 | 4.753e-2 | 4.861e-2 | 2.603e-2 | 3.930e-2 | 1.541e-1 | 7.275e-1 | 7.629e-1 | 4.786e-1 | 1.029e+0 |

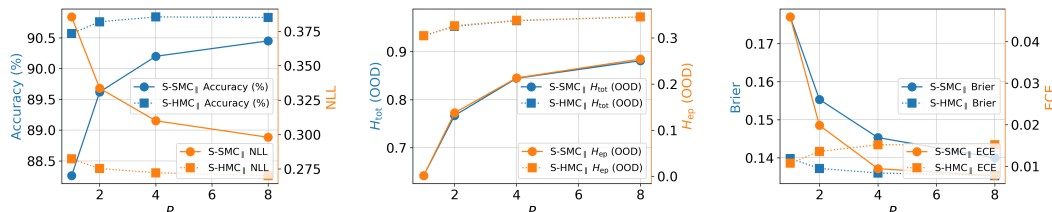

Figure 30: Comparison of S-SMC$_\parallel$ ($P$ chains with $N = 10$) and S-HMC$_\parallel$ ($NP$ chains), with fixed number of leapfrog $L = 1$, $B = 200$, $M = 4$, $v = 0.2$ and $s = 0.1$, on CIFAR10 (5 realizations).

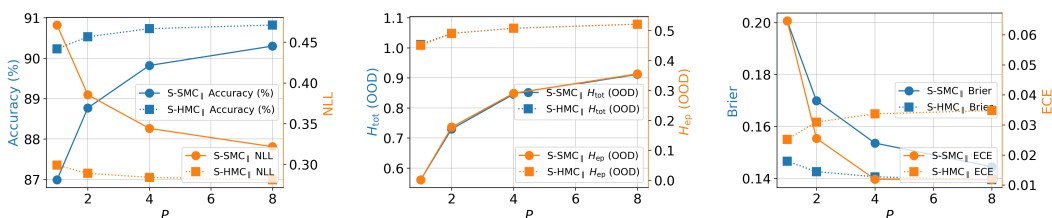

Figure 31: Comparison of S-SMC$_\parallel$ ($P$ chains with $N = 10$) and S-HMC$_\parallel$ ($NP$ chains), with fixed number of leapfrog $L = 1$, $B = 200$, $M = 4$, $v = 0.2$ and $s = 0.2$, on CIFAR10 (5 realizations).

# I  ALL-INCLUSIVE DATA TABLES

Table 21: Comparison in all domains among S-SMC$_\parallel$ ($P = 1$ chain with $N = 10$), S-HMC$_\parallel$ ($NP$ chains), HMC (GS) ($2e4$ samples), DE ($N$ models) and MAP, with fixed number of leapfrog $L = 1$, $v = 0.1$ and $s = 0.1$, on MNIST7 (5 realizations and $\pm$ s.e. in entropy).

| Group | MAP | DE | | | S-HMC ($s = 0.1$) | | | S-SMC ($s = 0.1$) | | | HMC (GS) ($s = 1$) | | |
| --- | --- | --- | --- | --- | --- | --- | --- | --- | --- | --- | --- | --- | --- |
| | $H_{tot}$ | $H_{tot}$ | $H_{al}$ | $H_{ep}$ | $H_{tot}$ | $H_{al}$ | $H_{ep}$ | $H_{tot}$ | $H_{al}$ | $H_{ep}$ | $H_{tot}$ | $H_{al}$ | $H_{ep}$ |
| Digit 0 | 1.276e-1 | 1.307e-1 | 1.237e-1 | 7.076e-3 | 1.671e-1 | 1.228e-1 | 4.427e-2 | 1.110e-1 | 8.528e-2 | 2.574e-2 | 2.157e-1 | 1.268e-1 | 8.893e-2 |
| Digit 1 | 1.768e-1 | 1.840e-1 | 1.781e-1 | 5.823e-3 | 1.889e-1 | 1.585e-1 | 3.038e-2 | 1.408e-1 | 1.255e-1 | 1.535e-2 | 2.124e-1 | 1.621e-1 | 5.025e-2 |
| Digit 2 | 2.294e-1 | 2.266e-1 | 2.142e-1 | 1.236e-2 | 2.980e-1 | 2.168e-1 | 8.122e-2 | 2.090e-1 | 1.679e-1 | 4.118e-2 | 3.410e-1 | 1.988e-1 | 1.423e-1 |
| Digit 3 | 3.168e-1 | 3.493e-1 | 3.222e-1 | 2.711e-2 | 3.883e-1 | 2.896e-1 | 9.873e-2 | 2.686e-1 | 2.245e-1 | 4.404e-2 | 4.229e-1 | 2.616e-1 | 1.613e-1 |
| Digit 4 | 2.158e-1 | 2.221e-1 | 2.103e-1 | 1.182e-2 | 2.753e-1 | 2.095e-1 | 6.583e-2 | 1.981e-1 | 1.654e-1 | 3.272e-2 | 2.925e-1 | 1.847e-1 | 1.078e-1 |
| Digit 5 | 3.993e-1 | 4.058e-1 | 3.787e-1 | 2.712e-2 | 4.395e-1 | 3.224e-1 | 1.171e-1 | 3.056e-1 | 2.520e-1 | 5.366e-2 | 4.428e-1 | 2.655e-1 | 1.773e-1 |
| Digit 6 | 1.856e-1 | 2.045e-1 | 1.927e-1 | 1.180e-2 | 2.836e-1 | 2.047e-1 | 7.891e-2 | 1.856e-1 | 1.495e-1 | 3.605e-2 | 2.968e-1 | 1.785e-1 | 1.182e-1 |
| Digit 7 | 1.897e-1 | 1.957e-1 | 1.859e-1 | 9.730e-3 | 2.403e-1 | 1.802e-1 | 6.008e-2 | 1.693e-1 | 1.396e-1 | 2.967e-2 | 2.528e-1 | 1.569e-1 | 9.589e-2 |
| Digit 8 | 9.507e-1 | 9.821e-1 | 9.078e-1 | 7.433e-2 | 1.091e+0 | 7.836e-1 | 3.072e-1 | 8.975e-1 | 7.591e-1 | 1.384e-1 | 1.121e+0 | 6.333e-1 | 4.873e-1 |
| Digit 9 | 6.157e-1 | 6.393e-1 | 6.046e-1 | 3.468e-2 | 7.567e-1 | 5.626e-1 | 1.941e-1 | 6.591e-1 | 5.651e-1 | 9.406e-2 | 9.210e-1 | 5.554e-1 | 3.657e-1 |
| Perturbed | 7.528e-1 | 8.112e-1 | 7.006e-1 | 1.106e-1 | 8.443e-1 | 4.227e-1 | 4.216e-1 | 8.001e-1 | 6.169e-1 | 1.832e-1 | 1.228e+0 | 2.819e-1 | 9.466e-1 |
| White Noise | 7.703e-1 | 9.806e-1 | 7.117e-1 | 2.690e-1 | 1.453e+0 | 5.221e-1 | 9.304e-1 | 9.744e-1 | 5.800e-1 | 3.944e-1 | 1.398e+0 | 3.444e-1 | 1.053e+0 |
| All ID | 2.301e-1 | 2.398e-1 | 2.069e-1 | 3.291e-2 | 2.821e-1 | 2.111e-1 | 7.090e-2 | 1.966e-1 | 1.623e-1 | 3.429e-2 | 1.021e+0 | 5.943e-1 | 4.265e-1 |

Table 22: Comparison in all domains among S-SMC$_\parallel$ ($P = 1$ chain with $N = 10$), S-HMC$_\parallel$ ($NP$ chains), DE ($N$ models) and MAP, with fixed number of leapfrog $L = 1$, $B = 25$, $M = 1$, $v = 0.025$ and $s = 0.1$, on IMDb (5 realizations).

| Group | MAP | DE | | | S-HMC$_\parallel$ | | | S-SMC$_\parallel$ | | |
|---|---|---|---|---|---|---|---|---|---|---|
| | $H_{\text{tot}}$ | $H_{\text{tot}}$ | $H_{\text{al}}$ | $H_{\text{ep}}$ | $H_{\text{tot}}$ | $H_{\text{al}}$ | $H_{\text{ep}}$ | $H_{\text{tot}}$ | $H_{\text{al}}$ | $H_{\text{ep}}$ |
| Negative | 4.352e-1 | 4.407e-1 | 4.406e-1 | 9.314e-5 | 4.892e-1 | 4.889e-1 | 2.482e-4 | 4.929e-1 | 4.927e-1 | 1.148e-4 |
| Positive | 5.716e-1 | 5.688e-1 | 5.687e-1 | 1.188e-4 | 5.031e-1 | 5.029e-1 | 2.659e-4 | 5.072e-1 | 5.071e-1 | 1.240e-4 |
| Meta | 6.117e-1 | 6.098e-1 | 6.097e-1 | 5.026e-5 | 5.331e-1 | 5.326e-1 | 4.633e-4 | 5.463e-1 | 5.461e-1 | 2.200e-4 |
| Full Meta | 6.658e-1 | 6.649e-1 | 6.649e-1 | 5.548e-5 | 6.185e-1 | 6.180e-1 | 5.410e-4 | 6.260e-1 | 6.257e-1 | 3.212e-4 |
| Reviews | 5.814e-1 | 5.793e-1 | 5.793e-1 | 5.064e-5 | 5.165e-1 | 5.160e-1 | 4.598e-4 | 5.251e-1 | 5.249e-1 | 1.792e-4 |
| Full reviews | 6.705e-1 | 6.702e-1 | 6.701e-1 | 6.302e-5 | 6.400e-1 | 6.395e-1 | 5.057e-4 | 6.457e-1 | 6.454e-1 | 3.285e-4 |
| Lipsum | 5.894e-1 | 5.882e-1 | 5.881e-1 | 4.909e-5 | 5.079e-1 | 5.074e-1 | 4.260e-4 | 5.142e-1 | 5.140e-1 | 1.749e-4 |
| All ID | 5.034e-1 | 5.048e-1 | 5.046e-1 | 1.060e-4 | 4.962e-1 | 4.959e-1 | 2.570e-4 | 5.000e-1 | 4.999e-1 | 1.194e-4 |

Table 23: Comparison in all domains among S-SMC$_\parallel$ ($P = 8$ chain with $N = 10$), S-HMC$_\parallel$ ($NP$ chains), DE ($N$ models) and MAP, with fixed number of leapfrog $L = 1$, $B = 26$, $M = 2$, $v = 1$ and $s = 0.35$, on IMDb (5 realizations).

| Group | MAP | DE | | | S-HMC$_\parallel$ | | | S-SMC$_\parallel$ | | |
|---|---|---|---|---|---|---|---|---|---|---|
| | $H_{\text{tot}}$ | $H_{\text{tot}}$ | $H_{\text{al}}$ | $H_{\text{ep}}$ | $H_{\text{tot}}$ | $H_{\text{al}}$ | $H_{\text{ep}}$ | $H_{\text{tot}}$ | $H_{\text{al}}$ | $H_{\text{ep}}$ |
| Negative | 2.489e-1 | 2.494e-1 | 2.483e-1 | 1.033e-3 | 3.107e-1 | 2.928e-1 | 1.794e-2 | 2.962e-1 | 2.860e-1 | 1.016e-2 |
| Positive | 3.629e-1 | 3.675e-1 | 3.659e-1 | 1.631e-3 | 3.296e-1 | 3.099e-1 | 1.971e-2 | 3.160e-1 | 3.048e-1 | 1.126e-2 |
| Meta | 5.598e-1 | 5.767e-1 | 5.507e-1 | 2.594e-2 | 5.626e-1 | 4.887e-1 | 7.391e-2 | 5.555e-1 | 5.004e-1 | 5.514e-2 |
| Full Meta | 5.561e-1 | 5.737e-1 | 5.469e-1 | 2.683e-2 | 6.088e-1 | 5.251e-1 | 8.360e-2 | 5.920e-1 | 5.277e-1 | 6.435e-2 |
| Reviews | 4.071e-1 | 4.251e-1 | 4.030e-1 | 2.212e-2 | 4.188e-1 | 3.657e-1 | 5.315e-2 | 3.987e-1 | 3.611e-1 | 3.756e-2 |
| Full reviews | 5.695e-1 | 6.007e-1 | 5.574e-1 | 4.331e-2 | 6.156e-1 | 5.246e-1 | 9.098e-2 | 6.025e-1 | 5.420e-1 | 6.049e-2 |
| Lipsum | 5.462e-1 | 5.556e-1 | 5.283e-1 | 2.733e-2 | 5.386e-1 | 4.695e-1 | 6.917e-2 | 5.304e-1 | 4.778e-1 | 5.260e-2 |
| All ID | 3.059e-1 | 3.084e-1 | 3.071e-1 | 1.332e-3 | 3.202e-1 | 3.013e-1 | 1.883e-2 | 3.061e-1 | 2.954e-1 | 1.071e-2 |

Table 26: Comparison of S-SMC$_\parallel$ ($P$ chains with $N = 10$) and S-HMC$_\parallel$ ($NP$ chains), with fixed $L = 1$, $B = 25$, $M = 1$, $v = 0.025$, $s = 0.1$, on IMDb (5 realizations and $\pm$ s.e. in accuracy).

| $P$ | Method | Ep. | Acc. | NLL | $H_{\text{ep}}$ | | | | | | |
|---|---|---|---|---|---|---|---|---|---|---|---|
| | | | | | ID | | OOD | | | | |
| | | | | | cor. | inc. | reviews | meta | lipsum | full reviews | full meta |
| 1 | S-SMC$_\parallel$ | 18.60 | 86.70±0.03 | 3.655e-1 | 1.122e-4 | 1.664e-4 | 1.792e-4 | 2.200e-4 | 1.749e-4 | 3.285e-4 | 3.212e-4 |
| 1 | S-HMC$_\parallel$ | 25 | 86.70±0.01 | 3.634e-1 | 2.418e-4 | 3.565e-4 | 4.598e-4 | 4.633e-4 | 4.260e-4 | 5.057e-4 | 5.410e-4 |
| 2 | S-SMC$_\parallel$ | 18.70 | 86.72±0.03 | 3.656e-1 | 1.936e-4 | 2.798e-4 | 3.061e-4 | 3.358e-4 | 4.211e-4 | 4.491e-4 | 4.465e-4 |
| 2 | S-HMC$_\parallel$ | 25 | 86.69±0.01 | 3.634e-1 | 2.697e-4 | 3.955e-4 | 5.604e-4 | 5.852e-4 | 4.916e-4 | 6.022e-4 | 6.819e-4 |
| 4 | S-SMC$_\parallel$ | 19.10 | 86.68±0.02 | 3.654e-1 | 2.370e-4 | 3.452e-4 | 4.433e-4 | 4.874e-4 | 5.515e-4 | 5.848e-4 | 6.201e-4 |
| 4 | S-HMC$_\parallel$ | 25 | 86.72±0.01 | 3.635e-1 | 2.776e-4 | 4.042e-4 | 5.940e-4 | 6.264e-4 | 5.629e-4 | 7.074e-4 | 7.288e-4 |
| 8 | S-SMC$_\parallel$ | 19.15 | 86.69±0.01 | 3.653e-1 | 2.531e-4 | 3.697e-4 | 4.744e-4 | 4.971e-4 | 4.862e-4 | 5.876e-4 | 6.105e-4 |
| 8 | S-HMC$_\parallel$ | 25 | 86.72±0.00 | 3.633e-1 | 2.766e-4 | 4.022e-4 | 5.694e-4 | 6.062e-4 | 5.637e-4 | 7.438e-4 | 7.051e-4 |

| $P$ | Method | Brier | ECE | $H_{\text{tot}}$ | | | | | | |
|---|---|---|---|---|---|---|---|---|---|---|
| | | | | ID | | OOD | | | | |
| | | | | cor. | inc. | reviews | meta | lipsum | full reviews | full meta |
| 1 | S-SMC$_\parallel$ | 1.093e-1 | 3.699e-1 | 4.792e-1 | 6.357e-1 | 5.251e-1 | 5.463e-1 | 5.142e-1 | 6.457e-1 | 6.261e-1 |
| 1 | S-HMC$_\parallel$ | 1.086e-1 | 3.694e-1 | 4.750e-1 | 6.340e-1 | 5.165e-1 | 5.331e-1 | 5.079e-1 | 6.400e-1 | 6.186e-1 |
| 2 | S-SMC$_\parallel$ | 1.093e-1 | 3.702e-1 | 4.793e-1 | 6.356e-1 | 5.243e-1 | 5.466e-1 | 5.131e-1 | 6.462e-1 | 6.270e-1 |
| 2 | S-HMC$_\parallel$ | 1.086e-1 | 3.695e-1 | 4.752e-1 | 6.342e-1 | 5.172e-1 | 5.342e-1 | 5.092e-1 | 6.409e-1 | 6.196e-1 |
| 4 | S-SMC$_\parallel$ | 1.092e-1 | 3.699e-1 | 4.787e-1 | 6.355e-1 | 5.230e-1 | 5.440e-1 | 5.127e-1 | 6.459e-1 | 6.259e-1 |
| 4 | S-HMC$_\parallel$ | 1.086e-1 | 3.699e-1 | 4.754e-1 | 6.342e-1 | 5.180e-1 | 5.351e-1 | 5.085e-1 | 6.419e-1 | 6.203e-1 |
| 8 | S-SMC$_\parallel$ | 1.092e-1 | 3.698e-1 | 4.788e-1 | 6.355e-1 | 5.213e-1 | 5.406e-1 | 5.115e-1 | 6.430e-1 | 6.236e-1 |
| 8 | S-HMC$_\parallel$ | 1.086e-1 | 3.701e-1 | 4.752e-1 | 6.341e-1 | 5.184e-1 | 5.359e-1 | 5.085e-1 | 6.426e-1 | 6.209e-1 |

Table 24: Comparison of S-SMC$_\parallel$ ($P$ chains with $N = 10$) and S-HMC$_\parallel$ ($NP$ chains), with fixed number of leapfrog $L = 1$, $B = 160$, $M = 7$, $v = 0.1$ and $s = 0.25$, on MNIST7 (5 realizations and $\pm$ s.e. in accuracy).

| $P$ | Method | Ep. | Acc. | NLL | Brier |
|---|---|---|---|---|---|
| 1 | S-SMC$_\parallel$ | 166.6 | 90.35±0.26 | 3.300e-1 | 1.441e-1 |
| 1 | S-HMC$_\parallel$ | 160 | 92.79±0.19 | 2.571e-1 | 1.156e-1 |
| 2 | S-SMC$_\parallel$ | 160.3 | 92.00±0.24 | 2.726e-1 | 1.239e-1 |
| 2 | S-HMC$_\parallel$ | 160 | 92.97±0.14 | 2.536e-1 | 1.140e-1 |
| 4 | S-SMC$_\parallel$ | 164.5 | 92.59±0.15 | 2.504e-1 | 1.152e-1 |
| 4 | S-HMC$_\parallel$ | 160 | 93.13±0.07 | 2.506e-1 | 1.133e-1 |
| 8 | S-SMC$_\parallel$ | 161.5 | 93.00±0.11 | 2.366e-1 | 1.096e-1 |
| 8 | S-HMC$_\parallel$ | 160 | 93.15±0.05 | 2.490e-1 | 1.127e-1 |
|  | HMC (GS) | 2e4 | 92.87±0.48 | 2.376e-1 | 1.079e-1 |

| $P$ | Method | $H_{\text{ep}}$ | | | | | | $H_{\text{tot}}$ | | | | | |
|---|---|---|---|---|---|---|---|---|---|---|---|---|---|
|  |  | ID | | OOD | | | | ID | | OOD | | | |
|  |  | cor. | inc. | 8 | 9 | wn | per. | cor. | inc. | 8 | 9 | wn | per. |
| 1 | S-SMC$_\parallel$ | 2.257e-2 | 1.094e-1 | 1.146e-1 | 7.791e-2 | 3.847e-1 | 1.914e-1 | 1.508e-1 | 6.679e-1 | 8.384e-1 | 5.945e-1 | 8.290e-1 | 7.110e-1 |
| 1 | S-HMC$_\parallel$ | 1.133e-1 | 4.232e-1 | 4.985e-1 | 3.225e-1 | 1.281e+0 | 6.314e-1 | 3.149e-1 | 1.026e+0 | 1.220e+0 | 8.606e-1 | 1.614e+0 | 1.019e+0 |
| 2 | S-SMC$_\parallel$ | 5.445e-2 | 2.442e-1 | 1.975e-1 | 1.402e-1 | 5.073e-1 | 2.923e-1 | 1.371e-1 | 6.768e-1 | 9.108e-1 | 6.629e-1 | 9.559e-1 | 8.111e-1 |
| 2 | HMC$_\parallel$ | 1.298e-1 | 4.568e-1 | 5.380e-1 | 3.638e-1 | 1.321e+0 | 6.720e-1 | 3.358e-1 | 1.050e+0 | 1.250e+0 | 8.975e-1 | 1.680e+0 | 1.050e+0 |
| 4 | S-SMC$_\parallel$ | 7.492e-2 | 3.240e-1 | 2.544e-1 | 1.797e-1 | 6.264e-1 | 3.749e-1 | 1.287e-1 | 6.669e-1 | 9.591e-1 | 7.064e-1 | 1.051e+0 | 8.923e-1 |
| 4 | HMC$_\parallel$ | 1.356e-1 | 4.718e-1 | 5.508e-1 | 3.634e-1 | 1.327e+0 | 7.084e-1 | 3.443e-1 | 1.065e+0 | 1.261e+0 | 9.031e-1 | 1.689e+0 | 1.078e+0 |
| 8 | S-SMC$_\parallel$ | 8.828e-2 | 3.717e-1 | 2.984e-1 | 2.089e-1 | 7.384e-1 | 4.475e-1 | 1.247e-1 | 6.641e-1 | 1.001e+0 | 7.354e-1 | 1.152e+0 | 9.627e-1 |
| 8 | S-HMC$_\parallel$ | 1.384e-1 | 4.788e-1 | 5.572e-1 | 3.678e-1 | 1.349e+0 | 7.235e-1 | 3.456e-1 | 1.070e+0 | 1.267e+0 | 9.025e-1 | 1.715e+0 | 1.094e+0 |
|  | HMC (GS) | 7.199e-2 | 3.432e-1 | 3.887e-1 | 2.748e-1 | 1.169e+0 | 5.579e-1 | 2.045e-1 | 8.566e-1 | 9.984e-1 | 7.425e-1 | 1.574e+0 | 8.725e-1 |

Table 25: Comparison of S-SMC$_\parallel$ ($P$ chains with $N = 10$) and S-HMC$_\parallel$ ($NP$ chains), with fixed number of leapfrog $L = 1$, $B = 160$, $M = 10$, $v = 0.1$ and $s = 0.1$, on MNIST7 (5 realizations and $\pm$ s.e. in accuracy).

| $P$ | Method | Ep. | Acc. | NLL | Brier |
|---|---|---|---|---|---|
| 1 | S-SMC$_\parallel$ | 170.0 | 92.17±0.37 | 2.671e-1 | 1.186e-1 |
| 1 | S-HMC$_\parallel$ | 160 | 92.96±0.17 | 2.326e-1 | 1.071e-1 |
| 2 | S-SMC$_\parallel$ | 179.0 | 92.52±0.30 | 2.507e-1 | 1.127e-1 |
| 2 | S-HMC$_\parallel$ | 160 | 93.10±0.12 | 2.310e-1 | 1.067e-1 |
| 4 | S-SMC$_\parallel$ | 180.5 | 93.01±0.29 | 2.369e-1 | 1.069e-1 |
| 4 | HMC$_\parallel$ | 160 | 93.12±0.09 | 2.306e-1 | 1.069e-1 |
| 8 | S-SMC$_\parallel$ | 178.0 | 93.26±0.16 | 2.259e-1 | 1.025e-1 |
| 8 | S-HMC$_\parallel$ | 160 | 93.12±0.09 | 2.310e-1 | 1.072e-1 |
|  | HMC (GS) | 2e4 | 92.92±0.41 | 2.366e-1 | 1.084e-1 |

| $P$ | Method | $H_{\text{ep}}$ | | | | | | $H_{\text{tot}}$ | | | | | |
|---|---|---|---|---|---|---|---|---|---|---|---|---|---|
|  |  | ID | | OOD | | | | ID | | OOD | | | |
|  |  | cor. | inc. | 8 | 9 | wn | per. | cor. | inc. | 8 | 9 | wn | per. |
| 1 | S-SMC$_\parallel$ | 2.642e-2 | 1.288e-1 | 1.384e-1 | 9.406e-2 | 3.972e-1 | 1.776e-1 | 1.536e-1 | 7.042e-1 | 8.975e-1 | 6.591e-1 | 9.753e-1 | 7.859e-1 |
| 1 | S-HMC$_\parallel$ | 5.624e-2 | 2.645e-1 | 3.072e-1 | 1.941e-1 | 9.259e-1 | 4.100e-1 | 2.343e-1 | 9.132e-1 | 1.091e+0 | 7.567e-1 | 1.443e+0 | 8.248e-1 |
| 2 | S-SMC$_\parallel$ | 4.233e-2 | 2.042e-1 | 1.836e-1 | 1.227e-1 | 5.306e-1 | 2.620e-1 | 1.449e-1 | 7.053e-1 | 9.422e-1 | 6.870e-1 | 1.058e+0 | 8.577e-1 |
| 2 | S-HMC$_\parallel$ | 6.494e-2 | 2.844e-1 | 3.395e-1 | 2.198e-1 | 9.977e-1 | 4.432e-1 | 2.472e-1 | 9.223e-1 | 1.112e+0 | 7.769e-1 | 1.506e+0 | 8.436e-1 |
| 4 | S-SMC$_\parallel$ | 5.315e-2 | 2.471e-1 | 2.200e-1 | 1.465e-1 | 6.363e-1 | 3.288e-1 | 1.403e-1 | 7.019e-1 | 9.791e-1 | 7.130e-1 | 1.126e+0 | 9.159e-1 |
| 4 | S-HMC$_\parallel$ | 6.733e-2 | 2.924e-1 | 3.479e-1 | 2.179e-1 | 1.010e+0 | 4.740e-1 | 2.520e-1 | 9.300e-1 | 1.119e+0 | 7.823e-1 | 1.493e+0 | 8.866e-1 |
| 8 | S-SMC$_\parallel$ | 5.872e-2 | 2.725e-1 | 2.440e-1 | 1.637e-1 | 7.309e-1 | 3.750e-1 | 1.374e-1 | 7.075e-1 | 1.001e+0 | 7.307e-1 | 1.220e+0 | 9.649e-1 |
| 8 | S-HMC$_\parallel$ | 6.982e-2 | 2.993e-1 | 3.524e-1 | 2.258e-1 | 1.066e+0 | 4.817e-1 | 2.553e-1 | 9.380e-1 | 1.127e+0 | 7.893e-1 | 1.539e+0 | 8.946e-1 |
|  | HMC (GS) | 5.034e-2 | 2.417e-1 | 2.713e-1 | 1.900e-1 | 9.303e-1 | 3.683e-1 | 2.076e-1 | 8.477e-1 | 1.001e+0 | 7.264e-1 | 1.430e+0 | 7.518e-1 |

Table 27: Comparison of S-SMC$_\parallel$ ($P$ chains with $N = 10$) and S-HMC$_\parallel$ ($NP$ chains), with fixed number of leapfrog $L = 1$, $B = 26$, $M = 1$, $v = 1$ and $s = 0.35$, on IMDb (5 realizations and $\pm$s.e. in accuracy ).

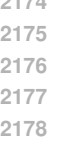

| $P$ | Method | Ep. | Acc. | NLL | $H_{\text{ep}}$ ID | | $H_{\text{ep}}$ OOD | | | | |
|---|---|---|---|---|---|---|---|---|---|---|---|
| | | | | | cor. | inc. | reviews | meta | lipsum | full reviews | full meta |
| 1 | S-SMC$_\parallel$ | 27.40 | 88.27±0.07 | 2.803e-1 | 6.177e-4 | 1.460e-3 | 1.581e-3 | 2.495e-3 | 2.187e-3 | 2.279e-3 | 2.504e-3 |
| 1 | S-HMC$_\parallel$ | 26.00 | 88.81±0.01 | 2.750e-1 | 1.565e-2 | 3.463e-2 | 5.414e-2 | 7.021e-2 | 6.872e-2 | 8.407e-2 | 7.930e-2 |
| 2 | S-SMC$_\parallel$ | 27.70 | 88.65±0.04 | 2.744e-1 | 5.513e-3 | 1.323e-2 | 2.132e-2 | 3.443e-2 | 2.751e-2 | 2.232e-2 | 3.585e-2 |
| 2 | S-HMC$_\parallel$ | 26.00 | 88.86±0.03 | 2.745e-1 | 1.620e-2 | 3.584e-2 | 5.510e-2 | 7.622e-2 | 7.118e-2 | 8.766e-2 | 8.490e-2 |
| 4 | S-SMC$_\parallel$ | 28.55 | 88.78±0.03 | 2.726e-1 | 8.040e-3 | 1.881e-2 | 3.057e-2 | 4.854e-2 | 5.097e-2 | 5.547e-2 | 5.766e-2 |
| 4 | S-HMC$_\parallel$ | 26.00 | 88.88±0.01 | 2.740e-1 | 1.640e-2 | 3.610e-2 | 5.388e-2 | 7.488e-2 | 6.705e-2 | 8.588e-2 | 8.184e-2 |
| 8 | S-SMC$_\parallel$ | 27.63 | 88.88±0.03 | 2.714e-1 | 9.342e-3 | 2.164e-2 | 3.756e-2 | 5.515e-2 | 5.260e-2 | 6.049e-2 | 6.435e-2 |
| 8 | S-HMC$_\parallel$ | 26.00 | 88.93±0.02 | 2.737e-1 | 1.662e-2 | 3.651e-2 | 5.315e-2 | 7.391e-2 | 6.917e-2 | 9.098e-2 | 8.360e-2 |

| $P$ | Method | Brier | ECE | $H_{\text{tot}}$ ID | | $H_{\text{tot}}$ OOD | | | | |
|---|---|---|---|---|---|---|---|---|---|---|
| | | | | cor. | inc. | reviews | meta | lipsum | full reviews | full meta |
| 1 | S-SMC$_\parallel$ | 8.547e-2 | 3.832e-1 | 2.643e-1 | 5.482e-1 | 3.802e-1 | 5.116e-1 | 5.172e-1 | 5.581e-1 | 5.286e-1 |
| 1 | S-HMC$_\parallel$ | 8.298e-2 | 3.889e-1 | 2.890e-1 | 5.681e-1 | 4.289e-1 | 5.583e-1 | 5.556e-1 | 6.133e-1 | 6.120e-1 |
| 2 | S-SMC$_\parallel$ | 8.327e-2 | 3.868e-1 | 2.694e-1 | 5.548e-1 | 3.937e-1 | 5.456e-1 | 5.091e-1 | 5.790e-1 | 5.742e-1 |
| 2 | S-HMC$_\parallel$ | 8.281e-2 | 3.896e-1 | 2.891e-1 | 5.681e-1 | 4.285e-1 | 5.684e-1 | 5.521e-1 | 6.125e-1 | 6.107e-1 |
| 4 | S-SMC$_\parallel$ | 8.254e-2 | 3.884e-1 | 2.720e-1 | 5.576e-1 | 3.985e-1 | 5.601e-1 | 5.295e-1 | 5.963e-1 | 5.815e-1 |
| 4 | S-HMC$_\parallel$ | 8.262e-2 | 3.898e-1 | 2.891e-1 | 5.684e-1 | 4.232e-1 | 5.673e-1 | 5.377e-1 | 6.139e-1 | 6.100e-1 |
| 8 | S-SMC$_\parallel$ | 8.206e-2 | 3.899e-1 | 2.744e-1 | 5.596e-1 | 3.987e-1 | 5.555e-1 | 5.304e-1 | 6.025e-1 | 5.920e-1 |
| 8 | S-HMC$_\parallel$ | 8.254e-2 | 3.904e-1 | 2.893e-1 | 5.683e-1 | 4.188e-1 | 5.626e-1 | 5.386e-1 | 6.156e-1 | 6.088e-1 |

Table 28: Comparison of S-SMC$_\parallel$ ($P$ chains with $N = 10$) and S-HMC$_\parallel$ ($NP$ chains), with fixed number of leapfrog $L = 1$, $B = 26$, $M = 2$, $v = 1$ and $s = 0.25$, on IMDb (5 realizations and $\pm$ s.e. in accuracy).



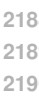

| $P$ | Method | Ep. | Acc. | NLL | $H_{\text{ep}}$ ID | | $H_{\text{ep}}$ OOD | | | | |
|---|---|---|---|---|---|---|---|---|---|---|---|
| | | | | | cor. | inc. | reviews | meta | lipsum | full reviews | full meta |
| 1 | S-SMC$_\parallel$ | 29.6 | 88.27±0.10 | 2.807e-1 | 2.512e-4 | 6.069e-4 | 4.733e-4 | 7.166e-4 | 8.590e-4 | 9.313e-4 | 9.520e-4 |
| 1 | S-HMC$_\parallel$ | 26 | 88.83±0.02 | 2.745e-1 | 1.269e-2 | 2.830e-2 | 4.522e-2 | 5.927e-2 | 5.886e-2 | 7.342e-2 | 6.803e-2 |
| 2 | S-SMC$_\parallel$ | 27.2 | 88.64±0.05 | 2.752e-1 | 4.948e-3 | 1.174e-2 | 1.344e-2 | 2.120e-2 | 1.410e-2 | 2.411e-2 | 2.792e-2 |
| 2 | S-HMC$_\parallel$ | 26 | 88.84±0.03 | 2.741e-1 | 1.304e-2 | 2.918e-2 | 4.586e-2 | 6.400e-2 | 6.096e-2 | 7.596e-2 | 7.216e-2 |
| 4 | S-SMC$_\parallel$ | 28.3 | 88.77±0.04 | 2.737e-1 | 7.211e-3 | 1.662e-2 | 2.329e-2 | 4.029e-2 | 4.913e-2 | 5.544e-2 | 4.798e-2 |
| 4 | S-HMC$_\parallel$ | 26 | 88.90±0.1 | 2.736e-1 | 1.321e-2 | 2.932e-2 | 4.487e-2 | 6.264e-2 | 5.689e-2 | 7.369e-2 | 6.930e-2 |
| 8 | S-SMC$_\parallel$ | 28.5 | 88.87±0.03 | 2.720e-1 | 8.124e-3 | 1.872e-2 | 3.066e-2 | 5.057e-2 | 4.691e-2 | 6.403e-2 | 5.777e-2 |
| 8 | S-HMC$_\parallel$ | 26 | 88.92±0.02 | 2.734e-1 | 1.337e-2 | 2.964e-2 | 4.442e-2 | 6.215e-2 | 5.863e-2 | 7.869e-2 | 7.108e-2 |

| $P$ | Method | Brier | ECE | $H_{\text{tot}}$ ID | | $H_{\text{tot}}$ OOD | | | | |
|---|---|---|---|---|---|---|---|---|---|---|
| | | | | cor. | inc. | reviews | meta | lipsum | full reviews | full meta |
| 1 | S-SMC$_\parallel$ | 8.548e-2 | 3.825e-1 | 2.673e-1 | 5.500e-1 | 3.562e-1 | 4.872e-1 | 4.727e-1 | 5.548e-1 | 5.609e-1 |
| 1 | S-HMC$_\parallel$ | 8.286e-2 | 3.896e-1 | 2.873e-1 | 5.667e-1 | 4.239e-1 | 5.585e-1 | 5.533e-1 | 6.117e-1 | 6.076e-1 |
| 2 | S-SMC$_\parallel$ | 8.343e-2 | 3.867e-1 | 2.723e-1 | 5.560e-1 | 3.696e-1 | 5.182e-1 | 4.861e-1 | 5.794e-1 | 5.849e-1 |
| 2 | S-HMC$_\parallel$ | 8.271e-2 | 3.894e-1 | 2.870e-1 | 5.668e-1 | 4.228e-1 | 5.661e-1 | 5.478e-1 | 6.112e-1 | 6.061e-1 |
| 4 | S-SMC$_\parallel$ | 8.283e-2 | 3.888e-1 | 2.758e-1 | 5.589e-1 | 3.823e-1 | 5.388e-1 | 5.321e-1 | 5.920e-1 | 5.801e-1 |
| 4 | S-HMC$_\parallel$ | 8.254e-2 | 3.902e-1 | 2.870e-1 | 5.665e-1 | 4.174e-1 | 5.633e-1 | 5.328e-1 | 6.118e-1 | 6.058e-1 |
| 8 | S-SMC$_\parallel$ | 8.220e-2 | 3.901e-1 | 2.772e-1 | 5.609e-1 | 3.891e-1 | 5.447e-1 | 5.340e-1 | 6.036e-1 | 5.874e-1 |
| 8 | S-HMC$_\parallel$ | 8.249e-2 | 3.904e-1 | 2.871e-1 | 5.665e-1 | 4.138e-1 | 5.595e-1 | 5.338e-1 | 6.135e-1 | 6.047e-1 |

Table 29: Comparison of S-SMC$_\parallel$ ($P$ chains with $N = 10$) and S-HMC$_\parallel$ ($NP$ chains), with fixed number of leapfrog $L = 1$, $B = 26$, $M = 2$, $v = 1$ and $s = 0.1$, on IMDb (5 realizations and $\pm$ s.e. in accuracy).

| $P$ | Method | Ep. | Acc. | NLL | $H_{\text{ep}}$ | | | | | | |
| --- | --- | --- | --- | --- | --- | --- | --- | --- | --- | --- | --- |
| | | | | | ID | | OOD | | | | |
| | | | | | cor. | inc. | reviews | meta | lipsum | full reviews | full meta |
| 1 | S-SMC$_\parallel$ | 24 | 88.54±0.11 | 2.763e-1 | 1.820e-4 | 4.315e-4 | 4.819e-4 | 5.915e-4 | 6.766e-4 | 7.711e-4 | 6.500e-4 |
| 1 | S-HMC$_\parallel$ | 26 | 88.86±0.02 | 2.726e-1 | 5.753e-3 | 1.319e-2 | 2.712e-2 | 3.551e-2 | 3.561e-2 | 5.110e-2 | 4.289e-2 |
| 2 | S-SMC$_\parallel$ | 23.2 | 88.78±0.08 | 2.727e-1 | 2.476e-3 | 5.724e-3 | 1.160e-2 | 1.973e-2 | 9.401e-3 | 1.314e-2 | 1.894e-2 |
| 2 | S-HMC$_\parallel$ | 26 | 88.90±0.02 | 2.724e-1 | 5.934e-3 | 1.365e-2 | 2.817e-2 | 3.918e-2 | 3.793e-2 | 5.296e-2 | 4.581e-2 |
| 4 | S-SMC$_\parallel$ | 23.5 | 88.85±0.05 | 2.722e-1 | 3.730e-3 | 8.583e-3 | 1.937e-2 | 2.851e-2 | 2.377e-2 | 3.216e-2 | 3.538e-2 |
| 4 | S-HMC$_\parallel$ | 26 | 88.92±0.01 | 2.722e-1 | 5.992e-3 | 1.368e-2 | 2.753e-2 | 3.839e-2 | 3.531e-2 | 5.049e-2 | 4.444e-2 |
| 8 | S-SMC$_\parallel$ | 23.7 | 88.92±0.01 | 2.711e-1 | 4.207e-3 | 9.768e-3 | 2.182e-2 | 3.140e-2 | 2.700e-2 | 3.352e-2 | 3.540e-2 |
| 8 | S-HMC$_\parallel$ | 26 | 88.93±0.01 | 2.721e-1 | 6.065e-3 | 1.386e-2 | 2.792e-2 | 3.888e-2 | 3.653e-2 | 5.408e-2 | 4.638e-2 |

| $P$ | Method | Brier | ECE | $H_{\text{tot}}$ | | | | | | |
| --- | --- | --- | --- | --- | --- | --- | --- | --- | --- | --- |
| | | | | ID | | OOD | | | | |
| | | | | cor. | inc. | reviews | meta | lipsum | full reviews | full meta |
| 1 | S-SMC$_\parallel$ | 8.387e-2 | 3.869e-1 | 2.721e-1 | 5.544e-1 | 3.970e-1 | 5.202e-1 | 5.063e-1 | 5.686e-1 | 5.684e-1 |
| 1 | S-HMC$_\parallel$ | 8.232e-2 | 3.898e-1 | 2.802e-1 | 5.618e-1 | 4.088e-1 | 5.505e-1 | 5.384e-1 | 6.066e-1 | 5.975e-1 |
| 2 | S-SMC$_\parallel$ | 8.252e-2 | 3.892e-1 | 2.734e-1 | 5.571e-1 | 3.939e-1 | 5.263e-1 | 5.028e-1 | 5.812e-1 | 5.855e-1 |
| 2 | S-HMC$_\parallel$ | 8.227e-2 | 3.900e-1 | 2.801e-1 | 5.614e-1 | 4.092e-1 | 5.567e-1 | 5.347e-1 | 6.068e-1 | 5.966e-1 |
| 4 | S-SMC$_\parallel$ | 8.229e-2 | 3.900e-1 | 2.750e-1 | 5.583e-1 | 3.944e-1 | 5.389e-1 | 5.114e-1 | 5.948e-1 | 5.854e-1 |
| 4 | S-HMC$_\parallel$ | 8.218e-2 | 3.903e-1 | 2.800e-1 | 5.612e-1 | 4.041e-1 | 5.526e-1 | 5.222e-1 | 6.068e-1 | 5.967e-1 |
| 8 | S-SMC$_\parallel$ | 8.190e-2 | 3.906e-1 | 2.752e-1 | 5.588e-1 | 3.920e-1 | 5.407e-1 | 5.143e-1 | 5.959e-1 | 5.865e-1 |
| 8 | S-HMC$_\parallel$ | 8.217e-2 | 3.906e-1 | 2.801e-1 | 5.613e-1 | 4.028e-1 | 5.510e-1 | 5.231e-1 | 6.081e-1 | 5.961e-1 |

Table 30: Comparison of S-SMC$_\parallel$ ($P$ chains with $N = 10$) and S-HMC$_\parallel$ ($NP$ chains), with fixed number of leapfrog $L = 1$, $B = 200$, $M = 4$, $v = 0.2$ and $s = 0.05$, on CIFAR10 (5 realizations and $\pm$s.e. in accuracy).

| $P$ | Method | Ep. | Acc. | NLL | Brier | ECE |
| --- | --- | --- | --- | --- | --- | --- |
| 1 | S-SMC$_\parallel$ | 168.8 | 89.26 ± 0.07 | 3.408e-1 | 1.580e-1 | 3.470e-2 |
| 1 | S-HMC$_\parallel$ | 200 | 90.67 ± 0.03 | 2.749e-1 | 1.366e-1 | 6.598e-3 |
| 2 | S-SMC$_\parallel$ | 172.0 | 90.12 ± 0.06 | 3.100e-1 | 1.466e-1 | 1.942e-2 |
| 2 | S-HMC$_\parallel$ | 200 | 90.80 ± 0.02 | 2.707e-1 | 1.351e-1 | 5.574e-3 |
| 4 | S-SMC$_\parallel$ | 173.4 | 90.34 ± 0.04 | 2.960e-1 | 1.399e-1 | 1.242e-2 |
| 4 | S-HMC$_\parallel$ | 200 | 90.84 ± 0.04 | 2.688e-1 | 1.344e-1 | 6.442e-3 |
| 8 | S-SMC$_\parallel$ | 174.3 | 90.63 ± 0.05 | 2.881e-1 | 1.371e-1 | 9.720e-3 |
| 8 | S-HMC$_\parallel$ | 200 | 90.84 ± 0.03 | 2.677e-1 | 1.340e-1 | 6.601e-3 |

| $P$ | Method | $H_{\text{ep}}$ | | | | | $H_{\text{tot}}$ | | | | |
| --- | --- | --- | --- | --- | --- | --- | --- | --- | --- | --- | --- |
| | | ID | | OOD | | | ID | | OOD | | |
| | | cor. | inc. | close | corrupt | far | cor. | inc. | close | corrupt | far |
| 1 | S-SMC$_\parallel$ | 4.258e-4 | 2.273e-3 | 2.515e-3 | 1.297e-3 | 2.185e-3 | 1.351e-1 | 6.620e-1 | 6.639e-1 | 4.300e-1 | 9.008e-1 |
| 1 | S-HMC$_\parallel$ | 4.060e-2 | 1.804e-1 | 2.308e-1 | 1.228e-1 | 2.243e-1 | 1.917e-1 | 8.073e-1 | 8.900e-1 | 5.558e-1 | 1.154e+0 |
| 2 | S-SMC$_\parallel$ | 1.796e-2 | 9.094e-2 | 1.115e-1 | 5.702e-2 | 1.340e-1 | 1.321e-1 | 6.646e-1 | 7.737e-1 | 4.864e-1 | 1.017e+0 |
| 2 | S-HMC$_\parallel$ | 4.376e-2 | 1.896e-1 | 2.454e-1 | 1.300e-1 | 2.387e-1 | 1.956e-1 | 8.144e-1 | 9.064e-1 | 5.626e-1 | 1.162e+0 |
| 4 | S-SMC$_\parallel$ | 2.855e-2 | 1.356e-1 | 1.700e-1 | 8.859e-2 | 1.862e-1 | 1.304e-1 | 6.700e-1 | 8.293e-1 | 5.188e-1 | 1.101e+0 |
| 4 | S-HMC$_\parallel$ | 4.523e-2 | 1.937e-1 | 2.520e-1 | 1.337e-1 | 2.417e-1 | 1.970e-1 | 8.161e-1 | 9.129e-1 | 5.660e-1 | 1.163e+0 |
| 8 | S-SMC$_\parallel$ | 3.507e-2 | 1.584e-1 | 2.027e-1 | 1.058e-1 | 1.961e-1 | 1.311e-1 | 6.684e-1 | 8.643e-1 | 5.368e-1 | 1.111e+0 |
| 8 | S-HMC$_\parallel$ | 4.579e-2 | 1.966e-1 | 2.564e-1 | 1.358e-1 | 2.472e-1 | 1.971e-1 | 8.203e-1 | 9.173e-1 | 5.684e-1 | 1.168e+0 |

Table 31: Comparison of S-SMC$_\parallel$ ($P$ chains with $N = 10$) and S-HMC$_\parallel$ ($NP$ chains), with fixed number of leapfrog $L = 1$, $B = 200$, $M = 4$, $v = 0.2$ and $s = 0.1$, on CIFAR10 (5 realizations and $\pm$s.e. in accuracy).

| $P$ | Method | Ep. | Acc. | NLL | Brier | ECE |
|---|---|---|---|---|---|---|
| 1 | S-SMC$_\parallel$ | 229.6 | $88.26 \pm 0.07$ | 3.855e-1 | 1.770e-1 | 4.593e-2 |
| 1 | S-HMC$_\parallel$ | 200 | $90.57 \pm 0.04$ | 2.823e-1 | 1.398e-1 | 1.073e-2 |
| 2 | S-SMC$_\parallel$ | 226.0 | $89.62 \pm 0.09$ | 3.336e-1 | 1.553e-1 | 1.983e-2 |
| 2 | S-HMC$_\parallel$ | 200 | $90.76 \pm 0.04$ | 2.753e-1 | 1.372e-1 | 1.356e-2 |
| 4 | S-SMC$_\parallel$ | 224.8e | $90.20 \pm 0.08$ | 3.100e-1 | 1.453e-1 | 9.413e-3 |
| 4 | S-HMC$_\parallel$ | 200 | $90.84 \pm 0.04$ | 2.722e-1 | 1.360e-1 | 1.517e-2 |
| 8 | S-SMC$_\parallel$ | 225.3 | $90.45 \pm 0.06$ | 2.980e-1 | 1.400e-1 | 7.737e-3 |
| 8 | S-HMC$_\parallel$ | 200 | $90.83 \pm 0.03$ | 2.701e-1 | 1.353e-1 | 1.517e-2 |

| $P$ | Method | $H_{\text{ep}}$ | | | | | $H_{\text{tot}}$ | | | | |
|---|---|---|---|---|---|---|---|---|---|---|---|
| | | ID | | OOD | | | ID | | OOD | | |
| | | cor. | inc. | close | corrupt | far | cor. | inc. | close | corrupt | far |
| 1 | S-SMC$_\parallel$ | 2.948e-4 | 1.507e-3 | 1.603e-3 | 8.256e-4 | 1.450e-3 | 1.309e-1 | 6.364e-1 | 6.121e-1 | 4.027e-1 | 9.110e-1 |
| 1 | S-HMC$_\parallel$ | 7.055e-2 | 2.795e-1 | 3.591e-1 | 1.972e-1 | 3.581e-1 | 2.241e-1 | 8.596e-1 | 9.703e-1 | 6.102e-1 | 1.219e+0 |
| 2 | S-SMC$_\parallel$ | 2.733e-2 | 1.343e-1 | 1.594e-1 | 8.581e-2 | 1.677e-1 | 1.256e-1 | 6.382e-1 | 7.735e-1 | 4.910e-1 | 1.037e+0 |
| 2 | S-HMC$_\parallel$ | 9.734e-2 | 2.968e-1 | 3.862e-1 | 2.112e-1 | 3.821e-1 | 2.316e-1 | 8.732e-1 | 9.990e-1 | 6.238e-1 | 1.235e+0 |
| 4 | S-SMC$_\parallel$ | 4.484e-2 | 2.002e-1 | 2.495e-1 | 1.365e-1 | 2.536e-1 | 1.235e-1 | 6.432e-1 | 8.577e-1 | 5.408e-1 | 1.135e+0 |
| 4 | S-HMC$_\parallel$ | 1.018e-1 | 3.072e-1 | 4.011e-1 | 2.200e-1 | 3.936e-1 | 2.360e-1 | 8.822e-1 | 1.014e+0 | 6.323e-1 | 1.247e+0 |
| 8 | S-SMC$_\parallel$ | 5.539e-2 | 2.369e-1 | 3.054e-1 | 1.636e-1 | 2.937e-1 | 1.217e-1 | 6.453e-1 | 9.184e-1 | 5.690e-1 | 1.156e+0 |
| 8 | S-HMC$_\parallel$ | 1.035e-1 | 3.121e-1 | 4.095e-1 | 2.244e-1 | 4.031e-1 | 2.367e-1 | 8.884e-1 | 1.023e+0 | 6.371e-1 | 1.257e+0 |

Table 32: Comparison of S-SMC$_\parallel$ ($P$ chains with $N = 10$) and S-HMC$_\parallel$ ($NP$ chains), with fixed number of leapfrog $L = 1$, $B = 200$, $M = 4$, $v = 0.2$ and $s = 0.2$, on CIFAR10 (5 realizations and $\pm$s.e. in accuracy).

| $P$ | Method | Ep. | Acc. | NLL | Brier | ECE |
|---|---|---|---|---|---|---|
| 1 | S-SMC$_\parallel$ | 289.6 | $86.99 \pm 0.08$ | 4.710e-1 | 2.007e-1 | 6.462e-2 |
| 1 | S-HMC$_\parallel$ | 200 | $90.23 \pm 0.08$ | 2.990e-1 | 1.466e-1 | 2.518e-2 |
| 2 | S-SMC$_\parallel$ | 289.6 | $88.77 \pm 0.07$ | 3.854e-1 | 1.699e-1 | 2.554e-2 |
| 2 | S-HMC$_\parallel$ | 200 | $90.53 \pm 0.04$ | 2.890e-1 | 1.426e-1 | 3.096e-2 |
| 4 | S-SMC$_\parallel$ | 289.4 | $89.82 \pm 0.04$ | 3.441e-1 | 1.536e-1 | 1.193e-2 |
| 4 | S-HMC$_\parallel$ | 200 | $90.73 \pm 0.02$ | 2.840e-1 | 1.406e-1 | 3.368e-2 |
| 8 | S-SMC$_\parallel$ | 289.3 | $90.30 \pm 0.03$ | 3.217e-1 | 1.445e-1 | 1.180e-2 |
| 8 | S-HMC$_\parallel$ | 200 | $90.82 \pm 0.03$ | 2.810e-1 | 1.395e-1 | 3.481e-2 |

| $P$ | Method | $H_{\text{ep}}$ | | | | | $H_{\text{tot}}$ | | | | |
|---|---|---|---|---|---|---|---|---|---|---|---|
| | | ID | | OOD | | | ID | | OOD | | |
| | | cor. | inc. | close | corrupt | far | cor. | inc. | close | corrupt | far |
| 1 | S-SMC$_\parallel$ | 3.682e-4 | 1.947e-3 | 2.063e-3 | 1.092e-3 | 1.629e-3 | 1.136e-1 | 5.613e-1 | 5.440e-1 | 3.630e-1 | 7.756e-1 |
| 1 | S-HMC$_\parallel$ | 1.159e-1 | 4.091e-1 | 5.195e-1 | 2.993e-1 | 5.333e-1 | 2.676e-1 | 9.231e-1 | 1.059e+0 | 6.768e-1 | 1.297e+0 |
| 2 | S-SMC$_\parallel$ | 3.863e-2 | 1.856e-1 | 2.117e-1 | 1.180e-1 | 2.030e-1 | 1.108e-1 | 5.822e-1 | 7.557e-1 | 4.901e-1 | 9.434e-1 |
| 2 | S-HMC$_\parallel$ | 1.300e-1 | 4.377e-1 | 5.681e-1 | 3.260e-1 | 5.774e-1 | 2.836e-1 | 9.479e-1 | 1.109e+0 | 7.039e-1 | 1.331e+0 |
| 4 | S-SMC$_\parallel$ | 6.722e-2 | 2.780e-1 | 3.470e-1 | 1.916e-1 | 3.332e-1 | 1.089e-1 | 5.885e-1 | 8.861e-1 | 5.634e-1 | 1.087e+0 |
| 4 | S-HMC$_\parallel$ | 1.368e-1 | 4.524e-1 | 5.899e-1 | 3.392e-1 | 5.947e-1 | 2.909e-1 | 9.611e-1 | 1.132e+0 | 7.169e-1 | 1.346e+0 |
| 8 | S-SMC$_\parallel$ | 8.362e-2 | 3.326e-1 | 4.244e-1 | 2.361e-1 | 4.065e-1 | 1.071e-1 | 5.954e-1 | 9.675e-1 | 6.080e-1 | 1.160e+0 |
| 8 | S-HMC$_\parallel$ | 1.405e-1 | 4.604e-1 | 6.042e-1 | 3.476e-1 | 6.129e-1 | 2.945e-1 | 9.687e-1 | 1.146e+0 | 7.256e-1 | 1.364e+0 |

