# OpenReview forum: "Scalable Bayesian Monte Carlo: fast uncertainty estimation beyond deep ensembles"
_ICLR.cc/2026/Conference — Submitted to ICLR 2026_

### Official Review · Reviewer_otV6 · 2025-10-16

**Soundness:** 2
**Presentation:** 1
**Contribution:** 2
**Rating:** 2
**Confidence:** 4

**Summary:**

The paper proposes Scalable Bayesian Monte Carlo (SBMC) as a new framework aiming to combine point estimation with posterior sampling for uncertainty quantification in deep learning. SBMC conceptually interpolates between deterministic estimators and full Bayesian inference through parallelized SMC or MCMC procedures. The authors assert that this yields superior epistemic uncertainty estimates at a computational cost similar to deep ensembles. Nevertheless, much of the approach appears to repackage established concepts rather than introducing fundamentally new methodology. Further, the authors devote substantial effort to extensive benchmarking.

**Strengths:**

* **Simple concept of prior anchoring:** The core and simple modeling idea of interpolating between a point estimate and a posterior approximation is conceptually interesting. The fact that this framework is sampler-agnostic is a definite plus, allowing for flexibility in its implementation. I also liked the differentiation from the tempering-induced interpolation in the literature concerned with Cold Posteriors.
* **Extensive Empirical Evaluation:** A significant amount of work has clearly gone into the experimental validation. The large number of experiments/ablations in the appendix demonstrates a thorough empirical effort, which is commendable.

**Weaknesses:**

* **Concerns Regarding Novelty:** The paper's primary weakness is its limited novelty, particularly on the algorithmic side. As far as I understand it, the proposed algorithm is essentially a parallel MCMC or SMC sampler initialized near a MAP estimate and with slightly altered prior. This is not a new concept; many old (e.g. [1]) and also recent works, including some cited by the authors in Section 4 (e.g., SMS-UBU, Bayesian Deep Ensembles), have explored similar ideas. The paper fails to provide a clear differentiation from or comparison to these highly related methods, making the contribution seem incremental. Furthermore, the authors claim that SMS-UBU "did not perform as well" in their experiments without providing any empirical evidence to substantiate this claim, which is a serious omission. Large portions of the paper (e.g., the discussion of SMC and MCMC on page 4 & algorithms 2 and 3) reiterate well-known results/methods.

* **Imprecise Framing:** The framing of the work contains several imprecise and potentially misleading statements. Referring to the $s=1$ case as the "full Bayesian posterior" is imprecise; any MCMC/SMC method with a finite compute budget can only ever produce an *approximation* of the true posterior. The introduction also makes claims about the method without formal definitions, making the initial exposition difficult to follow. The framing of Deep Ensembles as Bayesian is also problematic, as it's widely understood that DEs are not a consistent approximation of the Bayesian posterior (see [2]).

* **Choice of benchmarks:** Figure 1 is unclear, difficult to interpret, and uses the term "mixing" in a non-standard way for a multimodal toy target, where a low ACF is not a reliable indicator of global mixing. This highlights the fact that it is unclear how the target (if it exists), to which SBMC converges, is characterized.  Furthermore, the paper uses out-of-distribution (OOD) detection benchmarks to validate its claims on epistemic UQ. However, recent work argues convincingly that these two objectives are fundamentally misaligned [3], making the OOD benchmark results a questionable proxy for the quality of epistemic uncertainty estimation. Overall, the experiments feel scattered and do not follow a clear or consistent line of reasoning.

* **Structure and Presentation:** The paper's structure and overall presentation need improvement. The writing is often unclear (many concepts are not properly defined/introduced early on), and crucial components like the "anchoring" step are not explained with sufficient clarity. The decision to include preliminary results on a new experiment in Section 5.1 is an odd choice that makes the work feel unfinished. Finally, the manuscript suffers from numerous formatting errors and incomplete references (e.g., missing venues for Papamarkou et al., 2024; Paulin et al., 2024, the latter should be 2025), which detracts from its overall quality and polish.


## References


[1] Vehtari, A., Sarkka, S., & Lampinen, J. (2000, July). On MCMC sampling in Bayesian MLP neural networks. In Proceedings of the IEEE-INNS-ENNS International Joint Conference on Neural Networks. IJCNN 2000. Neural Computing: New Challenges and Perspectives for the New Millennium (Vol. 1, pp. 317-322). IEEE.

[2] Wild, V. D., Ghalebikesabi, S., Sejdinovic, D., & Knoblauch, J. (2023). A rigorous link between deep ensembles and (variational) bayesian methods. Advances in Neural Information Processing Systems, 36, 39782-39811.

[3] Li, Y. L., Lu, D., Kirichenko, P., Qiu, S., Rudner, T. G., Bruss, C. B., & Wilson, A. G. (2025). Out-of-Distribution Detection Methods Answer the Wrong Questions. ICML.

**Questions:**

Please address my following questions. Satisfactory responses, in particular to (1-2), would in my opinion significantly strengthen the paper.

**(1)** The primary benefit of the proposed method could simply stem from initializing sampling chains at or near the MAP solution, a strategy explored in prior work. To disentangle the benefit of your novel prior anchoring from this initialization effect, could you provide an ablation study? Did you investigate this question already? Specifically, the ablation I have in mind is to compare SBMC to a standard MCMC/SMC sampler that is simply initialized at the MAP but uses a standard, non-interpolated prior (i.e., fix $s=1$ and start at the MAP).

**(2)** The paper mentions several state-of-the-art methods in Section 4 but omits them from direct comparison in Table 1 (e.g., SMS-UBU, Bayesian Deep Ensembles). Can you either include these highly relevant baselines or provide a convincing justification for their exclusion? Furthermore, could you please provide the empirical evidence to substantiate your claim that SMS-UBU "did not perform as well in practice"?

**(3)** Could you provide practical guidance or heuristics on how to set the sampling duration for the MCMC/SMC component of SBMC? Additionally, the results in Figure 3 show that most sampling-based methods perform poorly and with extreme variance on the abstention recall metric. Do you have an explanation for this behavior?

**(4)** You explore both (SG)MCMC and SMC samplers for the BMC component in your model. There seem to be considerable differences in performance on a per-task basis between the two approaches. In the conclusion, you state that both algorithms are “attractive options”. Could you elaborate on which approach you would generally suggest a practitioner to use?

---

> ### Author Response · Authors · 2025-11-22
> **Rebuttal**
>
> Thank you very much for carefully reading our paper, for recognising its strengths, and for providing us with valuable and actionable recommendations for improvement. We have processed these carefully and taken all suggestions for improvement into account, as detailed below.
>
> ## Weaknesses
>
> > **W1. Concerns Regarding Novelty**: The paper's primary weakness is its limited novelty, particularly on the algorithmic side. As far as I understand it, the proposed algorithm is essentially a parallel MCMC or SMC sampler initialized near a MAP estimate and with a slightly altered prior. This is not a new concept; many old (e.g. [1]) and also recent works, including some cited by the authors in Section 4 (e.g., SMS-UBU, Bayesian Deep Ensembles), have explored similar ideas. The paper fails to provide a clear differentiation from or comparison to these highly related methods, making the contribution seem incremental. Furthermore, the authors claim that SMS-UBU "did not perform as well" in their experiments without providing any empirical evidence to substantiate this claim, which is a serious omission. Large portions of the paper (e.g., the discussion of SMC and MCMC on page 4 & algorithms 2 and 3) reiterate well-known results/methods.
>
> See the Global rebuttal on Novelty and Table 1 there, as well as the responses to Q1-2 below and WB64 W1. We appreciate the pointer to [1], and cited it in the revision.
>
> > **W2. Imprecise Framing**: The framing of the work contains several imprecise and potentially misleading statements. Referring to the $s=1$ case as the "full Bayesian posterior" is imprecise; any MCMC/SMC method with a finite compute budget can only ever produce an approximation of the true posterior.
>
> Correct, and apologies for the imprecise phrasing on line 106, which we have updated as follows, “SBMC method allows the practitioner to *target a model* which  interpolates between the MAP estimator for $s=0$ and the full posterior for $s=1$”
>
> > The introduction also makes claims about the method without formal definitions, making the initial exposition difficult to follow.
>
> On line 41 we introduced $\overline{\pi}_s$ before $\pi$, which was confusing. We moved the definition of the posterior $\pi$ before this, and point the user to the formal definitions in (2-3) for details.
>
> > The framing of Deep Ensembles as Bayesian is also problematic, as it's widely understood that DEs are not a consistent approximation of the Bayesian posterior (see [2]).
>
> We agree that Deep Ensembles do not deliver a consistent approximation of the posterior, and we only ever referred to them as approximations which are not consistent. We further emphasized this point and cited [2] and [4].
>
> [4] https://cims.nyu.edu/~andrewgw/deepensembles/

---

> ### Author Response · Authors · 2025-11-22
> **Rebuttal, continued (1)**
>
> > **W3. Choice of benchmarks**: Figure 1 is unclear, difficult to interpret, and uses the term "mixing" in a non-standard way for a multimodal toy target, where a low ACF is not a reliable indicator of global mixing. This highlights the fact that it is unclear how the target (if it exists), to which SBMC converges, is characterized. Furthermore, the paper uses out-of-distribution (OOD) detection benchmarks to validate its claims on epistemic UQ. However, recent work argues convincingly that these two objectives are fundamentally misaligned [3], making the OOD benchmark results a questionable proxy for the quality of epistemic uncertainty estimation. Overall, the experiments feel scattered and do not follow a clear or consistent line of reasoning.
>
> We changed the y-axes of panels (a) and (c ) from "Accuracy" to "Test Accuracy", and changed the term ‘mixing’ to ‘decorrelation’ in panel (d). We agree that ACF may in general underestimate mixing time for multimodal targets, and sometimes a lot. The intention here is just to emphasize that the chains converge faster, which you correctly point out is already quite obvious from panel 1(b).
>
> The limit $s=0$ may not be formally defined, but we only define the SBMC target for $s>0$. There is a typo on line 40 (former, now 50), which we changed to an *open* interval (see line 143, now 145).
>
> The paper [3] does an excellent job of exposing the limitations of using supervised models for OOD detection and makes a compelling call for rethinking the objective. We agree with this critique and explicitly cite and acknowledge it. In fact, [3] helps to *strengthen the framing of our work*. Our focus is not OOD detection in the traditional sense, and we appreciate the reviewer highlighting the opportunity to clarify this. Rather, we propose a novel approach to use improved epistemic uncertainty to predict confidence in model outputs, and then leverage this confidence to decide whether to respond or abstain, within a novel two-level estimator. This is fundamentally different from predicting whether an input is OOD, which [3] correctly argues is conceptually misspecified and misaligned with supervised classifiers. We acknowledge that some of our original language inadvertently blurred this distinction, and we will revise the manuscript to make our intention and framing precise.
>
> Furthermore, while many OOD and uncertainty-based methods rely on fixed scoring rules (e.g. entropy, energy, MSP), our two-level estimator uses 7 features, 3 of which specifically characterize epistemic uncertainty, as inputs to a learned meta-classifier that predicts the probability of correctness, not OOD membership. This aligns with the reframing proposed in [3]—the goal is not to detect distributional novelty (fundamentally misaligned), but to determine when the model's prediction (over its learned classes) will likely be unreliable, enabling principled abstention.
>
> We would like to emphasize that our first stage training can already predict confidence, as shown in Figure 2. Furthermore, we have conducted an ablation experiment in Table 4, which shows that indeed $H_{\sf ep}$ alone (without any second stage meta-classifier) can already predict confidence almost as well as the meta-classifier.
>
> > the paper uses out-of-distribution (OOD) detection benchmarks to validate its claims on epistemic UQ
>
> There has been a misunderstanding, and we will make this point much more clear. *We have not used OOD detection to validate epistemic UQ quality. We validate the quality of the epistemic UQ using very long HMC runs as a ‘gold-standard’, which is a well-accepted empirical validation approach (see Table 1 and Figures 1, 2, and 3, as well as the results in the Appendix)*.
>
> We then go on to empirically demonstrate with examples that this improved epistemic UQ has discriminative power for *predicting confidence in model output* (Tables 3,4 and Figure 5, revised version, T2 and F4 original). And we furthermore show how that confidence can be used to *decide whether to abstain from responding*, with good empirical results – see the results of the 2-level estimator in Figure 5 (previously 4) and Table 4. We also provided an additional figure to clarify the meta-classifier and 2-level estimator (Figure 4 in the uploaded revision -- the only change).
>
> We clearly summarized this narrative and reasoning at the very beginning of Section 3, for solid grounding. And we included a careful discussion of and contextualisation with respect to [3] in Appendix B.

---

> ### Author Response · Authors · 2025-11-22
> **Rebuttal, continued (2)**
>
> > **W4. Structure and Presentation**: The paper's structure and overall presentation need improvement. The writing is often unclear (many concepts are not properly defined/introduced early on), and crucial components like the "anchoring" step are not explained with sufficient clarity. The decision to include preliminary results on a new experiment in Section 5.1 is an odd choice that makes the work feel unfinished. Finally, the manuscript suffers from numerous formatting errors and incomplete references (e.g., missing venues for Papamarkou et al., 2024; Paulin et al., 2024, the latter should be 2025), which detracts from its overall quality and polish.
>
> The “anchor” is introduced in equation (3) and below that it says, “which we will refer to as the anchored posterior. We will refer to $\theta_{MAP}$ as the anchor.” We would be grateful if you could share any more detail about what you are finding unclear there, and we will clarify it.
>
> We moved the GPT-2 example from Section 5.1 to the end of Section 3, positioning it clearly as a preliminary scalability case study rather than a fully developed LLM UQ demonstration. The results are preliminary simply because properly treating LLMs requires characterizing *semantic* epistemic uncertainty, which is beyond the scope of the present work. We wanted to demonstrate a very large example and point to where we intend to go next by showing token-level uncertainty, however it does not make sense to do more extensive token-level experiments when semantic uncertainty is needed.
>
> Thank you for pointing out the reference errors, which we corrected. We also clarified all of the specific suggestions, and will do another very careful pass over the paper to catch any other formatting errors and further polish it. We would welcome and gladly address any other specific suggestions or criticism.
>
>
> > References
>
> > [1] Vehtari, A., Sarkka, S., & Lampinen, J. (2000, July). On MCMC sampling in Bayesian MLP neural networks. In Proceedings of the IEEE-INNS-ENNS International Joint Conference on Neural Networks. IJCNN 2000. Neural Computing: New Challenges and Perspectives for the New Millennium (Vol. 1, pp. 317-322). IEEE.
>
> > [2] Wild, V. D., Ghalebikesabi, S., Sejdinovic, D., & Knoblauch, J. (2023). A rigorous link between deep ensembles and (variational) bayesian methods. Advances in Neural Information Processing Systems, 36, 39782-39811.
>
> > [3] Li, Y. L., Lu, D., Kirichenko, P., Qiu, S., Rudner, T. G., Bruss, C. B., & Wilson, A. G. (2025). Out-of-Distribution Detection Methods Answer the Wrong Questions. ICML.

---

> ### Author Response · Authors · 2025-11-22
> **Rebuttal, continued (3)**
>
> ## Questions:
> > Please address my following questions. Satisfactory responses, in particular to (1-2), would in my opinion significantly strengthen the paper.
>
> > **(Q1)** The primary benefit of the proposed method could simply stem from initializing sampling chains at or near the MAP solution, a strategy explored in prior work. To disentangle the benefit of your novel prior anchoring from this initialization effect, could you provide an ablation study? Did you investigate this question already? Specifically, the ablation I have in mind is to compare SBMC to a standard MCMC/SMC sampler that is simply initialized at the MAP but uses a standard, non-interpolated prior (i.e., fix $s=1$ and start at the MAP).
>
> Please see Novelty and Table 1 (DEI-MCMC) in the General rebuttal for the results for HMC, in comparison with S-HMC associated with $s=0.1$ and $s=0.25$.  These results are now included in Table 1 in the paper also.
>
> The results show that the performance in Accuracy and NLL is significantly improved in comparison to random initialization, as expected. However, the uncertainty estimates are still significantly worse than SBMC (S-HMC). This can be understood as a consequence of the slow mixing.
>
> For a chain of length T, when T=0 all are the same (and equal to the MAP). As T increases, S-HMC improves and converges much earlier than HMC. HMC eventually catches up, and prevails as T tends to \infty and we recover the full posterior. But for small T, S-HMC clearly improves uncertainty quality (Table 1, General rebuttal).
>
>
> > **(Q2)** The paper mentions several state-of-the-art methods in Section 4 but omits them from direct comparison in Table 1 (e.g., SMS-UBU, Bayesian Deep Ensembles). Can you either include these highly relevant baselines or provide a convincing justification for their exclusion? Furthermore, could you please provide the empirical evidence to substantiate your claim that SMS-UBU "did not perform as well in practice"?
>
> We included results, in the General rebuttal Table 1 as well as Table 1 of the paper, for S-HMC with linearly-scaled prior centered on $(1-s)\theta_{MAP}$ instead of $\theta_{MAP}$ [S-HMC(linear)] – this results from having a product form prior of the following form
> $$N(\theta ; 0, v {\sf Id}) N(\theta ; \theta_{MAP}, s' {\sf Id} )
> \propto N(\theta ; (1-s)\theta_{MAP}, s v {\sf Id}) \, , \quad
> s=\frac{s'}{v+s'} ,$$
> as done in SMS-UBU from [Paulin etal, 2025]. Please note that we did not claim to perform better than SMS-UBU, per se. We only claimed that centering on the unscaled MAP prior works better than the scaled MAP from linear interpolation.
>
> The prior with linearly-scaled MAP mean (as in SMS-UBU) performs worse than MAP mean (ours) across the board, giving lower accuracy, higher NLL, and higher $H_{\sf ep}$ for correct predictions and lower for incorrect and OOD [S-HMC(linear) in Table 1 of the General rebuttal and Table 1 of the revision]. This is more pronounced for larger s, where the scale factor is smaller. While continuity is generally speaking an attractive mathematical property, the mechanism underlying the success of the SBMC method is anchoring to the MAP(s) itself (themselves).
>
> *The comparison with SMS-UBU is running and will be included as another competing method in the rebuttal $(\dagger)$.*
>
> Bayesian DEI-MCMC [1,5] (called "Bayesian deep ensemble" in [5]) is also given in Table 1 in the General rebuttal (DEI-HMC). This is what is described in (Q1), from our understanding. If the reviewer meant another variant, we are happy to include it.
>
> [5] Sommer, E., Wimmer, L., Papamarkou, T., Bothmann, L., Bischl, B., & Rügamer, D. (2024). Connecting the Dots: Is Mode-Connectedness the Key to Feasible Sample-Based Inference in Bayesian Neural Networks?. ICML 45988-46018.

---

> ### Author Response · Authors · 2025-11-22
> **Rebuttal, continued (4)**
>
> > **(Q3)** Could you provide practical guidance or heuristics on how to set the sampling duration for the MCMC/SMC component of SBMC? Additionally, the results in Figure 3 show that most sampling-based methods perform poorly and with extreme variance on the abstention recall metric. Do you have an explanation for this behavior?
>
> A practical starting point is to set the duration to the number of epochs used for computing the MAP, to keep the costs comparable for each phase (or both equal to 1/2, which we show now in Table 1 is suitable). We have followed this heuristic and found the performance to be quite good. We clearly state this in the revision. For MCMC, further confidence can be achieved by monitoring the autocorrelation function (ACF), mean displacement, Rhat (function space), or a similar diagnostic (noting that these diagnostics themselves tend to have limited accuracy for short chains). For SMC, comparable diagnostics monitor local mixing for mutations and effective sample size for selections. These are described in Appendices C and D.
>
> Regarding abstention recall, MAP and DE do actually perform a bit better here: (0.95,0.89) for IMDb and (0.84,0.85) for CIFAR10. However, due to the min-max scaler normalization (which we have clarified in the revision) *the variance appears much larger than it is*: the smallest values are 0.88 for IMDb and 0.82 for CIFAR10. For other metrics, the relative variance is smaller because MAP and DE significantly underperform the others. So the mechanism at play here is really the good performance of MAP and DE on recall, which we understand as a consequence of their overconfidence – and they pay the expected penalty in precision (and F1 and AUROC). We clearly commented on this in the revision.
>
>
> > **(Q4)** You explore both (SG)MCMC and SMC samplers for the BMC component in your model. There seem to be considerable differences in performance on a per-task basis between the two approaches. In the conclusion, you state that both algorithms are “attractive options”. Could you elaborate on which approach you would generally suggest a practitioner to use?
>
> In most cases, we would recommend practitioners to use MCMC, as it is simpler and performance is similar or better. If it becomes an issue to store and compute the full likelihood, then we would recommend deferring to SG-MCMC. SMC has better theoretical guarantees, and can potentially provide an advantage for complex problems where severe multimodality harms performance, but generally speaking the implementation overhead and communication constraints are a hindrance that is not justified for comparable performance. In the conclusion of the revision we clearly state these recommendations.

---

> ### Comment · Reviewer_otV6 · 2025-11-24
>
> I thank the authors for their detailed response, the additional experiments, and the effort put into the rebuttal. I have read the other reviews and the global response carefully.
>
> Before I can finalize my assessment, I have several clarifying questions regarding the new data and claims presented in the rebuttal. A revision of the manuscript incorporating the promised changes is also necessary for me to fully evaluate the submission.
>
> 1. Global Table and Deep Ensembles:
> In Table 1 of the global response, the entry for Deep Ens (2x) lists P=1. Does P represent the number of ensemble members? If so, P=1 implies a single model rather than an ensemble. Deep Ensembles typically require M > 5 members to yield uncertainty benefits. Comparing a single model against SBMC (which uses multiple chains/particles) would be an unfair baseline. Please clarify if this is a typo or the actual configuration.
>
> 2. Tuning of New Baselines:
> How were the hyperparameters (e.g., learning rates, prior variances, leapfrog steps, mass matrix) tuned for the newly added baselines in the global table? Were they granted a tuning budget equivalent to the proposed method to ensure a fair comparison?
>
> 3. Diagnostics for Neural Networks:
> In your response regarding practical configuration, you suggest using diagnostics such as R-hat. I am skeptical about the applicability of such diagnostics for Bayesian Neural Networks. Due to the non-identifiability of neural networks (e.g., permutation symmetries), different chains often converge to functionally equivalent but parametrically distant modes. R-hat would therefore remain high even if chains have converged to the typical set, making it ill-suited here. Could you explain why you believe such diagnostics are appropriate given these well-known issues?
>
> 4. Meta-Classifier and OOD Generalization:
> The novelty of training a secondary classifier on uncertainty features appears limited. Moreover, if this stage relies on specific OOD data or validation errors to tune the abstention threshold, it transforms a general UQ problem into a supervised classification task. This relies on the assumption that the test OOD data comes from a similar data generating process (DGP) as the validation OOD data, defeating the purpose of detecting unknown unknowns. How does this compare conceptually to evidential methods like Natural Posterior Networks (Charpentier et al., 2021) that explicitly model ID/OOD distributions without this two-stage supervision? Furthermore, the results in Table 2 seem to rely on OOD datasets that are distributionally close to the training data. To demonstrate robustness, the test OOD DGP should be fundamentally distinct from the training distribution (e.g., evaluating on CelebA when trained on MNIST OOD data).

---

> ### Author Response · Authors · 2025-11-27
> **Re-rebuttal Clarifications**
>
> We thank the reviewer for carefully reading our response(s) and providing further detailed feedback, which we carefully address below. In particular, we are very pleased that you prompted us to dig deeper into the meta-classifier, which resulted in very positive outcomes.
>
> > Before I can finalize my assessment, I have several clarifying questions regarding the new data and claims presented in the rebuttal. A revision of the manuscript incorporating the promised changes is also necessary for me to fully evaluate the submission.
>
> A revision has been uploaded.
>
> >1. Global Table and Deep Ensembles:
> In Table 1 of the global response, the entry for Deep Ens (2x) lists P=1. Does P represent the number of ensemble members? If so, P=1 implies a single model rather than an ensemble. Deep Ensembles typically require M > 5 members to yield uncertainty benefits. Comparing a single model against SBMC (which uses multiple chains/particles) would be an unfair baseline. Please clarify if this is a typo or the actual configuration.
>
>
> All ensemble methods use exactly the same ensemble size. The total number of samples (models) is $NP$, with $N=10$ for the MNIST example associated with Table 1 in the paper and Table 1 in the Global rebuttal. The origin of this splitting of the ensemble is that in SMC the $N$ samples communicate, and $P$ is the number of independent SMCs. This has been clarified in the Table caption.
>
>
>
> >2. Tuning of New Baselines:
> How were the hyperparameters (e.g., learning rates, prior variances, leapfrog steps, mass matrix) tuned for the newly added baselines in the global table? Were they granted a tuning budget equivalent to the proposed method to ensure a fair comparison?
>
> Yes, all chains were tuned identically, for both S-HMC and DEI-HMC, to ensure a fair comparison. The prior variances have been fixed to begin with, for all methods -- i.e. all solve the same target problem. The MAPs are computed using the same learning rate r=0.001 (Adam optimizer) and number of epochs. The number of leapfrog steps is fixed to L=1, with initial stepsize dt=0.01. The step-size is adapted to target the (optimal) acceptance ratio of 0.65 (within the epochs reported). The mass matrix is the identity and is not tuned.
>
> > 3. Diagnostics for Neural Networks:
> In your response regarding practical configuration, you suggest using diagnostics such as R-hat. I am skeptical about the applicability of such diagnostics for Bayesian Neural Networks. Due to the non-identifiability of neural networks (e.g., permutation symmetries), different chains often converge to functionally equivalent but parametrically distant modes. R-hat would therefore remain high even if chains have converged to the typical set, making it ill-suited here. Could you explain why you believe such diagnostics are appropriate given these well-known issues?
>
> Thank you for requesting clarification on this important point. Rhat is defined for a quantity of interest, and that quantity of interest for our BDL problems would be the posterior predictive on the test set, i.e. softmax predictions. It is well-known that these may be functionally equivalent for distant modes in parameter/weight space due to non-identifiability, as you correctly point out. It has been shown before, e.g. in Figure 2 of [**], WB64 rebuttal, that this Rhat does converge in function space, whereas in weight space it converges much more slowly (or not at all).

---

> ### Author Response · Authors · 2025-11-27
> **Re-rebuttal clarifications, continued**
>
> > 4. Meta-Classifier and OOD Generalization:
> The novelty of training a secondary classifier on uncertainty features appears limited. Moreover, if this stage relies on specific OOD data or validation errors to tune the abstention threshold, it transforms a general UQ problem into a supervised classification task. This relies on the assumption that the test OOD data comes from a similar data generating process (DGP) as the validation OOD data, defeating the purpose of detecting unknown unknowns. How does this compare conceptually to evidential methods like Natural Posterior Networks (Charpentier et al., 2020) [6,7] that explicitly model ID/OOD distributions without this two-stage supervision?
>
> Our meta-classifier predicts whether the model prediction will be correct, rather than whether the input is OOD, per se. It does translate model UQ into a confidence metric for abstention, and learns how to do this from its training data, which includes OOD data in our case. However, it is but one down-stream application of the general UQ information, so it does not undermine the epistemic uncertainty itself (or transform that into something else).
>
> We do include OOD calibration data from a particular data generating process (DGP) in the failure class of the meta-training data, and we agree that you can expect it to be more challenging for the model to identify unknown unknowns (OOD) which come from a different DGP that the OOD meta-training data, i.e. meta-OOD data. One thing that could help here is a Bayesian/ensemble method for the *meta-classifier*. In particular, Bayesian models deliver $H_{\sf ep}$, and as we can see in Figure 2 (right), this is very good at identifying far-OOD data such as white-noise.
>
> Evidential methods fit neural networks to parametric uncertainty models (e.g., Dirichlet or NIG), typically via deterministic MLE/MAP [6]. Recent work experimented with ensembles of such point-estimators to improve robustness [7]. Thus, SBMC would be a natural extension to these approaches by sampling from the *posterior* over evidential parameters rather than relying on a single point estimate. We plan to explore this as a Future Direction, and have added it to the list in Appendix A.2.
>
> The outputs from [6,7] are able to model uncertainty using a point estimator (and given only the training data) because of the explicit parametric model. Our approach directly models the *posterior*, and as such it also directly models uncertainty at the *first training stage* (given only training data), as shown for example in the Tables (especially Table 4 in the revision, Table 2 in the Global rebuttal) and very explicitly and directly in Figure 2 (for ID correct/incorrect and near/far OOD data). This is exactly analogous to the direct predictions of [6,7]. As we can see in the new Table 4, $H_{\sf ep}$ alone (without any 2nd stage of supervision and without seeing any OOD data whatsoever) can perform almost as well as the full meta-classifier (and better Bayes means smaller gap). Since the meta-classifier is more robust and only requires a small overhead, we believe it is a worthwhile step. Evidential models like [6,7] could also leverage a second stage meta-classifier to improve their OOD detection performance or to predict output confidence.
>
> [6] Bertrand Charpentier, Daniel Zügner, and Stephan Günnemann. Posterior network: Uncertainty
> estimation without ood samples via density-based pseudo-counts. NeurIPS 33:1356–1367, 2020.
>
> [7] Bertrand Charpentier, Oliver Borchert, Daniel Zügner, Simon Geisler, and Stephan Günnemann.
> Natural posterior network: Deep bayesian predictive uncertainty for exponential family distributions. ICLR 2022. URL https://openreview.net/forum?id=tV3N0DWMxCg

---

> ### Author Response · Authors · 2025-11-27
> **Re-rebuttal clarifications, continued (2 -- more evidence)**
>
> > Furthermore, the results in Table 2 seem to rely on OOD datasets that are distributionally close to the training data. To demonstrate robustness, the test OOD DGP should be fundamentally distinct from the training distribution (e.g., evaluating on CelebA when trained on MNIST OOD data).
>
> Thank you for bringing up this point, which is important to clarify. The results of Table 2 are as follows. The original training data is a subset of MNIST digits 0,...,7. The OOD data is comprised of 4 categories: MNIST 8 and 9 (close OOD), MNIST random digit plus noise (close-ish), and white-noise. White noise is certainly not close to anything (including the training data), and from Figure 2 we can observe that $H_{\sf ep}$ does a good job of distinguishing that it is *more uncertain*--in particular, $H_{\sf ep}$(right) does a better job here than $H_{\sf tot}$(left), which attributes a similar level of uncertainty to white-noise and digit 8. The purpose of this exercise is to illustrate the distinction between close OOD and far OOD, and the relevance of $H_{\sf ep}$ in discriminating between them. Please note that Figure 2 is from the *baseline predictive model*, so the models have not seen any OOD data beforehand. What is plotted in Figure 2 are 2 of the 7 features which are used to train the meta-classifier. The meta-classifier *does* see (features of) examples from these 4 OOD categories, and learns to classify them all as 1 (abstain).
>
> We now show AU2LC from Table 4 for each OOD categories independently, in Table 8 of Appendix G.1.1, to observe the difference. We also include an example of further OOD data (CIFAR). This far-OOD data from a fundamentally distinct DGP is now included in Table 3 and 4 of the revision.
>
> We have updated the results to include super far-OOD examples from a completely different DGP in the unseen test set. We were pleased to see that our method performs very well there (as predicted from Figure 2). The Table below shows the metrics for MNIST, associated to a *500 ID and 500 CIFAR* test data, where the meta-classifier was trained on ID and the 4 original near/far OOD classes (and no CIFAR data whatsoever, i.e. it is from a fundamentally distinct DGP). This is also shown now in Table 13 in Appendix G.1.1 of the revision.
>
> ### Table 3. Metrics associated to 500 ID + 500 CIFAR (new unseen DGP OOD)
>
> | P | Method             | Precision | Recall | F1 | AUC-ROC |
> |-------|--------------------|-----------|--------|----|---------|
> | –     | MAP                | 0.743     | 0.906  | 0.812 | 0.835 |
> | –     | DE                 | 0.922     | 0.918  | 0.919 | 0.970 |
> | 1     | S-SMC$_\parallel$  | 0.882     | 0.885  | 0.883 | 0.934 |
> | 8     | S-SMC$_\parallel$  | 0.979     | **0.938** | **0.958** | **0.991** |
> | 1     | S-HMC$_\parallel$  | 0.970     | 0.924  | 0.946 | **0.988** |
> | 8     | S-HMC$_\parallel$  | **0.990** | 0.923  | **0.955** | **0.990** |
>
> The results in Tables 3 and 4 of the revision now include 500 unseen test examples from each of the 4 original OOD categories (training DGP) and 500 CIFAR examples (fundamentally distinct DGP), along with 2500 unseen ID test examples.
>
> We are thinking about how to emphasize this robustness result more in the final version. It currently appears in the Appendix.

---

> ### Comment · Reviewer_otV6 · 2025-11-28
>
> I would like to thank the authors for their detailed engagement during the rebuttal and the significant effort put into running additional experiments. Having reviewed the revised manuscript and responses, I plan on raising my score to a 4, though I retain some reservations regarding the methodological novelty (major) and baseline configurations (minor) that prevent me from recommending acceptance at this stage.
>
> My primary concern remains the limited novelty, as the framework appears to largely repackage established concepts rather than introducing a fundamental shift. The paper covers a wide breadth of topics including heuristics, scale, consistency, and OOD detection, but consequently feels somewhat cluttered and occasionally lacks depth in investigating each component individually (in my opinion most notably the OOD part). Additionally, I remain skeptical regarding the fairness of the DEI-HMC comparison. The comparison would have been more robust had more hyperparameter-insensitive algorithms, such as NUTS or MCLMC, or likely even more importantly established window adaptation schemes like those implemented in STAN for HMC type samplers, been employed. Furthermore, setting the number of leapfrog steps to $L=1$ is a very unusual choice; even the literature cited in your response ([**]) argues that HMC in BNNs typically requires a high number of leapfrog steps. While I realize this recipe was also followed for S-HMC, potentially for computational reasons, it is not an established configuration.
>
> I also have remaining concerns regarding convergence and the OOD approach. Thank you for the clarifications on functional $\hat{R}$; however, if you would suggest focusing on functional convergence, this should likely result in a revision of Figure 1d. Regarding the two-stage estimator, I am not fully convinced by the utility of relying on meta-OOD data compared to methods that explicitly model in-distribution data, such as Natural Posterior Networks (which fit a normalizing flow to the latent space). The former approach seems less practically useful, and I find the terminology of "close" vs. "far" OOD somewhat unusual. While the new CIFAR results are encouraging, the additional OOD experiments do not fully resolve my concerns regarding the utility of BMC in the OOD context and the supervised meta-classifier approach itself. Finally, please note there are still a few formatting errors in the revision (e.g., around line 1429). Thank you again for the clarifications.

---

> > ### Author Response · Authors · 2025-12-01
> > **Final rebuttal, pt. 1**
> >
> > > I would like to thank the authors for their detailed engagement during the rebuttal and the significant effort put into running additional experiments. Having reviewed the revised manuscript and responses, I plan on raising my score to a 4, though *I retain some reservations regarding the methodological novelty (major) and baseline configurations (minor) that prevent me from recommending acceptance at this stage.*
> >
> > We would like to thank the reviewer for raising your score, and clearly articulating the remaining points which prevent you from recommending acceptance. We will very carefully address each of these below, along with further supporting evidence, which we hope will finally tip the scales in our favor.
> >
> >
> >
> > > My primary concern remains the limited novelty, as the framework appears to largely repackage established concepts rather than introducing a fundamental shift. The paper covers a wide breadth of topics including heuristics, scale, consistency, and OOD detection, but consequently feels somewhat cluttered and occasionally lacks depth in investigating each component individually (in my opinion most notably the OOD part)...
> > > I also have remaining concerns regarding convergence and the OOD approach. Regarding the two-stage estimator, I am not fully convinced by the utility of relying on meta-OOD data compared to methods that explicitly model in-distribution data, such as Natural Posterior Networks (which fit a normalizing flow to the latent space). The former approach seems less practically useful, and I find the terminology of "close" vs. "far" OOD somewhat unusual. While the new CIFAR results are encouraging, the additional OOD experiments do not fully resolve my concerns regarding the utility of BMC in the OOD context and the supervised meta-classifier approach itself. Finally, please note there are still a few formatting errors in the revision (e.g., around line 1429). Thank you again for the clarifications.
> >
> > Thank you for reiterating your concerns regarding novelty. We do not claim that any individual piece of our work is a fundamental shift, yet we also do not consider it to be a simple repackaging of established ideas. The novelty of our work lies in the integration of existing ideas and approaches in a novel way towards the unified end goal of reliable and achievable UQ, and hence confident decision making, with a realistic budget. It is represents the *operationalization of deep expertise and thorough theoretical and empirical investigation*, which is the reason why we robustly achieve improved performance over existing Bayesian deep learning methods.
> >
> > We will replace the unusual terminology as follows:
> > - near-OOD: "similar to or overlapping the training DGP"
> > - far-OOD: "completely distinct from the training DGP"
> > - super-far-OOD: "completely distinct from the meta-training DGP"
> >
> > We appreciate your concerns regarding 1-stage and 2-stage estimators. We would like to clarify that our method *does not rely on OOD data or the meta-classifier to model in-distribution data.* The modelling of in-distribution behaviour is handled directly and intrinsically by the posterior predictive distribution, which is a property of the Bayesian model itself. In contrast, methods such as PostNet and NatPN introduce surrogate latent density models (e.g. normalizing flows) to approximate uncertainty. These approaches can be effective but differ conceptually from ours: they model density in latent space, whereas we approximate the full posterior predictive. *We agree that including a direct experimental comparison against these methods will further clarify this distinction and the relative merit of the appeoaches, and we will add it in the revision $(\dagger)$.*
> >
> > The 2-level estimator for abstention decisions relies on the 2-stage meta-classifier. It is important to note that any abstention rule or decision-making mechanism requires calibration or validation data. This is true for our methods as well as PostNet, NatPN, and other single-stage scoring approaches. A single-stage estimator (e.g. $H_{\sf ep}$, maximum class probability, or Dirichlet evidence) can effectively rank examples, but it does not itself define a *decision threshold*.
> >
> > PostNet and NatPN, for instance, focus on defining informative uncertainty scores but do not provide decision-level mechanisms derived from those scores. In our experiments, we observe that a strong score does not automatically translate into a reliable decision rule unless an additional calibration stage is applied. Our two-stage procedure enables this transition—from uncertainty scoring to robust Bayesian decision-making—while preserving the underlying posterior predictive information.
> >
> > We have done some further experiments in order to address the remaining concerns regarding the two-stage approach below.

---

> ### Author Response · Authors · 2025-12-01
> **Final rebuttal, part 2.**
>
> First, in Table 4, we show the AUCROC (score power) of several individual *single-stage* predictive scores derived from the original SBMC ensemble in comparison to our two-stage meta-classifier. From this, we can see that $H_{\sf ep}$ is a powerful 1-stage score overall, on par with the 2-stage meta-classifier $p_{\sf abst}$ score in terms of its ability to rank abstention examples (OOD/incorrect) higher than examples with correct predictions. We would also like to emphasize again the results in Table 3 above, and note that the final column shows AUCROC, a *single-stage metric associated to the score $H_{\sf ep}$ itself*, which shows that it performs exceptionally well specifically at ranking the examples from this distinct DGP higher than those from the ID DGP of the training data.
>
>
> ### Table 4. AUCROC: 2-stage (Meta-classifier) and 1-stage scores. MNIST7. $N=10$.
>
> | $P$ | Method             | Meta | H$_{\text{ep}}$ | p$_{\text{max}}$ | H$_{\sf tot}$ |
> |------|--------------------|--------------|------------------|-------------------|----------------|
> | -    | MAP                | 0.864        | 0.500            | 0.861             | 0.865          |
> | 1    | DE                 | 0.926        | 0.929            | 0.881             | 0.899          |
> | 1    | S-SMC$_\parallel$ | 0.895        | 0.898            | 0.853             | 0.878          |
> | 8    | S-SMC$_\parallel$ | **0.940**        | **0.940**            | 0.867             | 0.928          |
> | 1    | S-HMC$_\parallel$ | **0.932**        | **0.932**            | 0.867             | 0.926          |
> | 8    | S-HMC$_\parallel$ | **0.942**        | **0.944**            | 0.863             | **0.932**          |
>
>
> Next, we would like to refer again to Table 2, reproduced again below for convenience. Here we plot the Area under the 2-level estimator curve (AU2LC), which depends on the abstention rule learned at the second stage, from the meta-training data which includes OOD calibration data (the test set includes also meta-OOD data for robustness). We can see that again $H_{\sf ep}$ performs comparably to the full meta-classifier for the best cases (very good epistemic UQ), whereas for the cases where the epistemic quality falls short (DE and S-SMC with P=1) the meta-classifier provides a significant uplift in performance. Therefore, we can understand the meta-classifier as a mechanism which leverages the epistemic uncertainty information and also adds some robustness to the final decision rule.
>
>
> ### Table 2. Area under the 2-level estimator curve Ablations for MNIST7. $N=10$.
>
>
> | P | Method             | All fea. | Ep. fea. | ${\bf p_{\sf max}}$ | ${\bf H_{\sf{ep}}}$ |
> |-------|--------------------|--------------|--------------------|-------------------|------------------|
> | -     | MAP                | 0.717        | 0.501              | 0.701             | 0.501            |
> | 1     | DE                 | 0.795        | 0.790              | 0.734             | 0.789            |
> | 1     | S-SMC$_\parallel$  | 0.747        | 0.738              | 0.690             | 0.738            |
> | 8     | S-SMC$_\parallel$  | **0.820**    | **0.820**          | 0.718             | **0.818**        |
> | 1     | S-HMC$_\parallel$  | 0.804        | 0.801              | 0.712             | 0.798            |
> | 8     | S-HMC$_\parallel$  | **0.822**    | **0.822**          | 0.716             | **0.819**        |

---

> ### Author Response · Authors · 2025-12-01
> **Final Rebuttal, part 3**
>
> Finally, in Table 5, we demonstrate a possible underlying mechanism for this phenomenon, with another table showing AUCROC for the meta-classifier score (2-stage) and the $H_{\sf ep}$ score (1-stage) on 2 isolated datasets:
> - (i) the digit 9, which is somewhat similar to the training DGP (MNIST 1,...7) but still OOD.
> - (ii) CIFAR10, which is fundamentally distinct from both the training DGP and the meta-training DGP (for 2-stage).
>
> Note that we observe the *same thing* as in Table 2 above for the similar but OOD case (digit 9), where the meta-classifier performs better when $H_{\sf ep}$ falls short. Whereas, for the fundamentally distinct OOD case (CIFAR), $H_{\sf ep}$ alone is sufficient. This further emphasizes the added robustness delivered by the meta-classifier, even at the level of the score alone before any decision is made.
>
>
> ### Table 5. AUCROC for H$_{\sf ep}$ (1-stage) and full meta-classifier (2-stage) for digit 9 (similar but OOD) and CIFAR (fundamentally distinct OOD) categories. MNIST7. $N=10$.
>
> |     |       | **Digit 9**         |         |  | **CIFAR10**         |          |
> |-----|-------|---------------------|----|---------|---------------------|----------|
> | **P**   | **Method** | **All fea.**       | **H$_{\sf ep}$** | |**All feat.**      | **H$_{\sf ep}$** |
> | -   | MAP               | 0.809 | 0.500 | | 0.835 | 0.500 |
> | 1   | DE                | 0.807 | 0.800 | | 0.970 | 0.973 |
> | 1   | S-SMC$_\parallel$ | 0.816 |  0.808 | | 0.934 | 0.939 |
> | 8   | S-SMC$_\parallel$ | **0.831** |  **0.831** | | **0.991** | **0.991** |
> | 1   | S-HMC$_\parallel$ | 0.814 |  0.806 | | **0.988** | **0.988** |
> | 8   | S-HMC$_\parallel$ | 0.816 |  0.818 | | **0.990** | **0.991** |
>
>
> > Additionally, I remain skeptical regarding the fairness of the DEI-HMC comparison. The comparison would have been more robust had more hyperparameter-insensitive algorithms, such as NUTS or MCLMC, or likely even more importantly established window adaptation schemes like those implemented in STAN for HMC type samplers, been employed. Furthermore, setting the number of leapfrog steps to L=1 is a very unusual choice; even the literature cited in your response ([**]) argues that HMC in BNNs typically requires a high number of leapfrog steps. While I realize this recipe was also followed for S-HMC, potentially for computational reasons, it is not an established configuration.
>
> We chose these settings (including L=1) for S-HMC to match the kernel used in S-SMC, which also involves adaptive tempering and cannot afford elaborate adaptation at every stage.
>
> We will run a comparison using NUTS and other window adaptation schemes for both DEI-HMC and S-HMC, by splitting the budget in half and allocating half to adaptation. We do not expect any conclusions to change because chains targeting the full posterior will barely decorrelate within the very short windows we are constrained to. This is a challenge for adaptation itself since it will be constrained to an even shorter adaptation window. When it comes to the full target distribition, the regime we are considering is very much far-from-convergence. *These results will be included as soon as they are ready $(\dagger)$.*
>
>
> > Thank you for the clarifications on functional $\hat{R}$; however, if you would suggest focusing on functional convergence, this should likely result in a revision of Figure 1d.
>
> *We will update Figure 1d as requested $(\dagger)$.*

---

### Official Review · Reviewer_WB64 · 2025-10-30

**Soundness:** 2
**Presentation:** 3
**Contribution:** 2
**Rating:** 4
**Confidence:** 4

**Summary:**

The paper introduces a new Bayesian deep learning framework that bridges the gap between computationally expensive Bayesian Monte Carlo (BMC) samplers and fast but heuristic methods like deep ensembles. The core idea is an approximate model that interpolates between a maximum a posteriori (MAP) point estimator and the full Bayesian posterior via a scalar parameter $s$, allowing users to trade off between computational cost and uncertainty accuracy. Combined with parallel implementations of MCMC and SMC algorithms, SBMC delivers near-linear scalability and strong uncertainty quantification (UQ), particularly epistemic uncertainty, at a cost comparable to deep ensembles. Experiments on datasets demonstrate that SBMC achieves comparable accuracy but significantly better UQ and reliability, with applications such as confidence-based abstention.

**Strengths:**

1.	**Practical Motivation and Design:** The paper addresses a clear problem: scaling Bayesian inference to deep nets to obtain well-calibrated uncertainty. The SBMC model is intuitively motivated as interpolating between a cheap point estimate and the full posterior. This interpolation idea is simple and flexible: by tuning $s$ one can trade off bias vs. sampling difficulty. The algorithmic idea of running multiple short MCMC/SMC chains in parallel is straightforward and leverages modern parallel hardware.

2.	**Comprehensive Experiments:** The evaluation is extensive. The authors compare SBMC against several relevant baselines (MAP, SWA, MC Dropout, Laplace, Deep Ensembles) under a fixed compute budget (measured in epochs) on MNIST7, IMDb, and CIFAR10. They report accuracy, NLL, and epistemic entropy metrics. They also include “gold-standard” HMC runs (long chains) to estimate ground-truth uncertainty. The results consistently show that anchor+parallel BMC achieves accuracy comparable to strong baselines while yielding higher epistemic uncertainty on errors and out-of-domain inputs.

3.	**Clarity of Presentation:** The paper clearly lays out the method, with Algorithm boxes for SBMC (and underlying SMC, MCMC routines). The interpolation model (Gaussian prior mean = $\theta_{MAP}$, covariance ∝ $s$) is explicitly defined.

**Weaknesses:**

1.	**Limited Novelty / Relation to Prior Methods:** The core idea (anchoring the posterior at a point estimate and running multiple short MCMC chains) is conceptually similar to known techniques. The paper cites Randomized Maximum Likelihood (RML) and ensemble anchoring methods (e.g. Gu & Oliver 2007; Bardsley et al. 2014). While SBMC’s scalar interpolation $s$ is a convenient formalism, the idea of tempering/anchoring (even mentioning “cold posteriors”) is not fundamentally new. Indeed, Paulin et al. (2024) previously anchored to SWA and ran multiple Langevin chains; this work’s difference (anchoring to MAP instead of SWA mean, using HMC/SMC) seems incremental. The contributions list claims a “new SBMC method” and “thorough evaluation”, but the method is essentially a heuristic interpolation rather than a novel algorithmic insight.

2.	**Approximation Bias / Lack of Guarantees:** By construction, SBMC does not target the true Bayesian posterior except at $s=1$. For $0<s<1$ the estimator is biased. The authors acknowledge this (“since the method no longer targets the posterior for any $s<1$, we adopt a heuristic approach”), but they offer no theoretical analysis of the bias or uncertainty error introduced by $s<1$. The only justification is empirical: they show SBMC’s UQ looks sensible on their tasks. However, without any formal bounds or diagnostics, it is hard to trust SBMC’s uncertainty calibration in general. In fact, SBMC requires careful tuning of both $s$ and the prior variance $v$, as noted: small $s$ makes mixing easier but “introduces bias because $\pi_s \neq \pi$”. The paper’s advice to set $s=0.1$ as a default (with cross-validation for $v$) feels heuristic.


3.	**SWAG / More Baselines:** The authors reference SWAG (SWA-Gaussian) as a popular scalable Bayesian method, but they do not include SWAG results. Without comparing to SWAG or other modern UQ methods (e.g. KFAC-Laplace, SGD-based posteriors, deep Gaussian processes), it is unclear how SBMC stacks up.

4.	**Scale of Models:** The neural networks used (e.g. “architecture in Appendix E”) seem relatively small. There is no large-scale experiment (e.g. a full ResNet on CIFAR, or an NLP model beyond small GPT-2 finetuning). Claims of scalability are not validated on truly large networks where MCMC is most challenging.

5.	**Compute Budget Accounting:** The comparisons assume “equal wall-clock time” by measuring epochs. However, SBMC needs to compute the MAP (another pass of SGD) plus run parallel chains. In Table 1 the “Total Cost” for SBMC is roughly double an SMC baseline (e.g. SBMC (1 chain) uses 330 epochs vs. MAP’s 160). The authors note SBMC’s total time is roughly double due to MAP computation (Sec. 3.1), but in claiming “same wall-clock as DE” this overhead must be justified. It seems SBMC is only comparable if parallel resources make the MAP step negligible.

6.	**Evaluation Metrics:** The focus is on epistemic entropy as the UQ metric. While reasonable, this is somewhat non-standard. Calibration error (ECE/Brier) or negative log-likelihood on a held-out test set (beyond MNIST NLL) are not extensively reported.

**Questions:**

1.	**Comparison to SWAG and other baselines:** Why were methods like SWAG or deep Laplace (e.g. KFAC-Laplace) not included in the experiments? SWAG, in particular, is known to improve uncertainty on CIFAR with modest cost.

2.	**Hyperparameter Sensitivity:** You recommend $s=0.1$ by default and note the prior variance $v$ must be tuned. How sensitive are results to these choices across tasks?

3.	**Scale and Efficiency:** Can SBMC handle larger-scale networks and datasets? For example, have you tried SBMC on a full CNN (e.g. ResNet) for CIFAR10, or on language models beyond the tiny GPT-2 setting? Also, how does SBMC’s actual GPU/runtime compare to DE in wall-clock seconds, given the extra MAP pass?

4.	**Multi-modal Posteriors:** In problems where the posterior is highly multi-modal, anchoring to a single MAP might be limiting. Have you considered initializing parallel chains at different modes (e.g. using multiple MAPs) to better explore multiple modes?

5.	**Theoretical Guarantees:** Do you have any theoretical insight into how the bias $E_{\pi_s}[φ] - E_\pi[φ]$ depends on $s$? For example, can you bound the error in expectations as a function of $s$ or relate it to tempered posteriors? Without such guarantees, how should a practitioner choose $s$ to ensure reliability?

**Details Of Ethics Concerns:**

No ethical issues found.

---

> ### Author Response · Authors · 2025-11-22
> **Rebuttal**
>
> Thank you very much for carefully reading our paper, for recognising its strengths, and for providing us with valuable and actionable recommendations for improvement. We have processed these carefully and taken all suggestions for improvement into account, as detailed below.
>
>
> ## Weaknesses
>
> > **W1. Limited Novelty / Relation to Prior Methods**: The core idea (anchoring the posterior at a point estimate and running multiple short MCMC chains) is conceptually similar to known techniques. The paper cites Randomized Maximum Likelihood (RML) and ensemble anchoring methods (e.g. Gu & Oliver 2007; Bardsley et al. 2014). While SBMC’s scalar interpolation  is a convenient formalism, the idea of tempering/anchoring (even mentioning “cold posteriors”) is not fundamentally new. Indeed, Paulin et al. (2024) previously anchored to SWA and ran multiple Langevin chains; this work’s difference (anchoring to MAP instead of SWA mean, using HMC/SMC) seems incremental. The contributions list claims a “new SBMC method” and “thorough evaluation”, but the method is essentially a heuristic interpolation rather than a novel algorithmic insight.
>
> We do not claim that the SBMC method introduces the idea of anchoring. The particular method introduced here is distinct from existing methods that do similar things, and it cheaply delivers good performance on first order metrics like accuracy, NLL, ECE, Brier, and improved estimates of epistemic uncertainty via the ensemble of samples. While it is indeed largely heuristic, we do provide a reason for the improved performance and a sketch of the underlying theory in Appendix C, along with a lot of empirical evidence. We moved Appendix C to Sec 3.2 in the main text. See also Novelty and Table 1 in the Global rebuttal, and responses to otV6 (W1,Q1-2) below.
>
> > **W2. Approximation Bias / Lack of Guarantees**: By construction, SBMC does not target the true Bayesian posterior except at $s=1$. For $0<s<1$ the estimator is biased. The authors acknowledge this (“since the method no longer targets the posterior for any $s<1$, we adopt a heuristic approach”), but they offer no theoretical analysis of the bias or uncertainty error introduced by $s<1$. The only justification is empirical: they show SBMC’s UQ looks sensible on their tasks. However, without any formal bounds or diagnostics, it is hard to trust SBMC’s uncertainty calibration in general. In fact, SBMC requires careful tuning of both $s$ and the prior variance $v$, as noted: small $s$ makes mixing easier but “introduces bias because $\pi_s \neq \pi$”. The paper’s advice to set $s$ as a default (with cross-validation for $v$) feels heuristic.
>
> It is accurate that the method is heuristic. The prior variance $v$ needs to be chosen for any Bayesian method (as does the weight decay for frequentist inference). As a default, we find that $v=500/N$ is reasonable, corresponding to the standard choice of weight decay $\lambda=1e-3$. However, from our experience, performance tends to improve as $v$ increases and the prior predictive tends to a sum of Diracs in the corners so that the prior predictive uncertainty is purely epistemic uncertainty. See also Figure 9 of [**].
>
> In Appendix C we provided an explanation for the good performance and sketch a theoretical argument for this and the default choice of $s=0.1$. We have moved this to Sec. 3.2. See also the response to Q5 below, and Theory in the Global rebuttal.
>
> > The only justification is empirical: they show SBMC’s UQ looks sensible on their tasks
>
> Beyond the sketch of the theory mentioned above, we provide empirical evidence. However, would like to rebut here that we compare with "gold-standard" long HMC runs (e.g. in Tables 1 and 8), which is accepted by the community as a valid metric for performance. We would argue this is more than "looks sensible". The improved performance on downstream confidence prediction is also consistent with this line of reasoning.
>
>
>
>
> > **W3. SWAG / More Baselines**: The authors reference SWAG (SWA-Gaussian) as a popular scalable Bayesian method, but they do not include SWAG results. Without comparing to SWAG or other modern UQ methods (e.g. KFAC-Laplace, SGD-based posteriors, deep Gaussian processes), it is unclear how SBMC stacks up.
>
> See the response to Q1 below. Laplace in Table 1 is actually KFAC-Laplace and we will update the label to be more clear. SG-HMC is stochastic-gradient HMC, an SGD-based posterior. We will also make that more clear. We have added SWAG in Table 1 in the General Rebuttal, and it also underperforms SBMC. Deep GP [*] do not scale well to large models and datasets and will not work on the problems we look at here.
>
> [*] Damianou, A., & Lawrence, N. D. (2013). Deep gaussian processes. In AISTATS (207-215). PMLR

---

> ### Author Response · Authors · 2025-11-22
> **Rebuttal, continued (1)**
>
> > **W4. Scale of Models**: The neural networks used (e.g. “architecture in Appendix E”) seem relatively small. There is no large-scale experiment (e.g. a full ResNet on CIFAR, or an NLP model beyond small GPT-2 finetuning). Claims of scalability are not validated on truly large networks where MCMC is most challenging.
>
> The GPT2 model we consider has $1e8$ total parameters, and $2e5$ are trainable in our small example. The dataset size is $3e5$ tokens. MCMC is already extremely challenging at this scale (indeed some would say it is quite impossible). See also the response to Q3 below. *Full ResNet-20 CNN CIFAR experiments are in progress and will be added $(\dagger)$*.
>
>
> > **W5. Compute Budget Accounting**: The comparisons assume “equal wall-clock time” by measuring epochs. However, SBMC needs to compute the MAP (another pass of SGD) plus run parallel chains. In Table 1 the “Total Cost” for SBMC is roughly double an SMC baseline (e.g. SBMC (1 chain) uses 330 epochs vs. MAP’s 160). The authors note SBMC’s total time is roughly double due to MAP computation (Sec. 3.1), but in claiming “same wall-clock as DE” this overhead must be justified. It seems SBMC is only comparable if parallel resources make the MAP step negligible.
>
> This is correct and fair and we have corrected the statement from "same" to "twice". To further clarify this we added experiments under matched wall-clock budgets, where SBMC runs with half the sampling epochs (Global rebuttal Table 1 S-HMC(1/2x)) and DE runs with double its epochs (DE(2x)). The results are comparable and conclusions do not change, and they have been included also in Table 1 in the revision. See also the response to Q3 below, the Global response above, and the response to H5Gj W2.
>
> > **W6. Evaluation Metrics**: The focus is on epistemic entropy as the UQ metric. While reasonable, this is somewhat non-standard. Calibration error (ECE/Brier) or negative log-likelihood on a held-out test set (beyond MNIST NLL) are not extensively reported.
>
> *ECE, Brier, and NLL on held-out test data were computed for all cases*. We have articulated this more clearly and provided the appropriate cross-references to the results. These results are reported in the Appendix Table 8 (17 in the revision) for MNIST, and Figure 1(c) and 3 for IMDb and CIFAR (in Figure 3 it was labelled “NL” rather than NLL, and has been updated). See also Appendix Tables 18-20 and the Figures therein (and supplementary materials) for further results on IMDb and CIFAR. We will also change Figure 5 (now 6) to show NLL instead of Accuracy -- the story is essentially the same, and the results are presented in the Appendix.
>
> We agree that it is somewhat non-standard to focus on epistemic entropy, and there is a well-thought-out rationale behind it. We argue that epistemic uncertainty is valuable and interesting beyond what is standard, and moreover that *it should be a primary focus of UQ and Bayesian methods in general*. The reason for this is that it is characterised by the uncertainty in the network weights themselves. That is the real novelty we get with ensemble-based methods, and that is why we focus on it here. We also demonstrate that features which characterise this epistemic uncertainty are discriminative of the accuracy in model outputs, and hence useful for estimating model confidence. This is shown in Figure 2 and the new Table 4.
>
> For a simple demonstration of this, consider binary classification problems with posterior probabilities $\pi(p) = \delta_{1/2}$ and $\pi(p) = \frac12(\delta_0 + \delta_1)$. Both have predictive estimator $\mathbb{E}(p)=1/2$, but the former is all alteatoric and the latter is all epistemic. Here we should clearly abstain in either case, but the point is that epistemic uncertainty is a rich source of information and a good source of features for predicting model confidence, beyond what any method can do with only the posterior predictive. We have also described this in the new Figure 5, and illustrated with ablations in Table 4, and discussion there.

---

> ### Author Response · Authors · 2025-11-22
> **Rebuttal, continued (2)**
>
> ## Questions
>
> > **Q1. Comparison to SWAG and other baselines**: Why were methods like SWAG or deep Laplace (e.g. KFAC-Laplace) not included in the experiments? SWAG, in particular, is known to improve uncertainty on CIFAR with modest cost.
>
> Our Laplace implementation is KFAC-Laplace, and we will make this more clear by referring to it more precisely. SWAG has been included and it also underperforms SBMC. We did not include it originally simply because we already have comparisons with KFAC-Laplace, SWA, DE, MC Dropout, and SG-HMC, which are the main competitors.
>
> > **Q2. Hyperparameter Sensitivity**: You recommend s=0.1 by default and note the prior variance v must be tuned. How sensitive are results to these choices across tasks?
>
> The results are relatively insensitive for small values of $s \in [0.05,0.3]$. This is stated in line 365 in the paper and shown in column 1 of Figure 5 (submitted, or line 439 and Figure 6, revised). Performance degrades significantly as $s \rightarrow 1$. We provided a sketch of a theoretical argument for the choice $s=0.1$ in Appendix C, which is now in Sec 3.2 in the main text. See also the response to (Q5) below.
>
> The prior variance $v$ is a nuisance parameter across all methods, and in a Bayesian context this has been considered in previous works -- see e.g. Figure 9 of [**]. As shown there, the results are not that sensitive to these parameters, and performance improves as $v$ increases. We find that the common choice of weight decay $\lambda \approx 10^{-3}$ is suitable, i.e. $v=500/N$. See also the response to (W2) above.
>
> > **Q3. Scale and Efficiency**: Can SBMC handle larger-scale networks and datasets? For example, have you tried SBMC on a full CNN (e.g. ResNet) for CIFAR10, or on language models beyond the tiny GPT-2 setting? Also, how does SBMC’s actual GPU/runtime compare to DE in wall-clock seconds, given the extra MAP pass?
>
> Yes, we have found good performance here. *A full ResNet-20 model on CIFAR10 experiment is running and will be included in the rebuttal $(\dagger)$*. SBMC costs approximately twice that of DE, due to the extra MAP pass.
>
> When we say comparable wall-clock time, we meant that it is comparable in comparison to 100x or 1000x time required to get converged BMC results. We will revise the presentation to be more clear and precise everywhere time cost is mentioned. We also included experiments with half as many epochs for SBMC (Global rebuttal, Table 1, S-HMC(1/2x)) and twice as many epochs for MAP/DE (2x), for exactly even budgets.
>
> Finally, we note that GPT has 124M parameters and $2e5$ trainable parameters, and we fine-tuned on $3e5$ tokens. While this is indeed tiny relative to modern LLMs, nobody touches modern LLMs with Bayesian methods. One thing worth emphasizing is the *parameter-efficiency of SBMC: the cost of running our ensemble of 10 models adds an overhead of $<2\%$ on top of a single GPT2 run, which will become increasingly valuable at scale.*
>
> > **Q4. Multi-modal Posteriors**: In problems where the posterior is highly multi-modal, anchoring to a single MAP might be limiting. Have you considered initializing parallel chains at different modes (e.g. using multiple MAPs) to better explore multiple modes?
>
> Yes, we have considered this. In fact, the HMC results use different realizations from SGD (hence, ostensibly, different modes), and we will clarify that in the revision by adding an index to $\overline{\pi}^{(i)}_0$ in Algorithm 3. In Table 1 of the Global rebuttal, we have compared this to using a single MAP to initialize all chains, as originally described in Algorithm 3. The performance using a single MAP is similar to using different initializations (or even a little bit better, unexpectedly).
>
> > **Q5. Theoretical Guarantees**: Do you have any theoretical insight into how the bias $E_{\pi_s}(\varphi) - E_{\pi}(\varphi)$ depends on $s$ ? For example, can you bound the error in expectations as a function of $s$ or relate it to tempered posteriors? Without such guarantees, how should a practitioner choose $s$ to ensure reliability?
>
> To make sense of the approximation, one can leverage a sequential pseudo-Bayesian interpretation, and we sketched this out in Appendix C. The argument also justifies the choice $s=0.1$, given that the scaling factor $a$ is arbitrary.
>
> The approximation argument could be improved and tightened up by using a (KFAC-)Laplace approximation of the scaled/tempered posterior as a prior (instead or) as an intermediate approximation. We do not believe this would provide significant insight or practical benefit, and so it has been omitted. For small perturbations around the limits, we may be able to obtain more precise and rigorous estimates, but again we believe the practical value would be limited.
>
>
> [**] Izmailov, P., Vikram, S., Hoffman, M. D., & Wilson, A. G. (2021). What Are Bayesian Neural Network Posteriors Really Like?. arXiv preprint arXiv:2104.14421. -- arxiv version of ICML (pp. 4629-4640). PMLR.

---

### Official Review · Reviewer_H5Gj · 2025-10-30

**Soundness:** 3
**Presentation:** 2
**Contribution:** 3
**Rating:** 6
**Confidence:** 4

**Summary:**

This paper presents Scalable Bayesian Monte Carlo (SBMC) method to balance computational efficiency with high-quality uncertainty quantification. The approach combines an interpolated model by combining a point estimate and a full Bayesian treatment with parallelized Monte Carlo algorithms, such as sequential Monte Carlo and Markov chain Monte Carlo. The authors showed that SBMC delivers competitive or better accuracy while significantly improving epistemic uncertainty compared with baselines such as deep ensembles across multiple benchmarks.

**Strengths:**

The paper is well-written, and the empirical results are presented clearly. Below are the strengths of the presented method:

1. SBMC introduces a model approximation which is anchored posterior that uses a scalar interpolation parameter to tune the trade-off between the fast Maximum A Posteriori estimator and the full Bayesian posterior.

2. The method is highly scalable due to the parallel implementation of consistent Bayesian Monte Carlo algorithms and achieved near linear speed-up.

3. Empirical studies show that SBMC achieves comparable or better accuracy and substantially improved epistemic uncertainty quantification versus Deep Ensembles at similar wall-clock time.

4. SBMC also showed good performance in estimating prediction confidence where a meta-classifier based on SBMC posterior improves detection of incorrect/out-of-domain data.

**Weaknesses:**

I think the paper can be improved by addressing the following weak points:

1. The paper relies on empirical tuning for the scalar interpolation parameter s and provides no theoretical analysis linking s to the difference anchored posterior and true posteriors.

2. SBMC requires computing a good Maximum A Posteriori to act as the anchor before parallel Bayesian monte Carlo sampling and it is seen in the MNIST7 experiment that the total cost is roughly doubled compared deep ensembles method due to this prerequisite step.

3. The author used a discontinuous step for the parameter $\alpha(s)$ in equation 2 without providing good justification for these choices over continuous alternatives.

4. A comprehensive hyperparameter table is missing to provide detailed implementation details and a graphical schematic for meta-classifier pipeline is missing.

**Questions:**

1. Why is $\alpha(s)$ defined as a step function with $\alpha(s)=1$ for $s<1/2$ and 0 otherwise? Would a smooth interpolation such as linear $\alpha(s)=1-s$ can produce different results, and how sensitive are the results to this choice?

2. For the meta-classifier, how many posterior samples were used to compute expectations and variances of $p_{max}(x, \theta)$ p_max (x,θ) and $\Delta_{max}(x, \theta)$? How is total entropy estimated for MAP and deep ensemble models that produce deterministic predictions?

3. Appendix 5.1 mentions preliminary results on GPT‑2; can you provide more details on applying SBMC to language models? What challenges may arise when scaling to tens of millions of parameters?

---

> ### Author Response · Authors · 2025-11-22
> **Rebuttal**
>
> Thank you very much for carefully reading our paper, for recognising its strengths, and for providing us with valuable and actionable recommendations for improvement. We have processed these carefully and taken all suggestions for improvement into account, as detailed below.
>
>
> ## Weaknesses
>
> > **W1**. The paper relies on empirical tuning for the scalar interpolation parameter s and provides no theoretical analysis linking s to the difference anchored posterior and true posteriors.
>
> We provided a sketch of the theory in Appendix C, following a sequential pseudo-Bayesian decomposition and update. This has been moved into the main text in the revision.
>
> > **W2**. SBMC requires computing a good Maximum A Posteriori to act as the anchor before parallel Bayesian monte Carlo sampling and it is seen in the MNIST7 experiment that the total cost is roughly doubled compared deep ensembles method due to this prerequisite step.
>
> This is true, and we have (a) revised the statements to be more precise, and (b) included experiments with twice as many epochs for MAP/DE (Global rebuttal Table 1, MAP/DE(2x)) and half as many for SBMC (Global rebuttal Table 1, S-HMC(1/2x)) in order to illustrate that results are comparable and conclusions are the same for genuinely equal time cost. Our rationale for the original presentation is that even a factor 2x is "comparable", if not negligible, in comparison to a factor of 100x or 1000x for full BMC.
>
> > **W3**. The author used a discontinuous step for the parameter $s$ in equation 2 without providing good justification for these choices over continuous alternatives.
>
> See Table 1 "S-HMC(linear)" in the Global rebuttal and response to Q1 below for discussion of this. We evaluated linear interpolation there and found it underperforms MAP-centered anchoring, particularly in epistemic separation. Our findings indicate that performance is most favorable in the small-s regime where the prior mean remains fixed at the MAP. There is also a sketch of a theoretical reason for this choice in Appendix C (moved to the end of Sec 3). We will clarify these conclusions in the revision.
>
>
> > **W4**. A comprehensive hyperparameter table is missing to provide detailed implementation details and a graphical schematic for meta-classifier pipeline is missing.
>
> We have included a comprehensive graphical schematic for the meta-classifier pipeline in a new Figure 3 in the revised manuscript (the additional figure is the *only* revision currently). The hyper-parameters are currently given in Appendix D.3, and new comprehensive hyper-parameter tables appear in Appendix E, Tables 5-6.

---

> ### Author Response · Authors · 2025-11-22
> **Rebuttal, continued**
>
> ## Questions
>
> > **Q1**. Why is $\alpha$ defined as a step function with $\alpha(s)=1$ for $s<1/2$ and 0 otherwise? Would a smooth interpolation such as linear $\alpha(s)=1-s$ can produce different results, and how sensitive are the results to this choice?
>
> The method was originally derived based on the simple observation that the MAP performs well to first order with a small cost, precisely where BMC methods suffer. So, we want to “be closer to the MAP(s)”, and in a way that is robust and controllable, rather than relying on not forgetting the initial condition (something that we ordinarily want to do with MCMC). See also comments below about distinguishing from a pure initialization strategy.
>
> We explored smooth link function, e.g. a linear interpolation $\alpha(s)=1-s$, like [Paulin, etal, 2025]. We found this does not perform as well. See Table 1 "S-HMC(linear)" in the Global rebuttal above. The point is that we will almost always want to operate in the small $s$ regime and we want the prior mean to be the MAP. We provide a sketch of a theoretical reason for this at the end of Sec 3 (moved up from Appendix C). Any narrow smooth link function can preserve this property; we have clarified this in the revision.
>
> > **Q2**. For the meta-classifier, how many posterior samples were used to compute expectations and variances of  $p_{max} (x,\theta)$ and $\Delta_{max}(x,\theta)$? How is total entropy estimated for MAP and deep ensemble models that produce deterministic predictions?
>
> $NP$ posterior samples are always used. For example, in Table 1 (paper and Global, above), $N=10$ and $P\in \{1,8\}$. Remarkably, the total (marginal) entropy is actually based only on the *posterior predictive*, $\hat{p}(y|x)$, just like other first order frequentist metrics (Accuracy/ECE/NLL/Brier/etc). For a single input it is simply given by
> $$H_{\sf tot}(x)=-\sum_{y\in Y}\log(\hat{p}(y|x)) \hat{p}(y|x) .$$ For the MAP, this is the simple plug-in predictive estimator $\hat{p}(y|x) = p(y \mid x, \theta_{MAP})$. For all particle-based methods, including deep ensembles, it is a Monte Carlo average over the ensemble of particles, $$\hat{p}(y|x) = \sum_{i=1}^N w^i p(y \mid x, \theta^{(i)})  ,$$
> with $w^i=1/N$ unless SMC. It is only the epistemic entropy (mutual information between the prediction and the parameter posterior) which is 0 for deterministic predictions. That is estimated by
> $$
> H_{\sf ep}(x)= H_{\sf tot}(x) + \frac1N\sum_{i=1}^N \sum_{y\in Y}\log({p}(y|x, \theta^{(i)})){p}(y|x, \theta^{(i)})  ,
> $$
> and the second term, the minus aleatoric (conditional) entropy, is identically $-H_{\sf tot}(x)$ for deterministic models.
>
> > **Q3**. Appendix 5.1 mentions preliminary results on GPT‑2; can you provide more details on applying SBMC to language models? What challenges may arise when scaling to tens of millions of parameters?
>
> Scale is always tricky for LLMs, even for simple training and inference. But there is nothing too onerous for our method in comparison to the usual requirements for possibly considering data/model parallelism. One attractive aspect of our formulation is that performance is quite good with very parameter-efficient fine-tuning approaches, which *keep the cost close to that of the deterministic base model*. For example, for GPT-2 with 124M parameters, we achieve good performance by fine-tuning an ensemble of 10x 0.2M parameter adapters, i.e. a total of 2M parameters for the whole ensemble. The cost is small in comparison to the frozen parameters. So ultimately, the UQ overhead is also small in comparison to deterministic inference – a must for UQ in the age of LLMs.
>
> The real challenge for LLMs is lifting our results beyond token-level predictions as we do here, to semantic epistemic uncertainty. There is not a unique way to do that. We have some ideas, but it is the subject of ongoing investigation. Hence we defer it to future work, and simply present a POC in Section 3 (formerly in the conclusion).

---

### Author Response · Authors · 2025-11-22
**Global (part 1)**

# Global (part 1)

We thank the reviewers for their detailed and constructive feedback. We have carefully addressed all concerns at both conceptual and empirical levels, and provide new experiments, improved clarifications, and corrected wording where requested. The revision will include
- (i) explicit ablation distinguishing MAP-initialized sampling from anchored SBMC sampling (R3 Q1),
- (ii) new comparisons using
    - a linearly scaled MAP prior as in SMS-UBU (R3 Q2)
    - full SMS-UBU itself (R3 Q2)
    - 2x epochs for MAP/DE and (1/2)x epochs for SBMC, to exactly match cost (R1-2)
    - SWAG (R2)
- (iii) a clearer theoretical interpretation of our anchored posterior (moved from Appendix C to the main text),
- (iv) calibration metrics (ECE, Brier) and clarification of compute budget definition, and
- (v) a new figure explaining the meta-classifier pipeline (requested by R1 and R3).
- (vi) *Full ResNet20 on CIFAR results are in progress and will be uploaded during the discussion phase. Forthcoming results are indicated with italics and $(\dagger)$.*

We also carefully respond in detail to every Weakness and Question in the individual responses.


## Main Points

We refer to the reviewers as follows
R1: H5Gj
R2: WB64
R3: otV6

The main points we address are as follows.
- **Novelty:** R2(W1,Q4) and R3(W1,Q1-2) had concerns about novelty, which we address in detail below and in Table 1 below [see DEI-MCMC, SWAG, and S-HMC(linear, single MAP)]. In particular, we give clear and definitive responses to R3 (Q1-2), who states that, "Satisfactory responses, in particular to [Q] (1-2), would in my opinion significantly strengthen the paper." This also relates to the concerns of R1(W3,Q1) around continuous link function for the anchor.
- **Theory:** R1(W1,Q1/3) and R2(W2,Q5) raised concerns around the theoretical basis of the method and choice of anchor. We provide a sketch of the theoretical argument in Appendix C, which will be highlighted more clearly in the revision, and discuss further below and in detailed replies.
- **Clarity:** R3(W2-4,Q3-4) raised concerns around clarity of specific parts, which have been addressed in the detailed replies. R1(Q2,W4) and R3(W3) requested clarification on the meta-classifier setup, which is now included in a **new Figure 3** in the paper. R2(W6) requested clarification that we indeed evaluate frequentist calibration errors (ECE/Brier) and negative log-likelihood on a held-out test set *for every example*.
- **Budget:** We state the following, on line 268, "Note that SBMC methods require the MAP estimator, so their total time cost is roughly **double**." However, R1(W2) and R2(W5,Q3) pointed out that some statements about compute time of SBMC do not account for  the MAP computation, which is unclear. We corrected these, and also included experiments with half the epochs for SBMC and double the epochs for MAP/DE for 1:1 comparisons (See MAP(2x), DE(2x), S-HMC(1/2x) in Table 1, below).
- **More experiments.** *All referees commended the extensive empirical evaluation*, however R2(W3-4,Q1/3) also requested SWAG and a larger-scale baseline, and R3(W1,Q2) requested SMS-UBU and DEI-MCMC. SWAG, SMS-UBU, DEI-MCMC, and S-HMC (linear -- SMS-UBU except with consistent HMC) are presented in Table 1 below and in the paper. *Full ResNet20 on CIFAR10 is forthcoming shortly $(\dagger)$.*

---

> ### Author Response · Authors · 2025-11-22
> **Global Table**
>
> ### Table 1: Comparison of methods on MNIST7 test data.  $N=10$.
>
> #### (a)  s = 0.1
>
> | Method | P | Time Cost | Accuracy ↑ | NLL ↓ | H_ep correct | H_ep incorrect | H_ep OOD |
> |--------|---|---|------------|-------|--------------|---------------|----------|
> | SWAG | 1 | 160(+) | 92.3±0.365 | 0.267±0.017 | 0.001±0.000 | 0.008±0.001 | 0.009±0.001 |
> | MAP (2x)                         | 1 | 320 | 92.1±0.264 | 0.260±0.010 | 0                       | 0                         | 0                   |
> | Deep Ens (2x)                                | 1 | 320 | 92.2±0.157 | 0.252±0.005 | 0.007±0.000             | 0.041±0.003               | 0.104±0.013         |
> | DEI-HMC | 1 | 160+160 | 91.4±0.037 | 0.310±0.001 | 0.034±0.001 | 0.104±0.002 | 0.237±0.003 |
> | SMS-UBU | 1       | 160+160    | 3200   | 92.6±0.067   | 0.247±0.003  | 0.046±0.004  | 0.179±0.007  | 0.265±0.018  |
> | **S-HMC** (1/2x)                       | 1 | 80+80 | 93.0±0.057 | 0.242±0.002 | 0.059±0.001             | 0.243±0.004               | 0.417±0.013         |
> | **S-HMC** | 1 | 160+160 | 93.0±0.166 | 0.232±0.002 | 0.056±0.001 | 0.264±0.002 | 0.463±0.009 |
> | **S-HMC (single MAP)** | 1 | 160+160 | 93.2±0.071 | 0.226±0.001 | 0.056±0.001 | 0.277±0.006 | 0.453±0.014 |
> | **S-HMC (linear)** | 1 | 160+160 | 92.9±0.140 | 0.239±0.003 | 0.059±0.001 | 0.249±0.002 | 0.450±0.011 |
> | SMS-UBU$_\parallel$ | 8       | 160+160    | 25600  | 92.6±0.107   | 0.247±0.002  | 0.055±0.001  | 0.201±0.001  | 0.316±0.003  |
> | DEI-HMC∥ | 8 | 160+160 | 91.6±0.000 | 0.308±0.001 | 0.038±0.000 | 0.114±0.001 | 0.230±0.006 |
> | **S-HMC∥** (1/2x)            | 8 | 80+80 | 93.1±0.071 | 0.237±0.001 | 0.069±0.000             | 0.272±0.002               | 0.484±0.005         |
> | **S-HMC∥** | 8 | 160+160 | 93.1±0.085 | 0.231±0.002 | 0.070±0.000 | 0.299±0.002 | 0.531±0.011 |
> | **S-HMC∥ (single MAP)** | 8 | 160+160 | 93.6±0.023 | 0.217±0.000 | 0.065±0.000 | 0.301±0.001 | 0.533±0.006 |
> | **S-HMC∥ (linear)** | 8 | 160+160 | 93.1±0.069 | 0.240±0.002 | 0.073±0.000 | 0.284±0.002 | 0.518±0.010 |
>
> ---
>
> #### (b)  s = 0.25
>
> | Method | P | Time Cost | Accuracy ↑ | NLL ↓ | H_ep correct | H_ep incorrect | H_ep OOD |
> |--------|---|---|------------|-------|--------------|---------------|----------|
> | **S-HMC** (1/2x)                        | 1 | 80+80 | 92.8±0.081 | 0.263±0.001 | 0.101±0.003             | 0.357±0.009               | 0.585±0.008         |
> | **S-HMC** | 1 | 160+160 | 92.8±0.190 | 0.257±0.006 | 0.113±0.005 | 0.423±0.008 | 0.684±0.013 |
> | **S-HMC (single MAP)** | 1 | 160+160| 93.14±0.131 | 0.240±0.003 | 0.095±0.002 | 0.408±0.006 | 0.615±0.015 |
> | **S-HMC (linear)** | 1 | 160+160| 92.8±0.160 | 0.269±0.004 | 0.095±0.002 | 0.333±0.003 | 0.611±0.019 |
> | **S-HMC∥** (1/2x)             | 8 | 80+80 | 93.1±0.001 | 0.256±0.002 | 0.122±0.001             | 0.403±0.003               | 0.671±0.006         |
> | **S-HMC∥** | 8 | 160+160| 93.1±0.050 | 0.249±0.001 | 0.138±0.002 | 0.479±0.005 | 0.751±0.013 |
> | **S-HMC∥ (single MAP)** | 8 | 160+160| 93.68±0.026 | 0.225±0.000 | 0.112±0.000 | 0.455±0.002 | 0.739±0.006 |
> | **S-HMC∥ (linear)** | 8 | 160+160| 93.1±0.050 | 0.267±0.001 | 0.115±0.001 | 0.377±0.003 | 0.666±0.014 |
>
>
>
>
> ---
> ####
> ### Table 2. Area under the 2-level estimator curve Ablations for MNIST7. $N=10$.
>
>
> | P | Method             | All fea. | Ep. fea. | ${\bf p_{\sf max}}$ | ${\bf H_{\sf{ep}}}$ |
> |-------|--------------------|--------------|--------------------|-------------------|------------------|
> | -     | MAP                | 0.717        | 0.501              | 0.701             | 0.501            |
> | -     | DE                 | 0.795        | 0.790              | 0.734             | 0.789            |
> | 1     | S-SMC$_\parallel$  | 0.747        | 0.738              | 0.690             | 0.738            |
> | 8     | S-SMC$_\parallel$  | **0.820**    | **0.820**          | 0.718             | **0.818**        |
> | 1     | S-HMC$_\parallel$  | 0.804        | 0.801              | 0.712             | 0.798            |
> | 8     | S-HMC$_\parallel$  | **0.822**    | **0.822**          | 0.716             | **0.819**        |

---

> ### Author Response · Authors · 2025-11-22
> **Global (part 2)**
>
> # Global (part 2)
>
> ## Novelty
>
> SBMC is not the first to propose anchoring or MAP initialization. Instead, its contribution lies in showing that anchoring parallel short-run BMC samplers *directly to the MAP* (or SWA) yields a computationally efficient Bayesian method that is distinct from existing methods, and achieves competitive accuracy, NLL, ECE, Brier, and improved epistemic uncertainty via an ensemble of samples, under practical compute budgets.
>
> **Linear interpolation.** The closest work to ours is [Paulin etal, 2025], who approximate *unadjusted Langevin chains* using a particular numerical method and anchor to a linear interpolation of a point estimator (SWA in that case, but it could be the MAP). We show in Table 1 that this linear interpolation strategy, i.e. the mean is scaled as $(1-s)\theta_{MAP}$ [S-HMC (linear)], underperforms in comparison to centering on the MAP itself as we do. We also provide a sketch of a theoretical justification for this choice in Appendix C. *Full SMS-UBU experiments will be added to the rebuttal soon $(\dagger)$*.
>
> **Initialization.** The works [1], [5], and others have already proposed to initialize chains at MAP estimators. The experiments in Table 1 show that this strategy [DEI-HMC] under-estimates uncertainty in comparison to our method, and performs generally worse. This demonstrates that anchoring provides a distinct benefit beyond initialization alone.
>
> **One or many MAPs.** We provide experiments using one MAP estimator for all chains to compare with experiments using a fresh MAP estimator for each chain. These choices perform comparably.
>
> **Meta-classifier for abstention.** We also introduce a novel meta-classifier that uses epistemic-derived features to predict confidence in model outputs--rather than detect OOD inputs. A two-level estimator then determines when to abstain based on predicted reliability, enabling principled selective prediction. Prior methods rely on frequentist heuristics or a single uncertainty score, whereas ours explicitly learns confidence from structured epistemic information estimated via SBMC. A new Figure 5 in the paper (only update) provides a clear schematic for the meta-classifier pipeline. See also Table 2 above for an illustration of the power of these features in comparison to $p_{\sf max}$ score alone.
>
> **Budget.**  We acknowledge that "approximately the same" is imprecise, and will replace with "approximately double", to properly account for the required cost of pre-computing the MAP(s) -- this is consistent with what we stated clearly elsewhere in the paper and in the Table. See also experiments with half the epochs for S-HMC (1/2x) and with twice the epochs for MAP and DE (2x) in Table 1 above, to demonstrate the regime of exactly equal computation. Performance degrades somewhat, but remains comparable in both cases and does not impact the conclusions.
>
>
>
> ## Theory
> To make sense of the approximation, and distinguish it from linear interpolation, one can leverage a sequential pseudo-Bayesian interpretation. We sketched out this argument in Appendix C. The argument also justifies the default choice $s=0.1$, given that the scaling factor $a$ is arbitrary. We will reference this in the main text in the revision, with an even shorter synopsis and pointer.
>
> The approximation argument could be improved and tightened up by using a (KFAC-)Laplace approximation of the scaled/tempered posterior as a prior (instead or) as an intermediate approximation. But we don't believe that would materially change the practical takeaway.
>
> **Continuous link function** is possible, but we would favour a steep sigmoid rather than linear. See the previous section and table. The point is that the more interesting practical regime is small $s$, and the scaling factor should be $\approx 0$ so that the prior mean is the MAP. See also more detailed responses below.
>
> ## Baselines and scale
>
> SWAG, SMS-UBU, and DEI-MCMC are presented in Table 1 above and in the revision. SBMC outperforms both methods.
> *Experiments are forthcoming on ResNet-20 on CIFAR $(\dagger)$.*
>
> ## Final Statement
>
> Together, the new ablations, theoretical clarification, and expanded baselines directly address the key concerns raised by Reviewers, particularly around novelty, initialization, baseline comparison, and clarity. The results collectively show that SBMC offers a practically useful method that is demonstrably distinct — both algorithmically and empirically — from existing methods.

---

### Author Response · Authors · 2025-12-03
**Brief High Level Summary for New AC**

We thank the new Area Chair for taking on the responsibility of making a decision about our manuscript. Given the circumstances, we provide here a short and self-contained summary. R1 was originally positive (6), while R2 was borderline negative (4), and R3 was strongly negative (2). We carefully and thoroughly addressed all of the concerns of all of the reviewers (summarised below), and produced a revised version of the manuscript.

R3 engaged heavily, and increased their score to borderline negative (4, as stated explicitly below), and articulated several very specific concerns that were preventing them from recommending acceptance. We have again very carefully addressed each of those concerns in the Final rebuttal below, including several new experiments. We would ideally have included a few more numerical experiments, which we plan to include (along with one other unfinished experiment) in the camera ready version, if accepted.

The changes and additional experiments described below have further reinforced our original conclusions and significantly improved the manuscript. We believe that R1 and R2 would have increased their scores as well.


## Changes

The primary changes are as follows. All these changes have been implemented in the revised version of the manuscript and uploaded to OpenReview. We
- Clarified the novelty of the method,
- Improved overall clarity of the narrative and various other aspects,
- Included a graphical schematic for meta-classifier pipeline in a new Figure 3,
- Moved the theoretical argument from the Appendix into the main text, and
- Conducted several further experiments, as outlined below.

The following additional comparison experiments were added, in Tables 1-5 here and Tables 1,3,4 in the revision, to resolve issues of novelty and value:
- (i) Deep ensemble initialized HMC (DEI-HMC).
- (ii) Linearly-scaled MAP prior as in SMS-UBU, and the full SMS-UBU itself.
- (iii) Stochastic weight average Gaussian (SWAG).
- (iv) MAP/DE with 2$\times$ epochs and SBMC with (1/2)$times$ epochs to observe exactly equal computation time.
- (v) S-HMC$_\parallel$ initialized at a single MAP vs. a fresh MAP for each chain.
- (vi) many more finescale additions to further clarify the power of SBMC for quantifying confidence and detecting OOD examples, and the value of the meta-classifier for abstention decisions. This has helped us to really clarify that
  - SBMC directly delivers a strong (1-stage) confidence score in the epistemic entropy $H_{\sf ep}$, and that
  - the 2-stage meta-classifier pipeline provides additional robustness at a modest cost, and also
  - a valuable abstention decision rule which can be used to build a 2-level estimator. We quantify the power of this estimator with the area under the 2-level curve (AU2LC) of p(abstain) vs accuracy.

The following experiments are underway and will be included at the latest in the camera ready version, if our paper is accepted.
- (i) DEI-HMC and S-HMC with more elaborate adaptive tuning.
- (ii) PostNet and NatPN OOD confidence scores in comparison to our single stage score $H_{\sf ep}$, in terms of AUCROC and AUCPR.
- (iii) A larger example (full ResNet on CIFAR) verifying that the method is indeed scalable, and continues to deliver UQ for the price of SGD as model size and depth grows.

---

### Meta-Review · Area_Chair_AnVV · 2026-01-06

**Summary:**

* SBMC achieves better UQ than deep ensembles for the same computational budget.
* The method scales linearly with parallel hardware because it eliminates communication between chains.
* Anchoring the sampling to a point estimate provides better results than simply starting from that estimate.
* The method is effective on larger models (eg. GPT-2 adapters, ResNet architectures)

Scalable Bayesian Monte Carlo (SBMC) bridges the gap between fast ensembles and slow Bayesian sampling. It uses a modified prior to anchor the posterior near a pre-computed point estimate. The authors argue that this simplifies the energy landscape and allows multiple short-run chains to generate useful uncertainty estimates in parallel.

The method is compared against deep ensembles, Laplace approximations, and SWAG across image and text tasks. The results show that the method provides better epistemic uncertainty features for downstream tasks like predicting when a model is likely to be incorrect.

**Reviewer Concerns:**

**Addressed by rebuttal**

* [Core] The computational cost of the point estimate was not fully accounted for in the initial comparison.
* [Core] The performance gains might have come from simple initialization at the point estimate rather than the anchoring mechanism.
* [Core] Evaluation on out-of-distribution data might be a poor proxy for the quality of uncertainty.
* [Core] Comparisons were missing against popular baselines like SWAG etc..

**Still outstanding**

* [Core] The use of a single leapfrog step in HMC comparisons may be an unusual baseline choice.
* [Core] The method relies on a single basin and may not explore the full multimodal posterior.
* [Non-core] Minor formatting errors in the revised manuscript (easy fix)

The rebuttal period was effective. The authors added cost-matched experiments showing that the method still outperforms deep ensembles. The ablation study confirming that anchoring outperforms simple initialization. The super-far out-of-distribution experiments on CIFAR data for an MNIST model demonstrated the robustness of the meta-classifier.

**Reviewer Scores:**

* Reviewer ID: H5Gj
* Original score: 6
* Estimated score shift: Unchanged
* Justification: The reviewer was already positive and their requests for a hyperparameter table and schematic were fully satisfied.

* Reviewer ID: WB64
* Original score: 4
* Estimated score shift: Increase
* Justification: The authors provided the requested SWAG comparison and clarified the wall-clock time budget, which were the reviewer's primary criticisms.

* Reviewer ID: otV6
* Original score: 2
* Estimated score shift: Increase
* Justification: The reviewer already acknowledged the value of the new ablations and increased their internal score to a 4 during the discussion.

The reviewing panel was initially split due to concerns about novelty and the fairness of the computational budget. The extensive new data provided during the rebuttal phase addressed the most critical technical doubts. While Reviewer otV6 remains skeptical about the degree of novelty, the practical utility and the success of the new "super-far" out-of-distribution tests provide strong evidence of the work's value.

---

### Decision · Program_Chairs · 2026-01-26

Reject